

# Marginal quenches and drives in Tomonaga-Luttinger liquids

**Shouvik Datta**[1⋆]**, Bastien Lapierre**[2†]**, Per Moosavi**[3‡] **and Apoorv Tiwari**[4∘]

**1** Department of Theoretical Physics, CERN, 1 Esplanade des Particules,
1211 Geneva 23, Switzerland
**2** Department of Physics, University of Zurich, Winterthurerstrasse 190,
8057 Zürich, Switzerland
**3** Institute for Theoretical Physics, ETH Zurich, Wolfgang-Pauli-Strasse 27,
8093 Zürich, Switzerland
**4** Department of Physics, KTH Royal Institute of Technology,
106 91 Stockholm, Sweden

⋆ sdatta@cern.ch , † bastien.lapierre@uzh.ch ,
‡ pmoosavi@phys.ethz.ch , ∘ apoorvt@kth.se

## Abstract

We study Tomonaga-Luttinger liquids thrown out of equilibrium by marginal deforma-
tions in the form of interaction modulations. This is modeled by quenching or peri-
odically driving the Luttinger parameter or, equivalently, the compactification radius of
the free boson conformal field theory between two different values. We obtain exact
analytical results for the evolution of the Loschmidt echo and observables such as the
particle and energy densities. Starting from generic initial states, the quench dynamics
are shown to exhibit revivals and temporal orthogonalities. For the periodic drive, we
show stability or instability of time-evolved physical quantities dependent on the drive
parameters. We also compare the corresponding marginally deformed thermal density
matrices by non-perturbatively evaluating their Rényi divergence as a Euclidean quench.
All the dynamics are shown to be crucially dependent on the ratio of the Luttinger pa-
rameters, which corresponds to the Zamolodchikov distance in the space of marginal de-
formations. Our setup is equivalently interpreted as the dynamics of the bosonic string
upon instantaneous changes of the target-space radius.

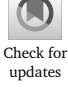

# 1 Introduction

Quantum quenches and Floquet drives are simple yet fruitful protocols for understanding physics out of equilibrium. In this paper we study the dynamics of gapless quantum many-body systems in one spatial dimension called Tomonaga-Luttinger liquids (TLLs) [1] under interaction quenches and drives. Their low-energy description is given by the simplest conformal field theory (CFT) that belongs to a continuous family of CFTs related by marginal deformations [2], namely 1+1-dimensional compactified free bosons. In this case, the continuous parameter that labels different CFTs is the compactification radius, which corresponds to the Luttinger parameter for TLLs. Our interaction modulations are modeled precisely by quenching or periodically driving this parameter between two different values.

The recent advent of experimental platforms capable of probing the non-equilibrium dynamics of quantum many-body systems has fueled an interest in related theoretical questions. These pertain to the physics of thermalization, prethermalization, equilibration to non-thermal states, and non-equilibrium phenomena such as quantum revivals, dynamical quantum phase transitions, Floquet topological phases, and discrete time crystals, to name a few [3,4]. Due to the inherent complexity of non-equilibrium quantum physics, it is often difficult to make analytical progress. In this context, CFTs in general and TLLs in particular provide examples that can be studied out of equilibrium by exact analytical means. The TLL description in terms of free bosons is also relevant to experiments, such as quasi-one-dimensional condensates of

ultra-cold atoms [5,6]. Moreover, the free boson CFT and its multi-component generalizations play a pivotal role in high-energy and mathematical physics, e.g., in the formulation of bosonic string theory [7] and as an exactly solvable quantum field theory (QFT) [8–10].

In general, quenches of critical theories are categorized as massive or massless, depending on whether the initial state has correlations with exponential or power-law decay, respectively. Massive or short-range correlated states can be well-approximated by suitable conformal boundary states, consequently, massive quenches have been studied extensively using powerful techniques in boundary CFT [11,12]. On the other hand, massless quenches pertain to initializing the system in a state that corresponds to one critical Hamiltonian and abruptly changing to another. Examples of works in this direction include interaction or marginal quenches in TLLs [13–26], the sine-Gordon model [27], quantum spin chains [28–31], the one-dimensional Hubbard model [32], and the Lieb-Liniger model [33–38]. Despite much progress in the study of quench dynamics for CFTs, few exact results are known for quenches from arbitrary excited states. Furthermore, the role played by the geometry of the space of marginally deformed theories [39] in any non-equilibrium setting is yet to be identified.

For driven critical systems, one example that has received recent attention is to periodically switch between CFTs with different spatially deformed Hamiltonians [40–45]. Such systems were shown to host rich dynamics containing stable and unstable phases, which have been numerically verified in spin-chain realizations of TLL theory. Motivated by these developments, a natural question arises: What is the dynamics of a CFT when subjected to modulations of marginal couplings? In a sense, this is a more canonical class of deformations since they do not break conformal invariance. A simple realization of such a protocol consists of periodically varying the Luttinger parameter in time for TLLs, as studied in [46–48]. However, many aspects, such as the nature of dynamical phases, transitions between them, and their signatures in physical quantities have not been analyzed.

In the present paper, we harness underlying symmetries to derive a number of exact analytical results for dynamical quantities for TLLs subjected to marginal quenches and drives. Among these are the evolution of particle and energy densities and the Loschmidt echo (return probability) starting from arbitrary excited or thermal states. For the quench, we show that the results exhibit revivals and periodic orthogonality signaling dynamical quantum phase transitions [49]. For the periodic drive, we find stable and unstable dynamical phases and infer the critical exponents of natural order parameters at the phase boundary. A key feature common to all our results is a dependence through the ratio of the two Luttinger parameters. This ratio also corresponds to the well-known Zamolodchikov distance in the space of CFTs related by marginal deformations [39]. Besides studying dynamical properties, we also use and extend our formalism to evaluate the Rényi divergence [50] and relative entropy [51] between thermal states of two TLLs with different Luttinger parameters. The Rényi divergence is a one-parameter generalization of the relative entropy, which serves as an information-theoretic measure of the distance between two density matrices, and our result establishes a relation between this distance and the Zamolodchikov distance for marginally deformed TLLs.

## Setup and methods

In TLL theory, all details are encapsulated in two parameters: The propagation velocity $v$ and the Luttinger parameter $K$ encoding the interactions of the original system. The Hamiltonian can be written as[1]

$$H_{v,K} = \frac{1}{2\pi} \int_{-L/2}^{L/2} dx : \left( \frac{v}{K} [\pi \Pi(x)]^2 + vK[\partial_x \varphi(x)]^2 \right) : - \frac{\pi v}{6L}, \tag{1}$$

---

[1]We use units so that $\hbar = k_B = 1$.

for a bosonic field $\varphi(x) = \varphi(x) + 2\pi$ with $x$ on the circle of length $L$, where $[\varphi(x), \Pi(y)] = i\delta(x - y)$ and $:\cdots:$ denotes Wick ordering. From a path integral perspective, this corresponds to the action[2]

$$S = \frac{R^2}{4\pi\alpha'} \int d^2x \, (\partial^\mu \varphi)(\partial_\mu \varphi) = \frac{1}{4\pi\alpha'} \int d^2x \, (\partial^\mu X)(\partial_\mu X), \tag{2}$$

describing free bosons $X = R\varphi$ with compactification radius $R$ satisfying

$$K = \frac{R^2}{2\alpha'}, \tag{3}$$

where $\alpha'$ is referred to as the string tension in bosonic string theory [7]. We recall that $\alpha'$ has dimension length$^2$ and is commonly set as $\alpha' = 2$, which we will also do here for simplicity.

In this paper, we study TLLs out of equilibrium by quenching or driving the interactions. This is modeled by changing the Luttinger parameter $K$ in the Hamiltonian in (1) between two different values, $K_1$ and $K_2$, see Figs. 1(a) and 1(b). For consistency, we also change the velocity $v$ between $v_1$ and $v_2$, although these can conveniently be absorbed into dimensionless times

$$\begin{cases} \tau_1 = v_1 t_1 / L, \\ \tau_2 = v_2 t_2 / L, \end{cases} \qquad \begin{cases} q_1 = e^{-2\pi i \tau_1}, \\ q_2 = e^{-2\pi i \tau_2}, \end{cases} \tag{4}$$

using which all our results for the drive can be stated (and similarly for the quench). On the other hand, changing $K$ is non-trivial. Indeed, $H_2 = H_{v_2, K_2}$ can be shown to correspond to a $J\bar{J}$ deformation of $H_1 = H_{v_1, K_1}$. Within the compactified free boson formulation, this is a marginal deformation that effectively changes the compactification radius from $R_1$ to $R_2$. Alternatively, within TLL theory, $K$ can be shown to determine the partitioning of excitations (quasi-particles) into right or left movers propagating with velocity $v$ and $-v$, respectively, and changing from $K_1$ to $K_2$ thus corresponds to a repartitioning.

The two non-equilibrium protocols we study are:

1. **Quantum quench**. Consider an initial state $\hat{\rho}$ defined with respect to the undeformed Hamiltonian $H_1$. For instance, its ground state $\hat{\rho} = |\Omega\rangle\langle\Omega|$, an arbitrary excited state $\hat{\rho} = |\Psi\rangle\langle\Psi|$ obtained by acting on $|\Omega\rangle$ (or any primary state) with bosonic creation operators, or a thermal state $\hat{\rho} = e^{-\beta H_1}/\text{Tr}[e^{-\beta H_1}]$ with inverse temperature $\beta$. We study the expectation values of observables when the system is evolved in time $t$ under $H_2$:

   $$\text{Tr}\left[\hat{\rho}\, e^{iH_2 t} \mathcal{O} e^{-iH_2 t}\right], \tag{5}$$

   where $\mathcal{O}$ is an operator such as the energy density associated with the Hamiltonian $H_1$, see Fig. 1(a). Another quantity of interest that we study is the Loschmidt echo $L_\Psi(t) = \left|\langle\Psi|e^{-iH_2 t}|\Psi\rangle\right|^2$ of the state $|\Psi\rangle$ following a quantum quench.

2. **Floquet drive**. We consider a two-step drive of a TLL such that the Hamiltonian switches periodically between $H_1$ and $H_2$, see Fig. 1(b). The Floquet operator describing such a time evolution is

   $$U_F = e^{-iH_1 t_1} e^{-iH_2 t_2}, \tag{6}$$

   where $t_1, t_2 \in \mathbb{R}$ are parameters of the drive. We study the stroboscopic (discrete) time evolution

   $$U_F^{-M} \mathcal{O} U_F^M \tag{7}$$

   of observables $\mathcal{O}$ for an integer number $M$ of cycles as well as the Loschmidt echo $L_\Psi(M[t_1 + t_2]) = \left|\langle\Psi|U_F^M|\Psi\rangle\right|^2$ of the state $|\Psi\rangle$.

---

[2]Here $\varphi = \varphi(x, t)$ with $X = R\varphi$ taking values on the circle $[0, 2\pi R]$ for $(x, t)$ on the cylinder $[-L/2, L/2] \times \mathbb{R}$. As usual, $x^0 = vt$, $x^1 = x$, $\partial_\mu = \partial/\partial x^\mu$ for $\mu = 0, 1$, and the metric is diag$(1, -1)$.

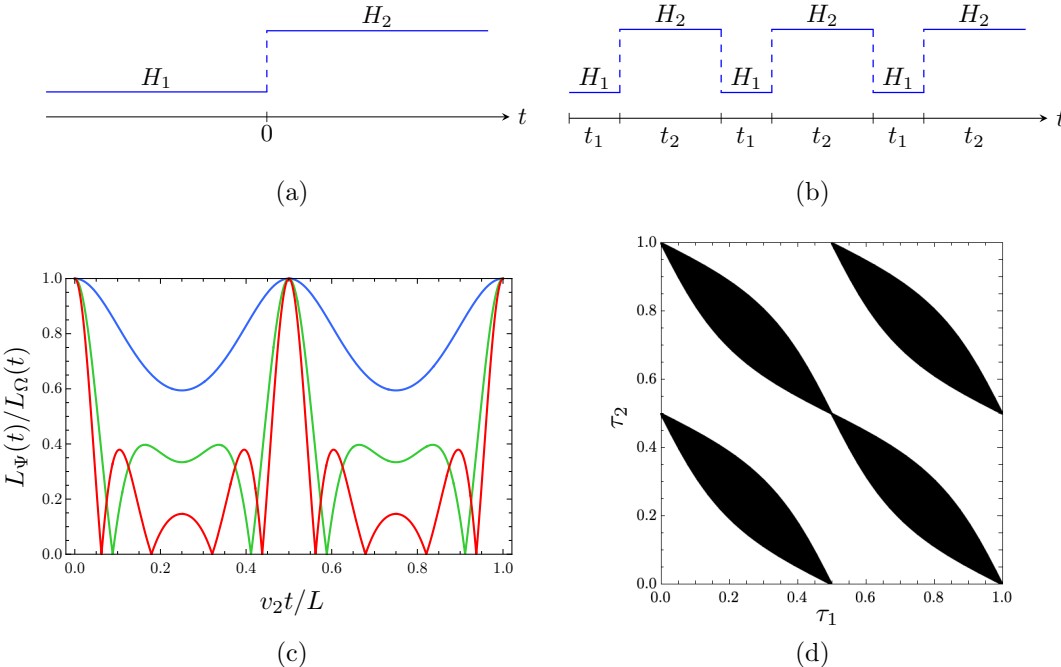

Figure 1: Illustrations of our (a) quantum quench and (b) Floquet drive with (c)–(d) selections of obtained results. (c) Exact zeros signalling dynamical quantum phase transitions in the Loschmidt echo for initial states $|\Psi\rangle$ that mix right- and left-moving excitations (green and red curves) compared to a state that do not mix them (blue curve) following a quench with $K_1/K_2 = 1.2$. (d) Stable (white) and unstable (black) regions in $(\tau_1, \tau_2)$-parameter space from $\mathfrak{su}(1,1)$-stability analysis of the Floquet operator $U_F^{(n)}$ for a single bosonic mode ($n = 1$) corresponding to a two-step drive with $K_1/K_2 = 1.4$.

Our Floquet drive can be seen as a discrete-time version of the continuous interaction drives in [46,47] and as a generalization of the equal-period two-step drive in [17]. We stress that our protocol and presented approach can be directly generalized to an arbitrary number of steps in the drive.

The key to our approach is the well-known existence of a unitary operator [10] that maps $H_2$ to $H_1$ (up to zero modes, which are handled separately) incorporating underlying $\mathfrak{su}(1,1)$-algebraic properties. Indeed, this operator, here denoted $\mathcal{I}_\nu$, implements a Bogoliubov transformation of the theory with $K = K_2$ to the one with $K = K_1$, which can be thought of as a 'rotation' by an 'angle'

$$\nu = \log\sqrt{K_1/K_2} = \log(R_1/R_2).\tag{8}$$

Geometrically, $\nu$ is the Zamolodchikov distance, defined as the geodesic length between two points in the conformal manifold of the compactified free boson CFT, also called Narain moduli space [39]. Strictly speaking, $\mathcal{I}_\nu$ is well defined only with an ultraviolet cutoff, which also has physical interpretations and significance for condensed-matter applications. However, for our purposes, we will mostly avoid such technical details, as every step and result can be repeated or restated with a cutoff in place. Lastly, we note that $\mathcal{I}_\nu$ bears a resemblance to interface operators in boundary CFT [52,53] in that it effectively "glues" two different bosonic theories along time interfaces in our quench and drive protocols.

## Summary of results

In the present paper, we derive and present the following results:

1. **Quantum quench.** We obtain exact analytical results for the Loschmidt echo $L_\Psi(t) = |\langle\Psi|e^{-iH_2 t}|\Psi\rangle|^2$ following a quench from $H_1$ to $H_2$ for any eigenstate $|\Psi\rangle$ of $H_1$. This generalizes earlier results in [17] that were limited to the ground state $|\Omega\rangle$. Our expressions notably factorize into the ground-state result $L_\Omega(t)$ and excitation contributions that feature a hypergeometric function. Besides exhibiting periodic revivals, which were observed previously, we find that excited states $|\Psi\rangle$ that mix right- and left-moving excitations lead to temporal orthogonality in $L_\Psi(t)$ at particular times where the return probability is exactly zero due to the hypergeometric function, signaling dynamical quantum phase transitions arising periodically in time, see Fig. 1(c). We also obtain exact analytical results for the quenched time evolution of the energy density for initial pure and thermal states. Starting from a thermal state at temperature $\beta^{-1}$, we show that the expectation value of the energy density equilibrates at late times to a thermal expectation with an effective temperature $\beta_{\text{eff}}^{-1} = \beta^{-1}(K_1/K_2 + K_2/K_1)/2$.

2. **Floquet drive.** We show stability or instability of time-evolved physical quantities dependent on the drive parameters and the Zamolodchikov distance between the TLLs defining the drive. Specifically, using a decomposition into bosonic modes (of the theory with $K = K_1$), the Floquet operator can be shown to be expressible as a product

$$U_F = \left(\text{zero modes}\right) \times \prod_{n>0} U_F^{(n)}, \qquad U_F^{(n)} = \exp\left(c_0^{(n)} K_0^{(n)} + c_-^{(n)} K_-^{(n)} + c_+^{(n)} K_+^{(n)}\right), \quad (9)$$

where $c_0^{(n)}$ and $c_\pm^{(n)}$ are computable coefficients and $K_0^{(n)}$ and $K_\pm^{(n)}$ are combinations of oscillator modes that satisfy the defining relations of the $\mathfrak{su}(1,1)$ algebra. The squared trace of the individual drive $U_F^{(n)}$ provides a stability measure, which can be obtained using properties of the $\mathfrak{su}(1,1)$ algebra:

$$\sigma_n = \left(\text{Tr}\left[U_F^{(n)}\right]\right)^2 = 4\cosh^2\left(\sqrt{\left(c_0^{(n)}\right)^2 - 4c_+^{(n)} c_-^{(n)}}/2\right), \qquad (10)$$

which is smaller (larger) than 4 if $\left(c_0^{(n)}\right)^2 - 4c_+^{(n)} c_-^{(n)}$ is negative (positive), which in turn corresponds to stability (instability). More concretely, we show that $\sigma_n$ depends on $(\tau_1, \tau_2)$ in (4) and $\nu$ in (8) through

$$\sigma_n = \left(\frac{(q_1^n + q_1^{-n})(q_2^n + q_2^{-n}) + (q_1^n - q_1^{-n})(q_2^n - q_2^{-n})\cosh(2\nu)}{2}\right)^2. \qquad (11)$$

This allows us to straightforwardly draw dynamical phase diagrams for each mode, see Fig. 1(d), with stable (unstable) regions corresponding to $\sigma_n < (>)4$. We show that this is observable in exact analytical results obtained for the Loschmidt echo, starting from any excited state, and in the evolution of the particle and energy densities. We also identify natural order parameters, and, by numerically studying their critical behavior near the boundary between stable and unstable regions, show that they have critical exponents of $1/2$.

3. **Rényi divergence.** We provide a non-perturbative computation of the Rényi divergence

$$D_\alpha(\hat{\rho}_1\|\hat{\rho}_2) = \frac{1}{\alpha-1}\log\text{Tr}\left[\hat{\rho}_1^\alpha \hat{\rho}_2^{1-\alpha}\right] \qquad (12)$$

between thermal states $\hat{\rho}_1$ and $\hat{\rho}_2$ of the two different TLL Hamiltonians $H_1$ and $H_2$, or equivalently of a compactified free boson CFT and its marginally deformed counterpart. While the Rényi divergence is an equilibrium property of TLLs, it can also be seen as a Euclidean quench [54,55], which enables us to use the formalism developed in our (Lorentzian) quantum-quench analysis of TLLs. We remark that the Rényi divergence has several mathematical properties that were recently used to put constraints in addition to the second law of thermodynamics from a holographic perspective [54,55]. Most QFT computations of the Rényi divergence have been perturbative so far. However, as we will show, the Rényi divergence for marginal deformations of a free boson CFT is amenable to a non-perturbative analysis. Upon taking the $\alpha \to 1$ limit in (12), we recover the relative entropy $S(\hat{\rho}_1||\hat{\rho}_2) = \text{Tr}[\hat{\rho}_1 \log \hat{\rho}_1] - \text{Tr}[\hat{\rho}_1 \log \hat{\rho}_2]$, which defines a quantum information-theoretic measure of distinguishability between the two TLLs. We show that the relative entropy between thermal states of two theories with different Luttinger parameters behaves as (setting $v_1 = v_2 = 1$ for simplicity)

$$S(\hat{\rho}_1||\hat{\rho}_2) \approx \frac{\pi L}{3\beta}\sinh^2(\nu), \tag{13}$$

which becomes exact in the thermodynamic limit. This gives a novel non-perturbative relation between an information-theoretic distance measure $S(\hat{\rho}_1||\hat{\rho}_2)$ and the Zamolodchikov distance $\nu$ in (8) in the space of CFTs. Some perturbative relations and similar ideas along these lines were presented previously in [56].

**Organization of the paper**

The rest of this paper is organized as follows. In Sec. 2, we discuss a number of applications of TLL theory to motivate the interpretation of our quench and drive protocols as interaction modulations. We also discuss how our setup translates to the dynamics of the bosonic string. In Sec. 3, we provide the necessary technical background and tools used for computations in the subsequent sections, including justifications for interpreting changes in the Luttinger parameter or the compactification radius as marginal $J\bar{J}$ deformations. In Secs. 4 and 5, we present our main results for quantum quenches and Floquet drives, respectively. In Sec. 6, we present the computation of the Rényi divergence and the relative entropy. Concluding remarks are given in Sec. 7. A regularization based on the Lerch zeta function and certain computational details are deferred to Appendices A and B.

## 2 Applications

To motivate the interpretation of our quench and drive protocols as effectively describing interaction modulations, we briefly discuss a number of applications of TLL theory and recall how the propagation velocity $v$ and the Luttinger parameter $K$ depend on model parameters. At the end of the section, we also briefly describe how our setup concretely translates to the dynamics of the bosonic string.

**Interacting massless fermions – The Luttinger model.** The prototype for TLLs is the Luttinger model of interacting massless fermions in one spatial dimension [8–10]. The fermions are either right or left moving, described by fermionic fields $\psi_+(x)$ and $\psi_-(x)$, respectively, satisfying $\{\psi_r(x),\psi_{r'}(x')^\dagger\} = \delta_{r,r'}\delta(x-x')$ and $\{\psi_r(x),\psi_{r'}(x')\} = 0$ $(r,r' = \pm)$ and suitable

boundary conditions. The Hamiltonian can be written as

$$H = \sum_{r=\pm} \int_{-L/2}^{L/2} dx\; :\psi_r(x)^\dagger (-i r v_F \partial_x) \psi_r(x):$$
$$+ \sum_{r,r'=\pm} \int_{-L/2}^{L/2} dx\; \frac{\pi v_F}{2} \left( \delta_{r,-r'} g_2 + \delta_{r,r'} g_4 \right) :\psi_r(x)^\dagger \psi_r(x)::\psi_{r'}(x)^\dagger \psi_{r'}(x): , \qquad (14)$$

where $v_F > 0$ denotes the Fermi velocity and $g_{2,4}$ are coupling constants satisfying $|g_2| < 2 + g_4$. The notation for the couplings is from 'g-ology' in condensed matter physics, see, e.g., [57], with $g_2$ and $g_4$ corresponding to different four-fermion interaction terms. The Luttinger model is well known to be exactly solvable by bosonization, using which $H$ is mapped precisely to the TLL Hamiltonian in (1) with

$$v = v_F \sqrt{(1 + g_4/2)^2 - (g_2/2)^2}, \qquad K = \sqrt{\frac{1 + g_4/2 - g_2/2}{1 + g_4/2 + g_2/2}}, \qquad (15)$$

see, e.g., [57–59] and references therein. Modulating $g_{2,4}$ in time thus corresponds to our interaction quenches or drives changing $K$ and $v$.

**Quantum XXZ spin chain in the gapless regime.** An example of a one-dimensional lattice model that falls into the TLL class is the spin-1/2 quantum XXZ Heisenberg chain for certain values of the anisotropy. This is a famous Bethe-ansatz integrable model of nearest-neighbor coupled spins described by spin operators $S_j^x$, $S_j^y$, and $S_j^z$ that act on lattice site $j = 1, \dots, N$. These satisfy $[S_j^\alpha, S_{j'}^\beta] = i\delta_{j,j'}\epsilon_{\alpha\beta\gamma}S_j^\gamma$ ($\alpha, \beta, \gamma \in \{x, y, z\}$), where $\epsilon_{\alpha\beta\gamma}$ is the totally anti-symmetric tensor ($\epsilon_{xyz} = 1$), and we impose periodic boundary conditions. The XXZ Hamiltonian is

$$H = -J \sum_{j=1}^{N} \left( S_j^x S_{j+1}^x + S_j^y S_{j+1}^y - \Delta S_j^z S_{j+1}^z \right) - h \sum_{j=1}^{N} S_j^z, \qquad (16)$$

where $J$ is the exchange-coupling strength, $\Delta$ is the anisotropy, $h$ is an external magnetic field, and $L = Na$ with $a$ the lattice spacing. We recall that the anisotropy term corresponds to four-fermion interactions after a Jordan-Wigner transformation. In fact, for $|\Delta| < 1$ and near (but not exactly at) half filling, applying this transformation to the Hamiltonian in (16) and taking a scaling limit effectively yields the Luttinger model in (14), see, e.g., [57, 58]. Indeed, in this regime, the low-energy description is given by TLL theory with

$$v = Ja \frac{\pi}{2} \frac{\sqrt{1 - \Delta^2}}{\arccos(\Delta)}, \qquad K = \frac{\pi}{2[\pi - \arccos(\Delta)]}, \qquad (17)$$

obtained from the exact Bethe-ansatz solution when $h = 0$, see, e.g., [58]. As before, modulations in $\Delta$ corresponds to our interaction quenches or drives changing $K$ and $v$.

**Interacting massless bosons – The Lieb-Liniger model.** Another well-known example of a Bethe-ansatz integrable model is the Lieb-Liniger model of interacting bosons in one spatial dimension. In second quantization, the Hamiltonian is

$$H = \int_{-L/2}^{L/2} dx \left( \frac{1}{2m} \partial_x \Psi(x)^\dagger \partial_x \Psi(x) + c\Psi(x)^\dagger \Psi(x)\Psi(x)^\dagger \Psi(x) \right), \qquad (18)$$

where $m$ is the particle mass, $c \geq 0$ is a repulsive coupling constant, and $\Psi(x)$ is a bosonic field satisfying $[\Psi(x)^\dagger, \Psi(x')] = \delta(x - x')$. If we define the dimensionless coupling $\gamma = 2mc/\rho_0$,

where $\rho_0$ is the density of particles, then $v = v(\gamma)$ and $K = K(\gamma)$ are functions of $\gamma$ for which analytical expressions are not known in general but whose product must equal $v_F = \pi\rho_0/m$. As limiting cases for large and small $\gamma$,

$$v = \frac{v_F}{K}, \qquad K \sim \begin{cases} 1 + \frac{4}{\gamma} & \text{for } \gamma \gg 1, \\ \frac{\pi}{\sqrt{\gamma}}\left(1 - \frac{\sqrt{\gamma}}{2\pi}\right)^{-1/2} & \text{for } \gamma \ll 1, \end{cases} \tag{19}$$

see, e.g., [60, 61]. Once again, modulations in $\gamma$, or rather in $c$ assuming $\rho_0$ is fixed, corresponds to our non-equilibrium protocols changing $K$ and $v$.

**Trapped ultra-cold atoms.** Besides its theoretical significance, TLL theory has direct experimental relevance to low-dimensional quantum many-body systems. Well-known and intensely studied examples are quasi-one-dimensional condensates of ultra-cold atoms. For a single condensate of bosons, such a system can be modeled by the Hamiltonian

$$H = \int_{-L/2}^{L/2} dx \left( \frac{1}{2m} \partial_x \Psi(x)^\dagger \partial_x \Psi(x) + \frac{g}{2} \Psi(x)^\dagger \Psi(x) \Psi(x)^\dagger \Psi(x) + [V(x) - \mu] \Psi(x)^\dagger \Psi(x) \right), \tag{20}$$

where $m$ is the atom mass, $g \geq 0$ is the effective interaction strength, $V(x)$ is the trapping potential, and $\mu$ is the chemical potential, see, e.g., [6, 60–62].[3] In the Thomas-Fermi regime, (20) can be approximated as an inhomogeneous TLL following the harmonic-fluid approach [63], setting $\Psi(x)^\dagger = \sqrt{\rho_0(x) + \pi\Pi(x)}e^{i\varphi(x)}$ and keeping only terms quadratic in the fields, with position-dependent

$$v(x) = \sqrt{\rho_0(x)g/m}, \qquad K(x) = \pi\sqrt{\rho_0(x)/mg}, \tag{21}$$

where $\rho_0(x) = [\mu - V(x)]/g$ denotes the mean-atom-density distribution. The effect of $v(x)$ and $K(x)$ on non-equilibrium dynamics was recently studied in [64]. It would be interesting to study quenched or driven inhomogeneous TLLs modulating $v(x)$ and $K(x)$ in time, which would be directly applicable to trapped ultra-cold atoms. However, this is beyond the scope of the present paper, as we only consider the homogeneous case, but which can be viewed as a first step in this direction. We remark that a related but different question concerns modulated tunnel couplings between pairs of quasi-one-dimensional condensates, see, e.g., [65, 66].

**Quantum circuits.** Another important application of TLL theory is to one-dimensional arrays of superconducting junctions. These have been proposed to simulate TLLs, the map between the parameters given by

$$v \sim a\sqrt{2E_{C_0}E_J}, \qquad K \sim \frac{1}{2\pi}\sqrt{\frac{2E_{C_0}}{E_J}}, \tag{22}$$

to lowest order in the regime $E_J \gg E_{C_0}$, where $E_J$ is the Josephson energy, $E_{C_0}$ is the charging energy, and $a$ is the array spacing, see, e.g., [67, 68]. It would be interesting if an array of driven junctions could be realized to simulate quenches and drives in TLLs.

**String theory.** The single compactified free boson in (2) also describes the closed bosonic string with target space being a circle of radius $R$. In this context, the bosonic field $\varphi$ plays the role of the target-space coordinate while $x$ and $t$ are the worldsheet coordinates. A sudden

---

[3]This model is that of a trapped Lieb-Liniger gas with $c = g/2$ using the notation in (18).

change in the radius from $R_1$ to $R_2$, with $R_2 > R_1$, realizes a toy scenario of sudden inflation. From a purely field-theoretic standpoint, one can imagine studying quenches caused by current-current deformations of more general sigma and WZW models [69]. These are integrable deformations and, therefore, the quench dynamics should be tractable. The analysis we are about to present is a first step in this direction.

## 3 Algebraic framework and Bogoliubov transformations

To establish our notation and conventions, following [70–72], we recall that the TLL Hamiltonian in (1) can equivalently be written as

$$H_{v,K} = \int_{-L/2}^{L/2} dx \, v \big[ T_+(x) + T_-(x) \big],$$
(23)

using the right- and left-moving components $T_+(x)$ and $T_-(x)$ of the energy-momentum tensor in light-cone coordinates. The latter can, in turn, be expressed in terms of current operators

$$T_\pm(x) = \frac{\pi}{K} :J_\pm(x)^2: - \frac{\pi}{12L^2},$$
(24)

where

$$J_\pm(x) = \frac{1}{2\pi} \Big[ \pi \Pi(x) \mp K \partial_x \varphi(x) \Big]$$
(25)

are the right- and left-moving components of a conserved U(1) current in TLL theory. In general, consider a 1+1-dimensional CFT with central charge $c$ (in our case $c = 1$) and a conserved U(1) current with $K$ appearing as the current-algebra central charge. Passing to Fourier space,

$$T_+(x) = \frac{2\pi}{L^2} \sum_{n=-\infty}^{\infty} e^{+2\pi i n x/L} \left( L_n - \frac{c}{24} \delta_{n,0} \right), \qquad J_+(x) = \frac{1}{L} \sum_{n=-\infty}^{\infty} e^{+2\pi i n x/L} J_n,$$
$$T_-(x) = \frac{2\pi}{L^2} \sum_{n=-\infty}^{\infty} e^{-2\pi i n x/L} \left( \bar{L}_n - \frac{c}{24} \delta_{n,0} \right), \qquad J_-(x) = \frac{1}{L} \sum_{n=-\infty}^{\infty} e^{-2\pi i n x/L} \bar{J}_n,$$
(26)

the operators $L_n$ and $J_n$ for $n \in \mathbb{Z}$ satisfy the commutation relations

$$\big[ L_n, L_m \big] = (n-m) L_{n+m} + \frac{c}{12} (n^3 - n) \delta_{n+m,0},$$
$$\big[ J_n, J_m \big] = K n \delta_{n+m,0}, \qquad \big[ L_n, J_m \big] = -m J_{n+m},$$
(27)

and commute with all $\bar{L}_n$ and $\bar{J}_n$, which in turn satisfy relations analogous to (27). We refer to [70] for an introduction to these and related topics.

### 3.1 Marginal deformations and the moduli space

Changes in the Luttinger parameter $K$ or equivalently the compactification radius $R = 2\sqrt{K}$ correspond to marginal deformations of the TLL or free boson CFT.[4] One way to arrive at this interpretation from a Lagrangian point of view is by identifying the marginal operator $\Phi$ responsible for changes in $R$, which by definition is a primary field with conformal weights $(h, \bar{h}) = (1, 1)$, see, e.g., [2]. The aim below is to identify this operator and explain how this gives a geometric interpretation to our space of marginal deformations.

---

[4]Recall that we set $\alpha' = 2$.

An infinitesimal change from $R$ to $R + \delta R$ implies the following change in the action (2):

$$\delta S = S_{R+\delta R} - S_R = \frac{R \delta R}{4\pi} \int \mathrm{d}^2 x \, (\partial^\mu \varphi)(\partial_\mu \varphi) . \tag{28}$$

Therefore, the marginal operator is

$$\Phi = \frac{R \delta R}{4\pi} (\partial^\mu \varphi)(\partial_\mu \varphi) = \frac{1}{4\pi} \frac{\delta R}{R} J(z) \bar{J}(\bar{z}) , \tag{29}$$

where we identified the currents $J(z) = -2\pi J_+(x^-)/\sqrt{K}$ and $\bar{J}(\bar{z}) = -2\pi J_-(x^+)/\sqrt{K}$ in complex coordinates $z = x + iv\tau = x^-$ and $\bar{z} = x - iv\tau = x^+$ with $\tau = it$ denoting imaginary time [cf. (25)]. The change is thus exactly in the form of a $J\bar{J}$ deformation.

The geometry of the 'theory space' generated by marginal deformations, known as the moduli space or conformal manifold, here denoted by $\mathcal{M}$, is given by the Zamolodchikov metric [39].[5] This is obtained from the ground-state correlation function of a pair of marginal operators on the sphere (or the infinite plane):

$$\langle \Phi(z, \bar{z}) \Phi(0, 0) \rangle = \frac{\mathrm{ds}_{\mathcal{M}}^2}{|z|^4} , \qquad \mathrm{ds}_{\mathcal{M}}^2 = \frac{1}{(4\pi)^2} \left( \frac{\mathrm{d}R}{R} \right)^2 . \tag{30}$$

Thus, up to an overall constant, the geodesic distance between two CFTs of compactification radii $R_1$ and $R_2$ in $\mathcal{M}$ is

$$\int_{R_2}^{R_1} \frac{\mathrm{d}R}{R} = \log \left( \frac{R_1}{R_2} \right) , \tag{31}$$

which is exactly $v$ in (8), giving it the geometric interpretation as the Zamolodchikov distance. It will turn out that the dynamics of our non-equilibrium protocols will crucially depend on this parameter.

## 3.2 Quantization using bosonic operators

Given our non-equilibrium protocols featuring $H_1 = H_{v_1, K_1}$ and $H_2 = H_{v_2, K_2}$, see Fig. 1, we find it convenient to let $H_1$ be our 'undeformed' theory, i.e., we set $K = K_1$ in (24), (25), and (27), and view $H_2 = H_{v_2, K_2}$ as our 'deformed' theory. To this end, we introduce two commuting sets of bosonic operators $a_n$ and $\bar{a}_n$, $n \in \mathbb{Z}$, for right- and left-moving excitations, respectively, satisfying $a_n^\dagger = a_{-n}$,

$$[a_n, a_m] = n \delta_{n+m, 0} = [\bar{a}_n, \bar{a}_m], \qquad [a_n, \bar{a}_m] = 0 , \tag{32}$$

and

$$a_n | \Omega \rangle = \bar{a}_n | \Omega \rangle = 0 \quad \forall \, n \geq 0 , \tag{33}$$

which also defines the vacuum $| \Omega \rangle$. The operators in the theory with $K = K_1$ can then be constructed as

$$J_n = \sqrt{K_1} a_n , \qquad \bar{J}_n = \sqrt{K_1} \bar{a}_n \tag{34}$$

and

$$L_n = \frac{1}{2} \sum_{m=-\infty}^{\infty} :a_{n-m} a_m: , \qquad \bar{L}_n = \frac{1}{2} \sum_{m=-\infty}^{\infty} :\bar{a}_{n-m} \bar{a}_m: , \tag{35}$$

where the Wick ordering $:\cdots:$ is with respect to $| \Omega \rangle$ (discussed further in Sec. 3.5). We recall that the latter identities are examples of the Sugawara construction, see, e.g., [70]. These

---

[5]The moduli space of the free boson CFT is parametrized by the radius $R \in [\sqrt{\alpha'}, \infty]$, obtained by quotienting $[0, \infty]$ by the action of T-duality $R \leftrightarrow \alpha'/R$.

operators can be shown to satisfy (27) with $K = K_1$ and $c = 1$. The Fourier modes of the bosonic fields $\varphi(x)$ and $\Pi(x)$ in (1) can then be constructed as

$$\varphi_n = \frac{1}{2\sqrt{K_1}} \frac{i}{n}\left(a_n - \bar{a}_{-n}\right), \qquad \Pi_n = \sqrt{K_1}\left(a_{-n} + \bar{a}_n\right) \tag{36}$$

for all $n \neq 0$, satisfying $[\varphi_n, \Pi_m] = i\delta_{n,m}$ and $[\varphi_n, \varphi_m] = 0 = [\Pi_n, \Pi_m]$ for $n, m \neq 0$. As usual, the case $n = 0$ has to be be handled separately. To fix our terminology, we will refer to $a_n$ and $\bar{a}_n$ for $n \neq 0$ as oscillator modes and $a_0$ and $\bar{a}_0$ as zero modes.

Given the above, we can express the undeformed Hamiltonian

$$H_1 = \frac{2\pi v_1}{L}\left(L_0 + \bar{L}_0\right) - \frac{\pi v_1}{6L} \tag{37}$$

in terms of the oscillator and zero modes: $H_1 = H_1^{(0)} + H_1^{(\text{osc})} - \pi v_1/6L$ with

$$\begin{aligned}
H_1^{(\text{osc})} &= \frac{\pi v_1}{L} \sum_{n \neq 0} :\!(a_{-n}a_n + \bar{a}_{-n}\bar{a}_n)\!: \,, \\
H_1^{(0)} &= \frac{\pi v_1}{L}\left(a_0^2 + \bar{a}_0^2\right).
\end{aligned} \tag{38}$$

It follows that $H_1$ does not couple right and left movers and has $|\Omega\rangle$ in (33) as its ground state. Let us also write the deformed Hamiltonian $H_2$ using the modes of the undeformed theory:[6] $H_2 = H_2^{(0)} + H_2^{(\text{osc})} - \pi v_2/6L$ with

$$\begin{aligned}
H_2^{(\text{osc})} &= \frac{\pi v_2}{L} \sum_{n \neq 0}\left[\cosh(2\nu) :\!\left(a_{-n}a_n + \bar{a}_{-n}\bar{a}_n\right)\!: + 2\sinh(2\nu)a_n\bar{a}_n\right] + E_2^0 \,, \\
H_2^{(0)} &= \frac{\pi v_2}{L}\left[\cosh(2\nu)\left(a_0^2 + \bar{a}_0^2\right) + 2\sinh(2\nu)a_0\bar{a}_0\right],
\end{aligned} \tag{39}$$

for the Zamolodchikov distance $\nu$ in (8) as a function of the two Luttinger parameters $K_{1,2}$ (or the two radii $R_{1,2}$), where $E_2^0 = -(2\pi v_2/L)\sum_{n>0}[\cosh(2\nu) - 1]n$ is a diverging constant due to Wick ordering with respect to $|\Omega\rangle$ (see Sec. 3.5). We note the presence of terms that couple right and left movers if $\nu \neq 0$, which makes it manifest that $H_2$ is a $J\bar{J}$ deformation of $H_1$ [cf. (34)]. For completeness and future reference, we note the following analogue of (37) for the deformed Hamiltonian:

$$H_2 = \frac{2\pi v_2}{L}\cosh(2\nu)(L_0 + \bar{L}_0) + \frac{2\pi v_2}{L}\frac{\sinh(2\nu)}{K_1}\sum_{n=-\infty}^{\infty} J_n\bar{J}_n - \frac{\pi v_2}{6L} + E_2^0 \,, \tag{40}$$

where all ingredients are operators of the undeformed theory with $K = K_1$.[7]

## 3.3 Underlying $\mathfrak{su}(1,1)$ algebras

The combinations of oscillator modes appearing in $H_1$ and $H_2$ in (38) and (39) can conveniently be written in terms of the generators of a countably infinite number of copies of the $\mathfrak{su}(1,1)$ algebra, labeled by $n \in \mathbb{Z}^+ = \{1, 2, \ldots\}$. More precisely, for $n > 0$, let

$$K_0^{(n)} = \frac{1}{2n}\left(a_{-n}a_n + \bar{a}_{-n}\bar{a}_n + n\right), \qquad K_-^{(n)} = \frac{1}{n}a_n\bar{a}_n, \qquad K_+^{(n)} = \frac{1}{n}a_{-n}\bar{a}_{-n}, \tag{41}$$

---

[6]The expression in (39) can also be derived from (50). We note that the coefficients $\cosh(2\nu)$ and $\sinh(2\nu)$ can be interpreted as coupling constants and correspond to $1 + g_4/2$ and $g_2/2$ in (14), respectively, if the latter are defined with respect to the theory with $K = K_1$ (instead of $K = 1$ as usual).

[7]Under an infinitesimal change of the Luttinger parameter, $K_2 = K_1 + \delta K_1$, the deformed Hamiltonian (40) is related to the undeformed one as $H_2/v_2 \approx H_1/v_1 - (2\pi/L)(\delta K_1/K_1^2)\sum_n J_n\bar{J}_n$.

which satisfy $\left(K_-^{(n)}\right)^\dagger = K_+^{(n)}$ and

$$[K_-^{(n)}, K_+^{(m)}] = 2K_0^{(n)}\delta_{n,m}, \qquad [K_0^{(n)}, K_\pm^{(m)}] = \pm K_\pm^{(n)}\delta_{n,m}, \tag{42}$$

see, e.g., [73]. For later reference, one can show that the associated Cartan-Killing form is

$$\mathcal{K}(X,Y) = \begin{pmatrix} x_0 & x_- & x_+ \end{pmatrix} \begin{pmatrix} 2 & 0 & 0 \\ 0 & 0 & -4 \\ 0 & -4 & 0 \end{pmatrix} \begin{pmatrix} y_0 \\ y_- \\ y_+ \end{pmatrix} \tag{43}$$

for $X = x_0 K_0^{(n)} + x_- K_-^{(n)} + x_+ K_+^{(n)}$ and $Y = y_0 K_0^{(n)} + y_- K_-^{(n)} + y_+ K_+^{(n)}$. We recall that the corresponding group, $\mathrm{SU}(1,1)$, is non-compact and, therefore, all unitary irreducible representations are infinite dimensional, see, e.g., [73]. However, one can construct a non-unitary $2 \times 2$-matrix representation of the generators:

$$K_0^{(n)}\Big|_{2\times 2} = \begin{pmatrix} -1/2 & 0 \\ 0 & 1/2 \end{pmatrix}, \qquad K_-^{(n)}\Big|_{2\times 2} = \begin{pmatrix} 0 & 1 \\ 0 & 0 \end{pmatrix}, \qquad K_+^{(n)}\Big|_{2\times 2} = \begin{pmatrix} 0 & 0 \\ -1 & 0 \end{pmatrix}. \tag{44}$$

Additionally, it is also useful to note the following commutation relations:

$$[K_-^{(n)}, a_m] = \delta_{n+m,0}\bar{a}_{-m}, \qquad\qquad [K_-^{(n)}, \bar{a}_m] = \delta_{n+m,0}a_{-m}, \tag{45a}$$

$$[K_+^{(n)}, a_m] = -\delta_{n+m,0}\bar{a}_{-m}, \qquad\qquad [K_+^{(n)}, \bar{a}_m] = -\delta_{n+m,0}a_{-m}. \tag{45b}$$

The Hamiltonians can also be written in terms of the $\mathfrak{su}(1,1)$ generators as

$$H_1 = H_1^{(0)} + \frac{2\pi v_1}{L}\sum_{n>0} 2n\left(K_0^{(n)} - \frac{1}{2}\right) - \frac{\pi v_1}{6L} \tag{46}$$

and

$$H_2 = H_2^{(0)} + \frac{2\pi v_2}{L}\sum_{n>0} 2n\left[\cosh(2v)\left(K_0^{(n)} - \frac{1}{2}\right) + \sinh(2v)\left(K_-^{(n)} + K_+^{(n)}\right)\right] - \frac{\pi v_2}{6L}, \tag{47}$$

with $v$ in (8).

### 3.4 Bogoliubov transformations

It is well-known that a TLL Hamiltonian can be 'diagonalized' by a Bogoliubov transformation, which effectively 'rotates' the oscillator modes by an 'angle' $v$, for a suitable choice of the latter. Below, we discuss the operator that implements this transformation and show that the relevant choice of $v$ is exactly the one in (8).

As explained in [10], the Bogoliubov transformation is implemented by the unitary operator[8]

$$\mathcal{I}_v = \exp\left[v\sum_{n\neq 0}\frac{1}{n}a_n\bar{a}_n\right] = \prod_{n>0}\exp\left[v\left(K_-^{(n)} - K_+^{(n)}\right)\right], \tag{48}$$

defined for any $v \in \mathbb{R}$. The second equality rewrites the operator as it appears in [10] in terms of the $\mathfrak{su}(1,1)$ generators, which will prove convenient later. Indeed, using (45), it is straightforward to show

$$\mathcal{I}_v a_n \mathcal{I}_v^\dagger = a_n\cosh(v) + \bar{a}_{-n}\sinh(v), \tag{49a}$$

$$\mathcal{I}_v \bar{a}_n \mathcal{I}_v^\dagger = \bar{a}_n\cosh(v) + a_{-n}\sinh(v) \tag{49b}$$

---

[8]Note that we use $a_n$ and $\bar{a}_n$ of the undeformed theory with $K = K_1$ to define $\mathcal{I}_v$ rather than the usual choice corresponding to $K = 1$.

for $n \neq 0$. The inverse relations are obtained by noting that $\mathcal{I}_\nu^\dagger = \mathcal{I}_\nu^{-1} = \mathcal{I}_{-\nu}$. Note that one must take the latter as a definition of $\mathcal{I}_\nu^\dagger$ when using (48) with the non-unitary representation in (44). By picking $\nu$ as in (8), one can show that

$$\mathcal{I}_\nu^\dagger H_2^{(\mathrm{osc})} \mathcal{I}_\nu = \frac{v_2}{v_1} H_1^{(\mathrm{osc})} + E_2^0, \tag{50}$$

up to the diverging constant $E_2^0$ due to Wick ordering with respect to $|\Omega\rangle$ (see Sec. 3.5). This allows us to write the Floquet operator in (6) as

$$U_F = e^{-iE_2^0 t_2} e^{-iH_1 t_1} \mathcal{I}_\nu e^{-iH_1 \tilde{t}_2} \mathcal{I}_\nu^\dagger e^{-i\left[H_2^{(0)} t_2 - H_1^{(0)} \tilde{t}_2\right]}, \qquad \tilde{t}_2 = (v_2/v_1) t_2, \tag{51}$$

where the overall phase $e^{-iE_2^0 t_2}$ will be of no consequence to the dynamical observables we study. The above expression for $U_F$ is the key to most of our subsequent computations. (The quantum quench can be studied as a special case by setting $t_1 = 0$ and $t_2 = t$.)

In Sec. 1, we noted that $\mathcal{I}_\nu$ brings to mind interface operators in boundary CFT since it connects two different bosonic theories along time interfaces in our non-equilibrium protocols. One can also observe that this operator, as defined in (48), has the form of a two-mode squeeze operator [74]. This class of operators play an important role in quantum optics, where they are associated to degenerate parametric amplification. In our present setup, the two modes correspond to the right- and left-moving sets of oscillator modes.

We also find it useful to introduce the $q$-modified operator

$$\mathcal{I}_\nu^{(q)} = q^{L_0 + \bar{L}_0} \mathcal{I}_\nu q^{-L_0 - \bar{L}_0} = \exp\left[\nu \sum_{n \neq 0} \frac{q^{-2n}}{n} a_n \bar{a}_n\right] = \prod_{n>0} \exp\left[\nu\left(q^{-2n} K_-^{(n)} - q^{2n} K_+^{(n)}\right)\right] \tag{52}$$

for $q \in \mathrm{U}(1)$. In the second equality, we used that

$$q^{L_0 + \bar{L}_0} a_n q^{-L_0 - \bar{L}_0} = a_n q^{-n}, \qquad q^{L_0 + \bar{L}_0} \bar{a}_n q^{-L_0 - \bar{L}_0} = \bar{a}_n q^{-n}. \tag{53}$$

(The latter is nothing but the inverse time evolution of $a_n$ and $\bar{a}_n$ under $H_1$ in (38) if one sets $q = e^{-2\pi i v_1 t/L}$.) The $q$-modified operators transform the oscillator modes as

$$\mathcal{I}_\nu^{(q)} a_n \left(\mathcal{I}_\nu^{(q)}\right)^\dagger = a_n \cosh(\nu) + \bar{a}_{-n} \sinh(\nu) q^{2n}, \tag{54a}$$

$$\mathcal{I}_\nu^{(q)} \bar{a}_n \left(\mathcal{I}_\nu^{(q)}\right)^\dagger = \bar{a}_n \cosh(\nu) + a_{-n} \sinh(\nu) q^{2n}, \tag{54b}$$

generalizing (49). Moreover, it also allows us to further rewrite (51) as

$$U_F = e^{-iE_2^0 t_2} q_1^{L_0 + \bar{L}_0} \mathcal{I}_\nu \left(\mathcal{I}_\nu^{(q_2)}\right)^\dagger q_2^{L_0 + \bar{L}_0} e^{-i\left[H_2^{(0)} - (v_2/v_1)H_1^{(0)}\right] t_2}, \tag{55}$$

for $q_{1,2}$ in (4), where we reiterate that the phase $e^{-iE_2^0 t_2}$ will be of no consequence in practice. Lastly, as for $\mathcal{I}_\nu = \mathcal{I}_\nu^{(1)}$, unitarity implies $\left(\mathcal{I}_\nu^{(q)}\right)^\dagger = \left(\mathcal{I}_\nu^{(q)}\right)^{-1} = \mathcal{I}_{-\nu}^{(q)}$, which must be taken as a definition when using (52) with the non-unitary representation in (44).

## 3.5 Wick ordering

The Wick ordering $:\cdots:$ we use is with respect to $|\Omega\rangle$ in (33) and is therefore the ordering of the Hilbert space of $H_1$ constructed from its primary states and their descendants. For bilinears of the form $a_n a_m$ and $\bar{a}_n \bar{a}_m$, this ordering is equivalent to subtracting the ground-state expectation value,

$$:a_n a_m: = a_n a_m - \langle\Omega|a_n a_m|\Omega\rangle = a_n a_m - \delta_{n+m,0} n\theta(n), \tag{56}$$

where $\theta(\cdot)$ is the Heaviside function, and similarly for $\bar{a}_n \bar{a}_m$.[9]

The constant $E_2^0$ in (39) and (50) appears due to re-ordering of the right-hand side, and diverges due to that the $J\bar{J}$ deformation affects all modes. A more rigorous approach would be to include an ultraviolet cutoff on the deformation, effectively a momentum dependence in the Luttinger parameter $K_2^{(n)}$ so that it tends to $K_1$ sufficiently fast for large $|n|$. As mentioned, this is related to making $\mathcal{I}_\nu$ well defined: This operator provides a map between the Hilbert spaces of our two theories with different Luttinger parameters, which strictly speaking become unitarily inequivalent in the absence of a cutoff, manifested by that the 'true ground state' of $H_2$ is separated from its 'ground state' $\mathcal{I}_\nu|\Omega\rangle$ in the Hilbert space of $H_1$ by a diverging constant. This necessitates an additive renormalization of $\mathcal{I}_\nu^\dagger H_2 \mathcal{I}_\nu$ for it to make sense on the Hilbert space of $H_1$, see, e.g., [59] for further discussion. We remark, however, that the presence of a cutoff can be motivated by physical applications and that all steps in this paper can be repeated with it in place since our quenched or driven theory corresponds to an infinite sequence of uncoupled (discrete-time) quantum (parametric) oscillators.

## 4 Quantum quench

In this section, we study the dynamics of a TLL after an interaction quench, starting from an arbitrary eigenstate of $H_1$, and switching the Luttinger parameter from $K_1$ to $K_2$ at time $t = 0$, see Fig. 1(a). As discussed in Sec. 3, this corresponds to quenching the original TLL Hamiltonian with a marginal ($J\bar{J}$) deformation. We compute the exact time-evolution after the quench of the following two quantities:

1. The Loschmidt echo, defined for a pure initial state $|\Psi\rangle$ as

$$L(t) = |\langle\Psi|e^{-iH_2 t}|\Psi\rangle|^2. \tag{57}$$

   This quantifies the time-dependent return probability of a state and can thereby be used to measure the probability of quantum revivals. Moreover, non-analyticities in $\log[L(t)]$ after a quantum quench can reveal rich dynamics and are a typical signature of dynamical quantum phase transitions [49].

2. The energy-density expectation, defined for a pure initial state $|\Psi\rangle$ as

$$\mathcal{E}_\Psi(x, t) = \langle\Psi|e^{iH_2 t} v_1 [T_+(x) + T_-(x)] e^{-iH_2 t}|\Psi\rangle. \tag{58}$$

   In addition to pure states, we also compute the time evolution of the energy-density expectation with respect to initial thermal states. For all the cases considered, the spatial-homogeneity of the initial state significantly simplifies the computations.

We note that, since the initial states we consider are spatially homogeneous, time evolution of the particle density would be trivial, which is why we do not study this observable in the present work. However, for spatially inhomogeneous initial states, the expectation value of particle density would generically have a non-trivial time evolution.

### 4.1 Loschmidt echo

**For the ground state**

We first compute the Loschmidt echo after the quench from the ground state $|\Omega\rangle$ of $H_1$,

$$L_\Omega(t) = \left|\langle\Omega|e^{-iH_2 t}|\Omega\rangle\right|^2. \tag{59}$$

---

[9]Starting from the usual definition of placing all creation operators to the left of all annihilation operators, (56) can be verified by identifying $a_n$ and $\bar{a}_n$ for $n < (>) 0$ as creation (annihilation) operators, meaning that the only non-trivial case is $n > 0 > m$, and using (32) and (33).

Using the framework introduced in Sec. 3, it can be shown that

$$\langle\Omega|e^{-iH_2 t}|\Omega\rangle = e^{-iE_2^0 t}\langle\Omega|\mathcal{I}_\nu\big(\mathcal{I}_\nu^{(q)}\big)^\dagger|\Omega\rangle. \tag{60}$$

Indeed, since (6) implies $U_F = e^{-iH_2 t}$ for $t_1 = 0$ and $t_2 = t$, the above follows from (55) for $q_1 = 1$ and $q_2 = q = e^{-2\pi i v_2 t/L}$ and (33). Note that $L_\Omega(t)$ is insensitive to the overall phase $e^{-iE_2^0 t}$. Our strategy to compute the right-hand side of (60) is to use the decomposition of $\mathcal{I}_\nu\big(\mathcal{I}_\nu^{(q)}\big)^\dagger$ in terms of the $\mathfrak{su}(1,1)$ generators in (41):

$$\begin{aligned}
\langle\Omega|\mathcal{I}_\nu\big(\mathcal{I}_\nu^{(q)}\big)^\dagger|\Omega\rangle &= \prod_{n>0}\langle\Omega|\exp\big(\zeta_+^{(n)}K_+^{(n)}\big)\exp\big(\zeta_0^{(n)}K_0^{(n)}\big)\exp\big(\zeta_-^{(n)}K_-^{(n)}\big)|\Omega\rangle \\
&= \prod_{n>0}\langle\Omega|\exp\big(\zeta_0^{(n)}K_0^{(n)}\big)|\Omega\rangle = \prod_{n>0}\exp\big(\zeta_0^{(n)}/2\big),
\end{aligned} \tag{61}$$

where we used $\langle\Omega|K_+^{(n)} = 0 = K_-^{(n)}|\Omega\rangle$, which follows from (33). One efficient way to find the coefficients $\zeta_0^{(n)}$ and $\zeta_\pm^{(n)}$ is to use the non-unitary $2\times 2$-matrix representation of the $\mathfrak{su}(1,1)$ in (44). In this representation, using (48) and (52) with $\big(\mathcal{I}_\nu^{(q)}\big)^\dagger = \mathcal{I}_{-\nu}^{(q)}$ as a definition,[10] we obtain

$$\mathcal{I}_\nu\big(\mathcal{I}_\nu^{(q)}\big)^\dagger\Big|_{2\times 2}^{(n)} = \begin{pmatrix} \cosh^2(\nu) - \sinh^2(\nu)q^{2n} & \frac{1}{2}\sinh(2\nu)\big(1 - q^{-2n}\big) \\ \frac{1}{2}\sinh(2\nu)\big(1 - q^{2n}\big) & \cosh^2(\nu) - \sinh^2(\nu)q^{-2n} \end{pmatrix} \tag{62}$$

for the $n$th mode. Comparing this with the product

$$e^{\zeta_+^{(n)}K_+^{(n)}}e^{\zeta_0^{(n)}K_0^{(n)}}e^{\zeta_-^{(n)}K_-^{(n)}}\Big|_{2\times 2} = \begin{pmatrix} e^{-\zeta_0^{(n)}/2} & \zeta_-^{(n)}e^{-\zeta_0^{(n)}/2} \\ -\zeta_+^{(n)}e^{-\zeta_0^{(n)}/2} & e^{\zeta_0^{(n)}/2} - \zeta_-^{(n)}\zeta_+^{(n)}e^{-\zeta_0^{(n)}/2} \end{pmatrix}, \tag{63}$$

we deduce that the Loschmidt echo after the quench starting from the ground state is

$$L_\Omega(t) = \prod_{n>0}\frac{1}{\big|\cosh^2(\nu) - \sinh^2(\nu)q^{2n}\big|^2}. \tag{64}$$

Considering all the modes by taking the infinite product in (64) into account, the resulting Loschmidt echo has a Dirac comb structure, i.e., it is zero at all times $t$ apart from $t = kL/2v_2$ for $k \in \mathbb{N} = \{0, 1, 2, \dots\}$, at which there are exact quantum revivals. These can be understood from a quasiparticle picture [75]: Right- and left-moving quasiparticles emitted from any position meet again after half-integer multiples of $L$ with periodic boundary conditions. A similar result was found in [76] for the Loschmidt echo by starting from a boundary state and quenching with a uniform CFT Hamiltonian, while we started from the ground state of a uniform compactified free boson CFT, and quenched with a $J\bar{J}$ deformed CFT. To compare with critical lattice systems, see Sec. 2, it is necessary to apply a cutoff on the number of momentum modes that appear in the infinite product. This in turn leads to a cutoff dependent Loschmidt echo, as shown in Fig. 2. Finally, we note that the Loschmidt echo starting from a primary state $|h, \bar{h}\rangle$ of conformal dimension $(h, \bar{h})$ is the same as starting from the ground state $|\Omega\rangle$. We present a proof for this statement in Appendix B.1.

**For excited states**

We now compute the exact time evolution of the Loschmidt echo starting from an initial state of the form

$$|\Psi_{\boldsymbol{p}, \bar{\boldsymbol{p}}}\rangle = \frac{1}{\sqrt{\mathcal{N}_{\boldsymbol{p}, \bar{\boldsymbol{p}}}}}\prod_{n=1}^{\infty}\bar{a}_{-n}^{\bar{p}_n}a_{-n}^{p_n}|\Omega\rangle, \quad \mathcal{N}_{\boldsymbol{p}, \bar{\boldsymbol{p}}} = \prod_{n=1}^{\infty}(n^{p_n}p_n!)(n^{\bar{p}_n}\bar{p}_n!) \tag{65}$$

---

[10]At a practical level, this is done in order to bypass the unitarity requirement on the SU(1, 1) representation.

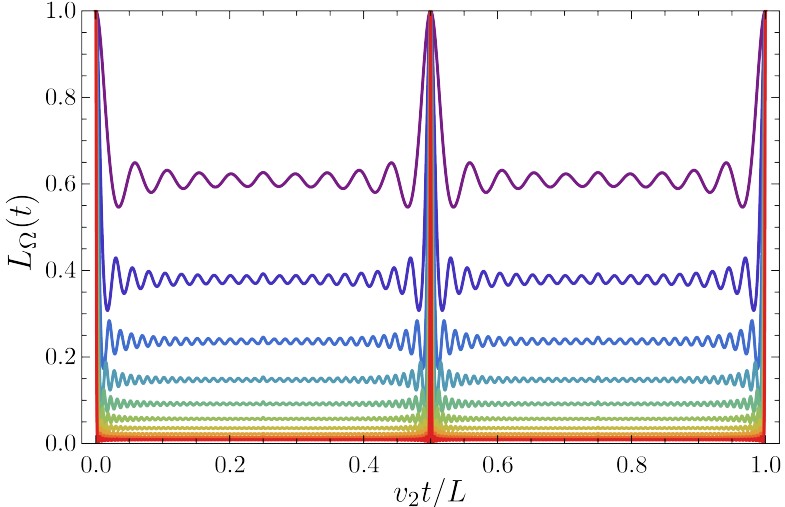

Figure 2: Time evolution of the Loschmidt echo $L_\Omega(t)$ in (64) following a quench with $K_1/K_2 = 7/6$ starting from the ground state. The results are plotted for a cutoff on the number of modes at $n = 10, 20, \ldots, 100$ (top to bottom). We observe that by increasing the number of terms in the product, $L_\Omega(t)$ tends to the exact CFT result of a Dirac comb with revivals at $kL/2v_2$, $k \in \mathbb{N}$.

for $\boldsymbol{p} = (p_n)_{n=1}^\infty$ and $\bar{\boldsymbol{p}} = (\bar{p}_n)_{n=1}^\infty$ with $p_n, \bar{p}_n \in \mathbb{N}$, i.e., any possible descendant state from the ground state $|\Omega\rangle$. Following the above reasoning, the Loschmidt echo has the form

$$L_{\boldsymbol{p},\bar{\boldsymbol{p}}}(t) = \left|\langle \Psi_{\boldsymbol{p},\bar{\boldsymbol{p}}}|\mathrm{e}^{-\mathrm{i}H_2 t}|\Psi_{\boldsymbol{p},\bar{\boldsymbol{p}}}\rangle\right|^2 = \frac{1}{\mathcal{N}_{\boldsymbol{p},\bar{\boldsymbol{p}}}^2}\left|C_{\boldsymbol{p},\bar{\boldsymbol{p}}}\right|^2, \tag{66}$$

where

$$C_{\boldsymbol{p},\bar{\boldsymbol{p}}} = \langle\Omega|\left(\prod_{n=1}^\infty a_n^{p_n}\bar{a}_n^{\bar{p}_n}\right)\mathcal{I}_\nu\left(\mathcal{I}_\nu^{(q)}\right)^\dagger\left(\prod_{n=1}^\infty \bar{a}_{-n}^{\bar{p}_n}a_{-n}^{p_n}\right)|\Omega\rangle \tag{67}$$

is the non-trivial part we need to compute.

Let us start by considering the initial state $\left(1/\sqrt{n^p p!}\right)a_{-n}^p|\Omega\rangle$, writing $p = p_n$ to lighten the notation. We thus need to compute

$$C_p = \langle\Omega|a_n^p\mathcal{I}_\nu\left(\mathcal{I}_\nu^{(q)}\right)^\dagger a_{-n}^p|\Omega\rangle. \tag{68}$$

This can be achieved by using (49) and (54) to move one $a_{-n}$ past $\mathcal{I}_\nu\left(\mathcal{I}_\nu^{(q)}\right)^\dagger$, which yields

$$C_p = \langle\Omega|a_n^p\left(A_n a_{-n} + B_n \bar{a}_n\right)\mathcal{I}_\nu\left(\mathcal{I}_\nu^{(q)}\right)^\dagger a_{-n}^{p-1}|\Omega\rangle, \tag{69}$$

with $A_n = A_n(t)$ and $B_n = B_n(t)$ given by

$$A_n(t) = \cosh^2(\nu) - \sinh^2(\nu)q^{-2n}, \qquad B_n(t) = \frac{1}{2}\sinh(2\nu)\left(1 - q^{-2n}\right), \tag{70}$$

using $q = \mathrm{e}^{-2\pi\mathrm{i}v_2 t/L}$. Noting that $\langle\Omega|a_n^p a_{-n} = pn\langle\Omega|a_n^{p-1}$, we obtain

$$C_p = npA_n C_{p-1} + B_n\langle\Omega|a_n^p\bar{a}_n\mathcal{I}_\nu\left(\mathcal{I}_\nu^{(q)}\right)^\dagger a_{-n}^{p-1}|\Omega\rangle. \tag{71}$$

The second term can be simplified by moving $\bar{a}_n$ to the right, leading to

$$\langle\Omega|a_n^p\bar{a}_n\mathcal{I}_\nu\left(\mathcal{I}_\nu^{(q)}\right)^\dagger a_{-n}^{p-1}|\Omega\rangle = A_n\langle\Omega|a_n^p\mathcal{I}_\nu\left(\mathcal{I}_\nu^{(q)}\right)^\dagger\bar{a}_n a_{-n}^{p-1}|\Omega\rangle - \overline{B_n}C_p, \tag{72}$$

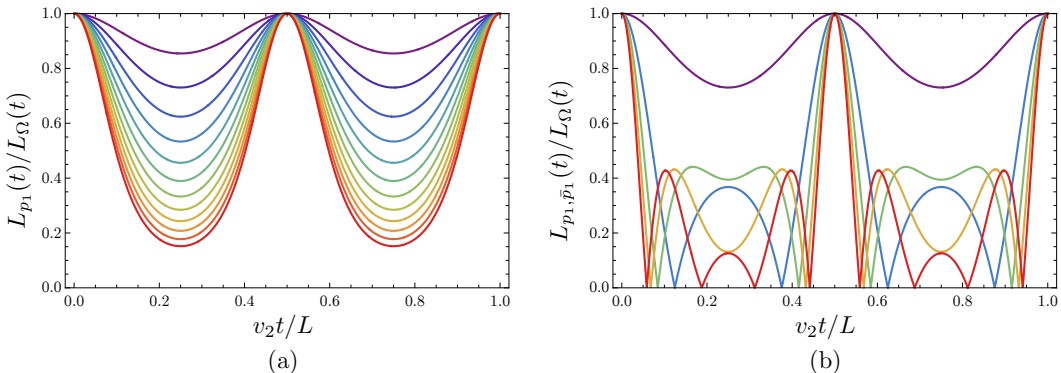

Figure 3: (a) Time evolution of the Loschmidt echo $L_{p_1}(t)/L_\Omega(t)$ in (75) following a quench with $K_1/K_2 = 4/3$ for initial states of the form $a_{-1}^{p_1}|\Omega\rangle$ for $p_1 = 1, 2, \ldots, 12$ (top to bottom). (b) Time evolution of the Loschmidt echo $L_{p_1, \bar{p}_1}(t)/L_\Omega(t)$ in (83) following the same quench for initial states of the form $a_{-1}^{p_1} \bar{a}_{-1}^{\bar{p}_1}|\Omega\rangle$ for $p_1 = 0, 1, 2, 3, 4$ (top to bottom) and $\bar{p}_1 = 2$. Temporal orthogonality and non-analyticities of the Loschmidt echo at discrete times can only be observed if the initial state mixes right- and left-moving excitations for a given mode $n$. On the other hand, quantum revivals at integer multiples of $L/2$ happen for any choice of pure initial state.

with the complex conjugated $\overline{B_n} = B_n(-t)$ given by (70). The first term vanishes and we conclude that $C_p$ must satisfy the recursion relation

$$C_p = np \frac{A_n}{1 + |B_n|^2} C_{p-1}, \qquad C_0 = \langle\Omega|\mathcal{I}_\nu \left(\mathcal{I}_\nu^{(q)}\right)^\dagger |\Omega\rangle. \tag{73}$$

Solving this recursion relation, we conclude that

$$C_p = n^p p! \left(\frac{A_n}{1 + |B_n|^2}\right)^p \langle\Omega|\mathcal{I}_\nu \left(\mathcal{I}_\nu^{(q)}\right)^\dagger |\Omega\rangle. \tag{74}$$

Thus, using (66) with $\mathcal{N}_{p_n} = n^{p_n}(p_n!)$, the Loschmidt echo starting from an initial state of the form $\left(1/\sqrt{n^{p_n}p_n!}\right)a_{-n}^{P_n}|\Omega\rangle$ is obtained by multiplying $L_\Omega(t)$ in (64) by a time-dependent factor. The result is

$$L_{p_n}(t) = L_\Omega(t) \left(\frac{|A_n(t)|}{1 + |B_n(t)|^2}\right)^{2p_n}, \tag{75}$$

with $A_n(t)$ and $B_n(t)$ in (70). A direct consequence of (75) is that $L_{p_n}(t)$ decreases exponentially with $p_n$ by starting from such an excited state instead of the ground state. However, the quantum revivals at times $t = kL/2v_2$ remain unchanged, see Fig. 3(a).

We now consider an initial state that mixes right- and left-moving excitations for a given mode $n$,

$$|\Psi_{p,\bar{p}}\rangle = \frac{1}{\sqrt{(n^p p!)(n^{\bar{p}} \bar{p}!)}} \bar{a}_{-n}^{\bar{p}} a_{-n}^p |\Omega\rangle, \tag{76}$$

again writing $p = p_n$ and $\bar{p} = \bar{p}_n$ to lighten the notation. The Hamiltonian $H_2$ after the quench acts non-trivially on such an initial state because the marginal $(J\bar{J})$ deformation effectively repartitions excitations into right and left moving. As before, we compute

$$C_{p,\bar{p}} = \langle\Omega|a_n^p \bar{a}_n^{\bar{p}} \mathcal{I}_\nu \left(\mathcal{I}_\nu^{(q)}\right)^\dagger \bar{a}_{-n}^{\bar{p}} a_{-n}^p |\Omega\rangle. \tag{77}$$

Once again, using (49) and (54) to move one $\bar{a}_{-n}$ from the right to the left, we obtain

$$C_{p,\bar{p}} = n\bar{p} A_n C_{p,\bar{p}-1} + B_n \langle\Omega|a_n^{p+1} \bar{a}_n^{\bar{p}} \mathcal{I}_\nu \left(\mathcal{I}_\nu^{(q)}\right)^\dagger \bar{a}_{-n}^{\bar{p}-1} a_{-n}^p |\Omega\rangle, \tag{78}$$

with $A_n = A_n(t)$ and $B_n = B_n(t)$ given by (70). The second term can be simplified by successively moving $\bar{a}_n$ from the left to the right, eventually leading to

$$B_n \langle \Omega | a_n^{p+1} \bar{a}_n^{\bar{p}} \mathcal{I}_\nu \left( \mathcal{I}_\nu^{(q)} \right)^\dagger \bar{a}_{-n}^{\bar{p}-1} a_{-n}^p | \Omega \rangle = -|B_n|^2 \sum_{j=0}^{\bar{p}-1} \frac{(\bar{p}-1)!}{(\bar{p}-j-1)!} (nA_n)^j C_{p+1,\bar{p}-1-j} . \tag{79}$$

Plugging into (78), we find the following two-variable recursion relation for $C_{p,\bar{p}}$:

$$C_{p,\bar{p}} = n\bar{p} A_n C_{p,\bar{p}-1} - |B_n|^2 \sum_{j=0}^{\bar{p}-1} \frac{(\bar{p}-1)!}{(\bar{p}-j-1)!} (nA_n)^j C_{p+1,\bar{p}-1-j} , \tag{80}$$

with (initial) conditions $C_{p,0} = C_p$ and $C_{0,\bar{p}} = C_{\bar{p}}$ given by (74). The solution to (80) takes the general form

$$C_{p,\bar{p}} = (n^p p!)(n^{\bar{p}} \bar{p}!) \left( \frac{A_n}{1+|B_n|^2} \right)^{p+\bar{p}} \sum_{j=0}^{\min\{p,\bar{p}\}} \binom{p}{j} \binom{\bar{p}}{j} (-|B_n|^2)^j \langle \Omega | \mathcal{I}_\nu \left( \mathcal{I}_\nu^{(q)} \right)^\dagger | \Omega \rangle . \tag{81}$$

Alternatively, this can be stated in terms of the hypergeometric function $_2F_1(a,b;c;z)$ as

$$C_{p,\bar{p}} = (n^p p!)(n^{\bar{p}} \bar{p}!) \left( \frac{A_n}{1+|B_n|^2} \right)^{p+\bar{p}} {}_2F_1(-p,-\bar{p};1;-|B_n|^2) \langle \Omega | \mathcal{I}_\nu \left( \mathcal{I}_\nu^{(q)} \right)^\dagger | \Omega \rangle , \tag{82}$$

from which the Loschmidt echo is obtained using (66) with $\mathcal{N}_{p,\bar{p}} = (n^p p!)(n^{\bar{p}} \bar{p}!)$. In conclusion, the final result starting from initial states of the form $\left( 1/\sqrt{(n^{p_n} p_n!)(n^{\bar{p}_n} \bar{p}_n!)} \right) \bar{a}_{-n}^{\bar{p}_n} a_{-n}^{p_n} | \Omega \rangle$ is

$$L_{p_n,\bar{p}_n}(t) = L_\Omega(t) \left( \frac{|A_n(t)|}{1+|B_n(t)|^2} \right)^{2(p_n+\bar{p}_n)} \left| {}_2F_1(-p_n,-\bar{p}_n;1;-|B_n(t)|^2) \right|^2 , \tag{83}$$

with $A_n(t)$ and $B_n(t)$ in (70). Note that this is consistent with (75) if $p_n = 0$ or $\bar{p}_n = 0$ since $_2F_1(0,b;c;z) = 1 = {}_2F_1(a,0;c;z)$.

It follows from (83) that mixing right- and left-moving excitations in the initial state leads to an additional factor of $_2F_1(-p_n,-\bar{p}_n;1;-|B_n|^2)$, which is a consequence of the repartitioning of the excitations due to the $J\bar{J}$ deformation in $H_2$. Note that this hypergeometric function is a polynomial of order $\min\{p_n,\bar{p}_n\}$. Thus, if such a polynomial admits real zeros, the Loschmidt echo might in turn admit exact zeros at particular values of $t$, leading to non-analytic times in $\log[L(t)]$. In particular, we apply Theorem 2(v) in [77] to conclude that all zeros of $_2F_1(-p_n,-\bar{p}_n;1;y)$ for the variable $y$ are real and negative. On the other hand, in order for the Loschmidt echo to develop an exact zero at finite times, a given zero $y_*$ of $_2F_1(-p_n,-\bar{p}_n;1;y)$ is required to fulfill

$$y_* \in \left[ -4\cosh^2(\nu)\sinh^2(\nu), 0 \right] . \tag{84}$$

In particular, in the limit where $\nu \to \infty$, all the $\min\{p_n,\bar{p}_n\}$ zeros correspond to different values of $t$ for which the Loschmidt echo is exactly zero. As can be seen on Fig. 3(b), the Loschmidt echo is non-analytic in the vicinity of these exact zeros. Thus, we interpret our result for the Loschmidt echo as dynamical quantum phase transitions arising periodically in time. We stress that this phenomena of temporal orthogonality [78] can only be observed for our quench protocol if the initial state mixes right- and left-moving excitations for the same mode $n$, and can be seen as Lee-Yang-Fisher zeros [49] in the complex Loschmidt amplitude crossing the real time axis whenever the condition in (84) is fulfilled.

Finally, we note that the non-normalized return amplitude $C_{p,\bar{p}}$ in (67) for a general excited state of the form in (65) can be obtained as $C_{p,\bar{p}} = \prod_n C_{p_n,\bar{p}_n}$. Therefore, we conclude that the Loschmidt echo after a quantum quench starting from a general excited state is

$$L_{p,\bar{p}}(t) = L_\Omega(t) \prod_{n=1}^{\infty} \left( \frac{|A_n(t)|}{1+|B_n(t)|^2} \right)^{2(p_n+\bar{p}_n)} \left| {}_2F_1(-p_n,-\bar{p}_n;1;-|B_n(t)|^2) \right|^2, \qquad (85)$$

with $A_n(t)$ and $B_n(t)$ in (70). As discussed in Appendix B.1, the result would be unchanged by considering excited states in the form of descendant states from other primary states than the ground state. Consequently, (85) is the most general result for the Loschmidt echo after an interaction quench starting from any eigenstate of $H_1$.

## 4.2 Energy density

We now turn to the energy density of the system initialized in an arbitrary eigenstate or a thermal state of $H_1$ and subsequently evolved in time under $H_2$. In each case, the initial state is spatially homogeneous and can be denoted by a density matrix $\hat{\rho}$. The corresponding energy density can be written as

$$\mathcal{E}_{\hat{\rho}}(x,t) = \text{Tr}\Big[ \hat{\rho}\, e^{iH_2 t} v_1 \big[ T_+(x) + T_-(x) \big] e^{-iH_2 t} \Big]$$
$$= \frac{2\pi v_1}{L^2} \sum_{n=-\infty}^{\infty} \text{Tr}\Big[ \hat{\rho}\, e^{iH_2 t} \big( L_n e^{2\pi i n x/L} + \bar{L}_n e^{-2\pi i n x/L} \big) e^{-iH_2 t} \Big] - \frac{\pi v_1}{6L^2}, \qquad (86)$$

where we used (26). The basic problem is therefore to study the quenched time evolution of the Virasoro generators $L_n$ and $\bar{L}_n$. For $L_n$, and analogously for $\bar{L}_n$, we can write

$$e^{iH_2 t} L_n e^{-iH_2 t} = \mathcal{U} \mathcal{I}_v^{(q)} \mathcal{I}_v^\dagger L_n \mathcal{I}_v \big( \mathcal{I}_v^{(q)} \big)^\dagger \mathcal{U}^\dagger, \qquad (87)$$

with

$$\mathcal{U} = q^{-(L_0+\bar{L}_0)} e^{i\left[ H_2^{(0)} - (v_2/v_1) H_1^{(0)} \right] t} \qquad (88)$$

and $q = e^{-2\pi i v_2 t/L}$, where we used $e^{-iH_2 t} = U_F$ and (55) for $t_1 = 0$ and $t_2 = t$. Momentarily, neglecting $\mathcal{U}$, the remaining object $\mathcal{I}_v^{(q)} \mathcal{I}_v^\dagger L_n \mathcal{I}_v \big( \mathcal{I}_v^{(q)} \big)^\dagger$ can be computed using (49) and the decomposition of $L_n$ into oscillator modes in (35). It takes the general form

$$\mathcal{I}_v^{(q)} \mathcal{I}_v^\dagger L_n \mathcal{I}_v \big( \mathcal{I}_v^{(q)} \big)^\dagger = \frac{1}{2} \sum_{m=-\infty}^{\infty} \Big[ C_{aa}^{(n)}(m) a_{n-m} a_m + C_{\bar{a}\bar{a}}^{(n)}(m) \bar{a}_{-n+m} \bar{a}_{-m}$$
$$+ C_{\bar{a}a}^{(n)}(m) \bar{a}_{-n+m} a_m + C_{a\bar{a}}^{(n)}(m) a_{n-m} \bar{a}_{-m} - \delta_{n,0} m \theta(m) \Big], \quad (89)$$

for certain coefficients $C_{(\cdot)(\cdot)}^{(n)}(m)$, where the last term comes from undoing the Wick ordering using (56). One can explicitly show that, when taking the trace in (86) with a spatially homogeneous $\hat{\rho}$, that the contributions from $\bar{a}_{-n+m} a_m$ and $a_{n-m} \bar{a}_{-m}$ vanish for all $n$, while those from $a_{n-m} a_m$ and $\bar{a}_{-n+m} \bar{a}_{-m}$ vanish unless $n = 0$, a fact which $\mathcal{U}$ in (88) cannot change. It follows that the only contributions we need to evaluate come from $L_0$ and $\bar{L}_0$ and that the energy density is constant in space. More concretely, $\mathcal{E}_{\hat{\rho}}(x,t) = \mathcal{E}_{\hat{\rho}}(t)$ with

$$\mathcal{E}_{\hat{\rho}}(t) = \frac{2\pi v_1}{L^2} \text{Tr}\Big[ \hat{\rho}\, \mathcal{U} \mathcal{I}_v^{(q)} \mathcal{I}_v^\dagger \big( L_0 + \bar{L}_0 \big) \mathcal{I}_v \big( \mathcal{I}_v^{(q)} \big)^\dagger \mathcal{U}^\dagger \Big] - \frac{\pi v_1}{6L^2}, \qquad (90)$$

where the only coefficients in (89) we need are

$$C_{aa}^{(0)}(m) = \frac{1}{4} \Big[ \big(1+q^{-2m}\big)\big(1+q^{2m}\big) + \cosh^2(2v)\big(1-q^{-2m}\big)\big(1-q^{2m}\big) \Big],$$
$$C_{\bar{a}\bar{a}}^{(0)}(m) = \frac{1}{4} \sinh^2(2v)\big(1-q^{-2m}\big)\big(1-q^{2m}\big), \qquad (91)$$

which manifestly satisfy $C_{aa}^{(0)}(-m) = C_{aa}^{(0)}(m)$ and $C_{\bar{a}\bar{a}}^{(0)}(-m) = C_{\bar{a}\bar{a}}^{(0)}(m)$.

To proceed, we specialize to different choices of the state $\hat{\rho}$ in which the system is initialized.

**For the ground state**

Consider the position-independent energy density $\mathcal{E}_{\Omega}(t) = \mathcal{E}_{\hat{\rho}}(t)$ in (90) for $\hat{\rho} = |\Omega\rangle\langle\Omega|$ given by the ground state $|\Omega\rangle$ of the theory with $K = K_1$. In this case, the contributions from $\mathcal{U}$ in (88) vanish since $a_0|\Omega\rangle = 0 = \bar{a}_0|\Omega\rangle$ and $L_0|\Omega\rangle = 0 = \bar{L}_0|\Omega\rangle$. It follows that

$$\text{Tr}\Big[\hat{\rho}\,\mathcal{U}\mathcal{I}_\nu^{(q)}\mathcal{I}_\nu^\dagger L_0 \mathcal{I}_\nu \big(\mathcal{I}_\nu^{(q)}\big)^\dagger \mathcal{U}^\dagger\Big] = \langle\Omega|\mathcal{I}_\nu^{(q)}\mathcal{I}_\nu^\dagger L_0 \mathcal{I}_\nu \big(\mathcal{I}_\nu^{(q)}\big)^\dagger |\Omega\rangle = \langle L_0\rangle_\Omega(t), \qquad (92)$$

which defines the time-dependent expectation

$$\begin{aligned}
\langle L_0\rangle_\Omega(t) &= \frac{1}{2}\sum_m \Big[ C_{aa}^{(0)}(m)\langle\Omega|a_{-m}a_m|\Omega\rangle + C_{\bar{a}\bar{a}}^{(0)}(m)\langle\Omega|\bar{a}_m\bar{a}_{-m}|\Omega\rangle - m\theta(m)\Big] \\
&= \frac{1}{2}\sum_{m>0} m\Big[ C_{aa}^{(0)}(m) + C_{\bar{a}\bar{a}}^{(0)}(m) - 1\Big] \\
&= \frac{1}{2}\sum_{m>0} m\Big[\cosh^2(2\nu) - \sinh^2(2\nu)\cos(4\pi m v_2 t/L) - 1\Big],
\end{aligned} \qquad (93)$$

where we used (91) and $q = \mathrm{e}^{-2\pi i v_2 t/L}$ in the last step. The corresponding expectation $\langle\bar{L}_0\rangle_\Omega(t)$ can be shown to be exactly the same.

We stress that the result in (93), in general, is not convergent when summing over $m$ and needs to be regularized. The appropriate regularization in this case is provided by the Lerch zeta function, $\zeta(s|v,w)$, see Appendix A.[11] This function satisfies the required finiteness and periodicity properties and is a natural generalization of the Riemann zeta function $\zeta(s)$ used in the regularization of the Casimir energy of the undeformed TLL theory. Using this function,

$$\langle L_0\rangle_\Omega(t) = \frac{1}{2}\Big[\cosh^2(2\nu) - 1\Big]\zeta(-1) - \frac{1}{4}\sinh^2(2\nu)\Big[\zeta(-1|0, 2v_2 t/L) + \zeta(-1|0, -2v_2 t/L)\Big], \quad (94)$$

where $\zeta(-1) = -1/12$ through analytic continuation. Inserting the above into (90), it follows that the ground-state energy density after the quantum quench is

$$\mathcal{E}_\Omega(t) = -\frac{\pi v_1}{6L^2}\Big(\cosh^2(2\nu) + 6\sinh^2(2\nu)\big[\zeta(-1|0, 2v_2 t/L) + \zeta(-1|0, -2v_2 t/L)\big]\Big). \qquad (95)$$

At $t = 0$, the function $\zeta(-1|0, \pm 2v_2 t/L)$ reduces to the Riemann zeta function, since $\zeta(-1|0,0) = \zeta(-1)$. This implies that the energy density at $t = 0$ is the familiar ground-state energy density of a TLL theory:

$$\mathcal{E}_\Omega(0) = -\frac{\pi v_1}{6L^2}. \qquad (96)$$

We note that the revivals observed in the Loschmidt echo at $t = kL/2v_2$ for $k \in \mathbb{N}$ are also present in the energy density. These lead to discontinuities at these discrete times, as seen in Fig. 4(a). Physically, the discontinuity in the vicinity of $t = 0$ appear due to the abrupt nature of the interaction quench, while the periodic revivals occur due to the integrability of the system.

---

[11]We are grateful to Pierre Vanhove for discussions on this.

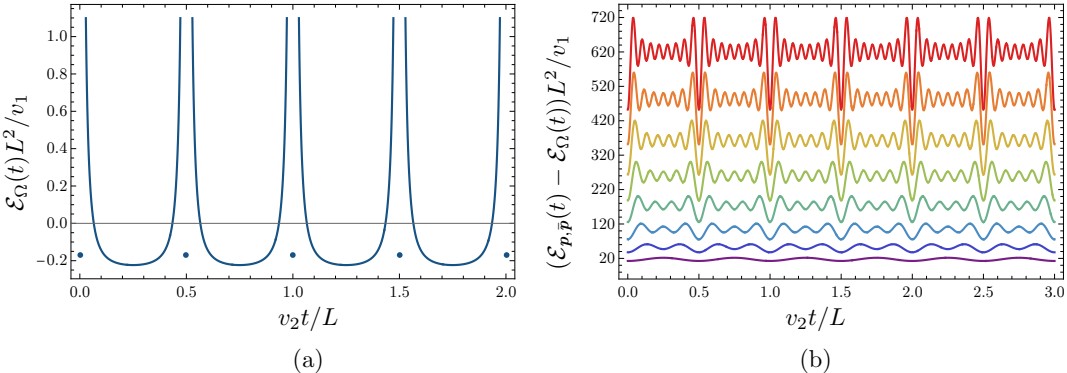

Figure 4: (a) Time evolution of the ground-state energy density $\mathcal{E}_\Omega(t)$ in (95) following a quench with $K_1/K_2 = 4/3$. We observe discontinuities in the evolution at $v_2 t/L = k/2$, $k \in \mathbb{N}$, where the energy density goes back to its equilibrium value. (b) Time evolution of the excitation contribution $\mathcal{E}_{\boldsymbol{p},\bar{\boldsymbol{p}}}(t) - \mathcal{E}_\Omega(t)$ given by (100) following the same quench for initial states of the form $a_{-m}\bar{a}_{-m}|\Omega\rangle$, $m = 1,...,8$ (bottom to top).

**For excited states**

Consider now instead the system initialized in the state $\hat{\rho} = |\Psi_{\boldsymbol{p},\bar{\boldsymbol{p}}}\rangle\langle\Psi_{\boldsymbol{p},\bar{\boldsymbol{p}}}|$ with $|\Psi_{\boldsymbol{p},\bar{\boldsymbol{p}}}\rangle$ of the form in (65). As for the computation of the ground-state energy density, the contributions from $\mathcal{U}$ and $\mathcal{U}^\dagger$ given by (88) can be shown to cancel each other. Thus, similar to before,

$$\text{Tr}\Big[\hat{\rho}\,\mathcal{U}\mathcal{I}_\nu^{(q)}\mathcal{I}_\nu^\dagger L_0 \mathcal{I}_\nu\big(\mathcal{I}_\nu^{(q)}\big)^\dagger \mathcal{U}^\dagger\Big] = \langle\Psi_{\boldsymbol{p},\bar{\boldsymbol{p}}}|\mathcal{I}_\nu^{(q)}\mathcal{I}_\nu^\dagger L_0 \mathcal{I}_\nu\big(\mathcal{I}_\nu^{(q)}\big)^\dagger|\Psi_{\boldsymbol{p},\bar{\boldsymbol{p}}}\rangle = \langle L_0\rangle_{\boldsymbol{p},\bar{\boldsymbol{p}}}(t), \qquad (97)$$

with

$$\langle L_0\rangle_{\boldsymbol{p},\bar{\boldsymbol{p}}}(t) = \frac{1}{2}\sum_m\Big[C_{aa}^{(0)}(m)\langle\Psi_{\boldsymbol{p},\bar{\boldsymbol{p}}}|a_{-m}a_m|\Psi_{\boldsymbol{p},\bar{\boldsymbol{p}}}\rangle + C_{\bar{a}\bar{a}}^{(0)}(m)\langle\Psi_{\boldsymbol{p},\bar{\boldsymbol{p}}}|\bar{a}_m\bar{a}_{-m}|\Psi_{\boldsymbol{p},\bar{\boldsymbol{p}}}\rangle - m\theta(m)\Big]$$

$$= \frac{1}{2}\sum_{m>0}\Big[C_{aa}^{(0)}(m)(2p_m+1)m + C_{\bar{a}\bar{a}}^{(0)}(m)(2\bar{p}_m+1)m - m\Big]$$

$$= \langle L_0\rangle_\Omega(t) + \sum_{m>0} m\Big[C_{aa}^{(0)}(m)p_m + C_{\bar{a}\bar{a}}^{(0)}(m)\bar{p}_m\Big]. \qquad (98)$$

In the second step, we used $\langle\Omega|a_n^{p_n}a_{-n}a_n a_{-n}^{p_n}|\Omega\rangle = np_n(n^{p_n}p_n!)$ for $n > 0$ to show that

$$\begin{aligned}
&\langle\Psi_{\boldsymbol{p},\bar{\boldsymbol{p}}}|a_{-m}a_m|\Psi_{\boldsymbol{p},\bar{\boldsymbol{p}}}\rangle = mp_m\,, && \langle\Psi_{\boldsymbol{p},\bar{\boldsymbol{p}}}|a_m a_{-m}|\Psi_{\boldsymbol{p},\bar{\boldsymbol{p}}}\rangle = m(p_m+1)\,,\\
&\langle\Psi_{\boldsymbol{p},\bar{\boldsymbol{p}}}|\bar{a}_m\bar{a}_{-m}|\Psi_{\boldsymbol{p},\bar{\boldsymbol{p}}}\rangle = m(\bar{p}_m+1)\,, && \langle\Psi_{\boldsymbol{p},\bar{\boldsymbol{p}}}|\bar{a}_{-m}\bar{a}_m|\Psi_{\boldsymbol{p},\bar{\boldsymbol{p}}}\rangle = m\bar{p}_m
\end{aligned} \qquad (99)$$

for all $m \in \mathbb{Z}^+$, as well as the symmetry properties of the coefficients in (91). The first term in (98) is the ground-state contribution in (94), while the second term is the additional contribution depending on the occupation numbers of the excited initial state in (65). For the corresponding expectation $\langle\bar{L}_0\rangle_{\boldsymbol{p},\bar{\boldsymbol{p}}}(t)$ one simply needs to swap the roles of $C_{aa}$ and $C_{\bar{a}\bar{a}}$.

It follows by inserting the above into (90) that the position-independent energy density $\mathcal{E}_{\boldsymbol{p},\bar{\boldsymbol{p}}}(t) = \mathcal{E}_{\hat{\rho}}(t)$ for $\hat{\rho} = |\Psi_{\boldsymbol{p},\bar{\boldsymbol{p}}}\rangle\langle\Psi_{\boldsymbol{p},\bar{\boldsymbol{p}}}|$ given by (65) is

$$\mathcal{E}_{\boldsymbol{p},\bar{\boldsymbol{p}}}(t) = \mathcal{E}_\Omega(t) + \frac{2\pi v_1}{L^2}\sum_{m>0}\Big[C_{aa}^{(0)}(m) + C_{\bar{a}\bar{a}}^{(0)}(m)\Big]m(p_m + \bar{p}_m)$$

$$= \mathcal{E}_\Omega(t) + \frac{2\pi v_1}{L^2}\sum_{m>0}\Big[\cosh^2(2\nu) - \sinh^2(2\nu)\cos(4\pi m v_2 t/L)\Big]m(p_m + \bar{p}_m), \quad (100)$$

with $\mathcal{E}_\Omega(t)$ in (95), where we used (91) and $q = \mathrm{e}^{-2\pi \mathrm{i} v_2 t/L}$. We plot the time evolution of the excitation contribution $\mathcal{E}_{p,\bar{p}}(t) - \mathcal{E}_\Omega(t)$ in Fig. 4(b). The energy density still oscillates in time with period $L/2v_2$ and reaches its minimum at the energy density of the given level $\sum_{m>0} m(p_m + \bar{p}_m)$. In contrast to the Loschmidt echo, the time evolution of the energy density does not crucially depend on whether or not the initial state mixes right- and left-moving excitations.

As a remark, note that we only considered initial states that are descendants of the ground state $|\Omega\rangle$. Non-zero contributions from the zero modes appear if one considers initial states that are descendants of other primary states than the ground state. These are, however, constant shifts of the energy density corresponding to the conformal dimensions of the primary states and are sub-leading in the system size.

**For thermal states**

Lastly, we study the position-independent energy density $\mathcal{E}_\beta(t) = \mathcal{E}_{\hat{\rho}}(t)$ for an initial thermal state given by $\hat{\rho} = Z_1^{-1} \mathrm{e}^{-\beta H_1}$, where $Z_1 = \mathrm{Tr}\big(\mathrm{e}^{-\beta H_1}\big)$ is the partition function of the undeformed theory with $K = K_1$. To compute $\mathcal{E}_\beta(t)$, it follows from (89) and (90) (and the discussion between them) that we need to evaluate

$$\mathrm{Tr}\Big[\hat{\rho}\,\mathcal{U}\mathcal{I}_\nu^{(q)}\mathcal{I}_\nu^\dagger L_0 \mathcal{I}_\nu \big(\mathcal{I}_\nu^{(q)}\big)^\dagger \mathcal{U}^\dagger\Big] = \frac{1}{2}\sum_m \left(C_{aa}^{(0)}(m)\frac{\mathrm{Tr}\big[\mathrm{e}^{-\beta H_1} a_{-m} a_m\big]}{\mathrm{Tr}\big[\mathrm{e}^{-\beta H_1}\big]} + C_{\bar{a}\bar{a}}^{(0)}(m)\frac{\mathrm{Tr}\big[\mathrm{e}^{-\beta H_1}\bar{a}_m \bar{a}_{-m}\big]}{\mathrm{Tr}\big[\mathrm{e}^{-\beta H_1}\big]} - m\theta(m)\right), \quad (101)$$

together with the corresponding expectation for $\bar{L}_0$. In the second line, we used that $\mathcal{U}$ and $\mathcal{U}^\dagger$ given by (88) cancel due to cyclicity of the trace. The remaining traces appearing in (101) are thermal expectation values of bosonic occupation numbers and can be calculated using the following manipulation for the oscillator modes ($m \neq 0$):

$$\mathrm{Tr}\Big[a_{-m} a_m z^{L_0 + \bar{L}_0}\Big] = z^m \,\mathrm{Tr}\Big[a_{-m} z^{L_0 + \bar{L}_0} a_m\Big] = z^m \,\mathrm{Tr}\big[a_m a_{-m} z^{L_0 + \bar{L}_0}\big]$$
$$= m z^m \,\mathrm{Tr}\Big[z^{L_0 + \bar{L}_0}\Big] + z^m \,\mathrm{Tr}\Big[a_{-m} a_m z^{L_0 + \bar{L}_0}\Big]. \quad (102)$$

Setting $z = \mathrm{e}^{-2\pi v_1 \beta/L}$ in the above, we obtain

$$\frac{\mathrm{Tr}\big[\mathrm{e}^{-\beta H_1} a_{-m} a_m\big]}{\mathrm{Tr}\big[\mathrm{e}^{-\beta H_1}\big]} = \frac{\mathrm{Tr}\big[a_{-m} a_m z^{L_0 + \bar{L}_0}\big]}{\mathrm{Tr}\big[z^{L_0 + \bar{L}_0}\big]} = \frac{m z^m}{1 - z^m} = \frac{m}{\mathrm{e}^{2\pi m v_1 \beta/L} - 1} = \langle a_{-m} a_m\rangle_{v_1\beta/L} \quad (103)$$

for $m \neq 0$, which reproduces the expected Bose-Einstein occupation number $\langle a_{-m} a_m\rangle_{v_1\beta/L}$. The same result is true for $\langle \bar{a}_m \bar{a}_{-m}\rangle_{v_1\beta/L}$. The corresponding time-dependent expectation of $L_0$ can therefore be expressed as

$$\langle L_0\rangle_{v_1\beta/L}(t) = \mathrm{Tr}\Big[\hat{\rho}\,\mathcal{U}\mathcal{I}_\nu^{(q)}\mathcal{I}_\nu^\dagger L_0 \mathcal{I}_\nu \big(\mathcal{I}_\nu^{(q)}\big)^\dagger \mathcal{U}^\dagger\Big] = \langle L_0^{(0)}\rangle_{v_1\beta/L} + \langle L_0^{(\mathrm{osc})}\rangle_{v_1\beta/L}(t), \quad (104)$$

where $\langle L_0^{(0)}\rangle_{v_1\beta/L}$ and $\langle L_0^{(\mathrm{osc})}\rangle_{v_1\beta/L}(t)$ are the contributions to the expectation value from the zero- and oscillator-mode parts of $L_0$, respectively. The zero-mode part is constant in time and sub-leading in the system size $L$, and thus not relevant to the post-quench dynamical properties. However, we present it here for completeness:

$$\langle L_0^{(0)}\rangle_{v_1\beta/L} = \frac{1}{4\Theta(v_1\beta/L)}\sum_{n,w\in\mathbb{Z}}\left(\frac{n^2}{2K_1} + 2w^2 K_1\right)\exp\left[-\pi\frac{v_1\beta}{L}\left(\frac{n^2}{2K_1} + 2w^2 K_1\right)\right]$$
$$= -\frac{L}{4\pi v_1}\frac{\partial \ln\Theta(v_1\beta/L)}{\partial\beta}, \quad (105)$$

where $\Theta$ is the Siegel theta function [cf. (171)]. On the other hand, the oscillator part depends non-trivially on time:

$$\langle L_0^{(osc)} \rangle_{v_1\beta/L}(t) = \frac{1}{2} \sum_{m\neq 0} \Big[ C_{aa}^{(0)}(m) \langle a_{-m} a_m \rangle_{v_1\beta/L} + C_{\bar{a}\bar{a}}^{(0)}(m) \langle \bar{a}_m \bar{a}_{-m} \rangle_{v_1\beta/L} - m\theta(m) \Big], \quad (106)$$

with $C_{aa}^{(0)}(m)$ and $C_{\bar{a}\bar{a}}^{(0)}(m)$ in (91) and $q = e^{-2\pi i v_2 t/L}$. From (56), we have

$$\langle :a_{-m} a_m: \rangle_{v_1\beta/L} = \langle a_{-m} a_m \rangle_{v_1\beta/L} + m\theta(-m),$$

where

$$\langle :a_{-m} a_m: \rangle_{v_1\beta/L} = \frac{|m|}{e^{2\pi |m| v_1\beta/L} - 1}, \quad (107)$$

which implies

$$\langle L_0^{(osc)} \rangle_{v_1\beta/L}(t) = \langle L_0 \rangle_{\Omega}(t) + \sum_{m>0} \frac{m\big[\cosh^2(2v) - \sinh^2(2v)\cos(4\pi m v_2 t/L)\big]}{e^{2\pi m v_1\beta/L} - 1}, \quad (108)$$

where the first term $\langle L_0 \rangle_{\Omega}(t)$ is given in (93). By inserting the above together with the analogous expressions for $\langle \bar{L}_0^{(0)} \rangle_{v_1\beta/L}$ into (90), we obtain the final result:

$$\mathcal{E}_\beta(t) = \mathcal{E}_\Omega(t) - \frac{1}{L} \frac{\partial \ln\Theta(v_1\beta/L)}{\partial \beta} + \frac{4\pi v_1}{L^2} \sum_{m>0} \frac{m\big[\cosh^2(2v) - \sinh^2(2v)\cos(4\pi m v_2 t/L)\big]}{e^{2\pi m v_1\beta/L} - 1}, \quad (109)$$

with $\mathcal{E}_\Omega(t)$ in (95).

As a consistency check, the equilibrium expectation value can be obtained from (105) and (108) by setting $t = 0$, yielding

$$\langle L_0 \rangle_{v_1\beta/L}(0) = -\frac{L}{4\pi v_1} \frac{\partial \ln\Theta(v_1\beta/L)}{\partial \beta} + \sum_{m>0} \frac{m}{e^{2\pi m v_1\beta/L} - 1}, \quad (110)$$

which is consistent with $\langle L_0 \rangle_{v_1\beta/L}$ obtained from (35) using (107).[12] The oscillator part can be expressed using a quasimodular form $E_2(\tau)$ known as the Eisenstein series of weight 2:

$$\langle L_0^{(osc)} \rangle_{v_1\beta/L}(0) = \sum_{m>0} \frac{m}{e^{2\pi m v_1\beta/L} - 1} = \sum_{m>0} \frac{mz^m}{1 - z^m} = -\frac{E_2(\tau)}{24} + \frac{1}{24}, \quad (111)$$

where $z = e^{2\pi i \tau}$ and $\tau = i v_1\beta/L$. The S-modular transformation

$$E_2(\tau) = (-1/\tau)^2 E_2(-1/\tau) - 6/\pi i\tau$$

can be used to extract the asymptotic behavior of (111) for $L/v_1\beta \gg 1$:

$$E_2(i v_1\beta/L) \approx -\frac{L^2}{v_1^2\beta^2} + \frac{6L}{\pi v_1\beta} \approx -\frac{L^2}{v_1^2\beta^2}, \quad (112)$$

which yields

$$\langle L_0^{(osc)} \rangle_{v_1\beta/L} \approx \frac{L^2}{24 v_1^2\beta^2}. \quad (113)$$

---

[12]This is also consistent with the relation between the torus one-point function of the holomorphic stress tensor and the partition function: $\langle L_0 - c/24 \rangle_\tau = (2\pi i)^{-1} \partial_\tau \log Z(\tau, \bar{\tau})$. For our case $\tau = i v_1\beta/L$ and the partition function is given in (171).

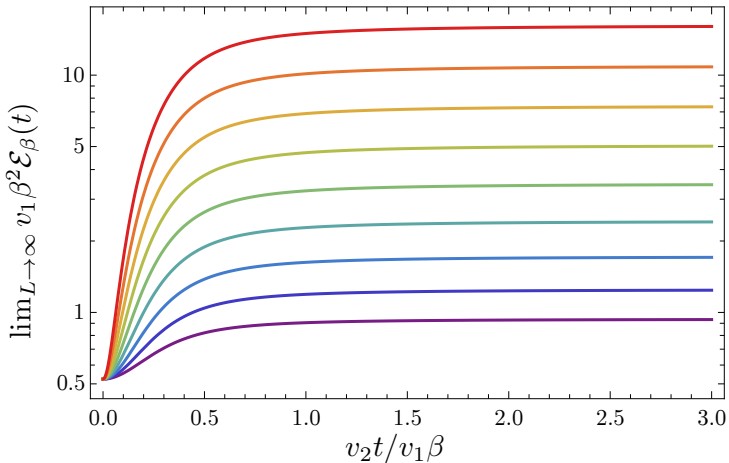

Figure 5: Time evolution of the thermal-state energy density $\lim_{L\to\infty}\mathcal{E}_\beta(t)$ in (117) in the thermodynamic limit following a quench with $K_1/K_2 = e^{2\nu}$, $\nu = 0.4, 0.5, ..., 1.2$ (from bottom to top). We observe that the TLL equilibrates to different energies depending on $\nu$ according to (118) with the effective temperature $\beta_{\text{eff}}^{-1}$ in (119).

This gives the expected equilibrium energy density in the thermodynamic limit $L \to \infty$. Indeed, by inserting the above into (90), it follows that

$$\lim_{t\to 0}\lim_{L\to\infty}\mathcal{E}_\beta(t) = \frac{\pi}{6v_1\beta^2}\,, \tag{114}$$

which is exact in the thermodynamic limit.

The evolution of the energy density from the thermal state can be evaluated analytically in the thermodynamic limit by replacing the sums over $m$ in (108) by integrals with respect to the dimensionless variable $\xi = 2\pi m v_1 \beta/L$. The integrals can then be performed by using the following identities:

$$\int_0^\infty d\xi\, \frac{\xi}{e^\xi - 1} = \frac{\pi^2}{6}\,, \qquad \int_0^\infty d\xi\, \frac{\xi\cos(w\xi)}{e^\xi - 1} = \frac{1}{2w^2} - \frac{\pi^2}{2\sinh^2(\pi w)} \quad (w \in \mathbb{R}^+). \tag{115}$$

In our case, $w = 2v_2 t/v_1\beta$. We conclude that

$$\langle L_0^{(\text{osc})}\rangle_{v_1\beta/L}(t) - \langle L_0\rangle_\Omega(t) \approx \frac{L^2\cosh^2(2\nu)}{24v_1^2\beta^2} - \frac{L^2\sinh^2(2\nu)}{8v_1^2\beta^2}\left[\left(\frac{v_1\beta}{2\pi v_2 t}\right)^2 - \text{csch}^2\left(\frac{2\pi v_2 t}{v_1\beta}\right)\right] \tag{116}$$

in the regime $L/v_1\beta \gg \infty$, with the same result for $\langle \bar{L}_0^{(\text{osc})}\rangle_{v_1\beta/L}(t)$. In the thermodynamic limit, at which point the results become exact, it finally follows from (90) that the energy density for the quenched thermal state is

$$\lim_{L\to\infty}\mathcal{E}_\beta(t) = \frac{\pi\cosh^2(2\nu)}{6v_1\beta^2} - \frac{\pi\sinh^2(2\nu)}{2v_1\beta^2}\left[\left(\frac{v_1\beta}{2\pi v_2 t}\right)^2 - \text{csch}^2\left(\frac{2\pi v_2 t}{v_1\beta}\right)\right], \tag{117}$$

where we used that $\langle L_0\rangle_\Omega(t)$, $\langle \bar{L}_0\rangle_\Omega(t)$, and the zero modes give sub-leading contributions in the system size $L$. In particular, the late-time asymptotic behavior in this regime is[13]

$$\lim_{t\to\infty}\lim_{L\to\infty}\mathcal{E}_\beta(t) = \frac{\pi}{6v_1\beta_{\text{eff}}^2}\,, \tag{118}$$

---

[13]Note that (114) is also recovered from (117) using that $\text{csch}(\xi) = \xi^{-1} - \xi/6 + O(\xi^3)$ for small $\xi$.

where we defined the effective temperature

$$\beta_{\text{eff}}^{-1} = \beta^{-1} \cosh(2\nu) = \beta^{-1} \frac{K_1/K_2 + K_2/K_1}{2}. \tag{119}$$

We thus observe, in the thermodynamic limit, an equilibration of the original TLL following the quench from an initial temperature $\beta^{-1}$ to an emergent temperature: The time evolution of the energy-density expectation reaches that of a steady state at an effective temperature $\beta_{\text{eff}}^{-1}$ given by (119), as seen in Fig. 5. We note that a similar large-scale equilibration to an effective temperature was observed in quenched TLLs in [79] for a different type of quenching protocol and through different physical observables.

## 5  Floquet drive

In this section, we study a two-step driven TLL whose Hamiltonian switches periodically between $H_1$ and $H_2$ with periods $t_1$ and $t_2$, respectively, as illustrated in Fig. 1(b). We recall that the two Hamiltonians $H_1$ and $H_2$ are particular combinations of the $\mathfrak{su}(1,1)$ generators $K_0^{(n)}$ and $K_\pm^{(n)}$ in (41) for each individual mode $n > 0$ [see (46) and (47)]. It follows that the Floquet operator $U_F^{(n)}$ in (9) for the $n$th mode can be expressed as $U_F^{(n)} = e^{-iH_F^{(n)}(t_1+t_2)}$ using a Floquet Hamiltonian $H_F^{(n)}$ that is also a combination of the $\mathfrak{su}(1,1)$ generators:

$$H_F^{(n)} = \frac{i}{t_1 + t_2} C_F^{(n)}, \qquad C_F^{(n)} = c_0^{(n)} K_0^{(n)} + c_-^{(n)} K_-^{(n)} + c_+^{(n)} K_+^{(n)}, \tag{120}$$

for certain coefficients $c_0^{(n)}$ and $c_\pm^{(n)}$. Our driven TLL thus corresponds to an infinite sequence of uncoupled discrete-time quantum parametric oscillators labeled by $n$. Consequently, as for harmonic oscillators with continuously and periodically driven frequency, see, e.g., [80, 81], famously leading to the Mathieu equation, similar algebraic stability arguments can be employed here. Namely, for each mode, a characterization into stable or unstable can be deduced from the different classes of orbits of $\mathfrak{su}(1,1)$, see Table 1. These are delineated by the value $\mathcal{K}(H_F^{(n)}, H_F^{(n)}) = -2(t_1+t_2)^{-2}\big[(c_0^{(n)})^2 - 4c_+^{(n)}c_-^{(n)}\big]$ of the Cartan-Killing form $\mathcal{K}(\cdot,\cdot)$ in (43), or equivalently by the squared trace $\sigma_n = \big(\text{Tr}[U_F^{(n)}]\big)^2$, which are related through

$$\sigma_n = 4\cosh^2\left(\sqrt{\mathcal{K}(C_F^{(n)}, C_F^{(n)})/8}\right) = 4\cos^2\left((t_1+t_2)\sqrt{\mathcal{K}(H_F^{(n)}, H_F^{(n)})/8}\right). \tag{121}$$

Using the $2 \times 2$-matrix representation of the $\mathfrak{su}(1,1)$ generators in (44), the coefficients in (120) can be computed, which inserted into (121) yields exactly $\sigma_n$ in (11) with $q_{1,2}$ in (4). This can be rewritten as

$$\sigma_n = 4\omega_n^2, \qquad \omega_n = \cos(2\pi n\tau_1)\cos(2\pi n\tau_2) - \sin(2\pi n\tau_1)\sin(2\pi n\tau_2)\cosh(2\nu), \tag{122}$$

which shows the explicit dependence on the dimensionless times $(\tau_1, \tau_2) = (v_1 t_1/L, v_2 t_2/L)$ and the Zamolodchikov distance $\nu$. Note that $\omega_n$ and $\sigma_n$ are manifestly invariant under change of sign in $n$.

Dynamical phase diagrams in the parameter space $(\tau_1, \tau_2)$ can be straightforwardly drawn using (122) for a given mode $n$ and Zamolodchikov distance $\nu$, computed for a pair of Luttinger parameters $(K_1, K_2)$ or radii $(R_1, R_2)$ through (8). Note that the phase diagram for any mode $n \in \mathbb{Z}^+$ is simply a rescaling of the phase diagram of that for $n = 1$, obtained by replacing $L$ by $L/n$, with each of its individual unstable regions having the shape of a leaf that shrinks to a line as $\nu \to 0$ and grows to approximate a square as $|\nu| \to \infty$, as illustrated in Fig. 6. The total

Table 1: Classes for a given mode $n \in \mathbb{Z}^+$ depending on the value $\mathcal{K}(H_F^{(n)}, H_F^{(n)})$ of the Cartan-Killing form or the squared trace $\sigma_n$ in (121) along with the corresponding stability characterization.

| Class | $\mathcal{K}(H_F^{(n)}, H_F^{(n)})$ | $\sigma_n$ | Stability characterization |
|---|---|---|---|
| Elliptic | $> 0$ | $< 4$ | Stable phase |
| Parabolic | $= 0$ | $= 4$ | Phase boundary |
| Hyperbolic | $< 0$ | $> 4$ | Unstable phase |

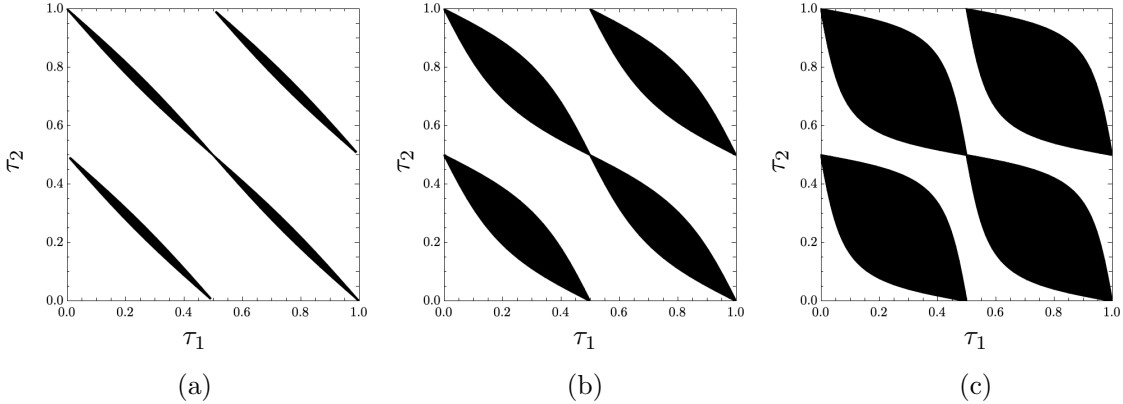

Figure 6: Dynamical phase diagrams in $(\tau_1, \tau_2)$ space for a single mode $n = 1$ and (a) $K_1/K_2 = 16/15$, (b) $K_1/K_2 = 7/5$, and (c) $K_1/K_2 = 7/3$.

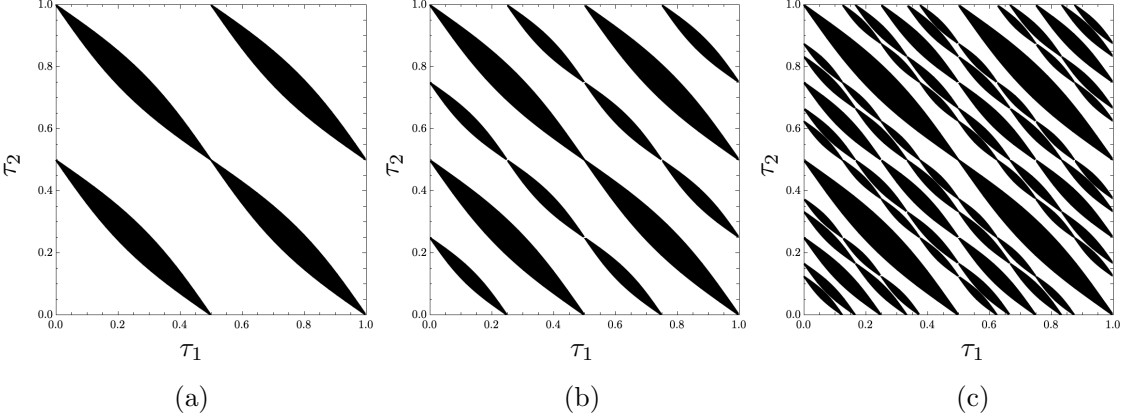

Figure 7: Dynamical phase diagrams in $(\tau_1, \tau_2)$ space for $K_1/K_2 = 1.2$ for a finite number of modes. The unstable regions are colored black. (a) Single mode $n = 1$. (b) Two modes $n = 1, 2$. (c) Four modes $n = 1, ..., 4$. Note that the lines $(k/2, \tau_2)$ and $(\tau_1, k/2)$, $k \in \mathbb{N}$ remain critical or stable even when an arbitrary number of modes are included. The phase diagrams are plotted for $(\tau_1, \tau_2) \in [0, 1] \times [0, 1]$ since they repeat themselves outside this domain.

phase diagram is obtained by overlaying the phase diagrams of each individual mode, with the unstable phase being the union of the unstable regions, see Fig. 7. Any remaining stable phase thus depends crucially on $\nu$ and the number of modes included. In particular, imposing a (physical) cutoff on the total number of allowed modes would ensure that an extended stable phase remains.

Below we investigate the physical consequences of the dynamical phases in Table 1 on

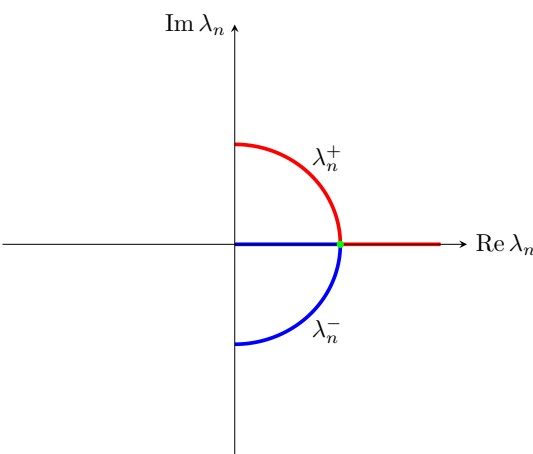

Figure 8: Eigenvalues $\lambda_n^+$ (red curve) and $\lambda_n^-$ (blue curve) in (123) as functions of $\sigma_n$ for the case $\mathrm{sgn}(\omega_n) > 0$. If $\sigma_n < (>) 4$, the eigenvalues lie on the unit circle (real line) and the $n$th-mode contribution is stable (unstable). The green dot corresponds to $\sigma_n = 4$, i.e., the value where $\lambda_n^\pm$ coincide.

certain physical quantities, specifically the Loschmidt echo and the particle and energy densities. We also identify natural order parameters and study their critical behavior near the phase boundary when approaching from the stable or the unstable phase. In a nutshell, we will show that the evolution of these physical quantities in stroboscopic time $M(t_1 + t_2)$ enters through factors of the form $(\lambda_n^\pm)^M$ with

$$\lambda_n^\pm = \omega_n \pm \mathrm{sgn}(\omega_n)\sqrt{\omega_n^2 - 1} = \mathrm{sgn}(\omega_n)\frac{\sqrt{\sigma_n} \pm \sqrt{\sigma_n - 4}}{2}, \tag{123}$$

which depends on $(\tau_1, \tau_2)$ and $\nu$ through $\sigma_n$ and $\omega_n$ in (122). The factors $\lambda_n^\pm$ can be interpreted as eigenvalues of a $2 \times 2$-matrix representation of our Floquet drive and have distinct behaviors for the following three cases:[14]

1. If $\sigma_n < 4$, then $\lambda_n^\pm = \mathrm{sgn}(\omega_n)\mathrm{e}^{\pm \mathrm{i}\phi}$ for $\phi = \arctan\left(\sqrt{(4 - \sigma_n)/\sigma_n}\right) \in (0, \pi/2]$.

2. If $\sigma_n = 4$, then $\lambda_n^\pm = \mathrm{sgn}(\omega_n)$.

3. If $\sigma_n > 4$, then $|\lambda_n^+| > 1 > |\lambda_n^-| > 0$.

In other words, $\lambda_n^\pm$ lie on segments of the unit circle in the complex plane when $0 < \sigma_n < 4$, starting at $\pm \mathrm{i}\,\mathrm{sgn}(\omega_n)$ for $\sigma_n = 0^+$ and moving toward $\mathrm{sgn}(\omega_n)$ as $\sigma_n$ grows toward 4, coinciding at $\mathrm{sgn}(\omega_n)$ exactly when $\sigma_n = 4$, and then moving on the real line in $\pm\mathrm{sgn}(\omega_n)$ directions as $\sigma_n$ grows beyond 4, see Fig. 8. Given that the stroboscopic time evolution enters as $(\lambda_n^\pm)^M$, this agrees with our stability discussion for individual modes based on classes of $\mathfrak{su}(1,1)$, see Table 1. In particular, if $\sigma_n > 4$, there are parametric instabilities since $|\lambda_n^+|^M$ diverges as $M$ increases, while if $\sigma_n < 4$, there are oscillations of the form $\mathrm{e}^{\pm\mathrm{i}M\phi}$ with $M$.

The above stability analysis is analogous to the well-known discussion of the quantum parametric oscillator, see, e.g., [81]. Indeed, while our periodic drive is step-like and not continuous, we can identify the corresponding quantities to construct phase diagrams of the same form as in [46] obtained from the Mathieu equation for a continuously driven TLL, see Fig. 9. In particular, the Mathieu characteristic exponent is identified with $\phi = \arctan\left(\sqrt{(4 - \sigma_n)/\sigma_n}\right)$ introduced above, see Fig. 9(b), and the amplitude of the drive with the ratio of Luttinger parameters, see Fig. 9(a).

---

[14]In the last case, we use that $\partial\lambda_n^\pm/\partial\sigma_n \gtrless 0$.

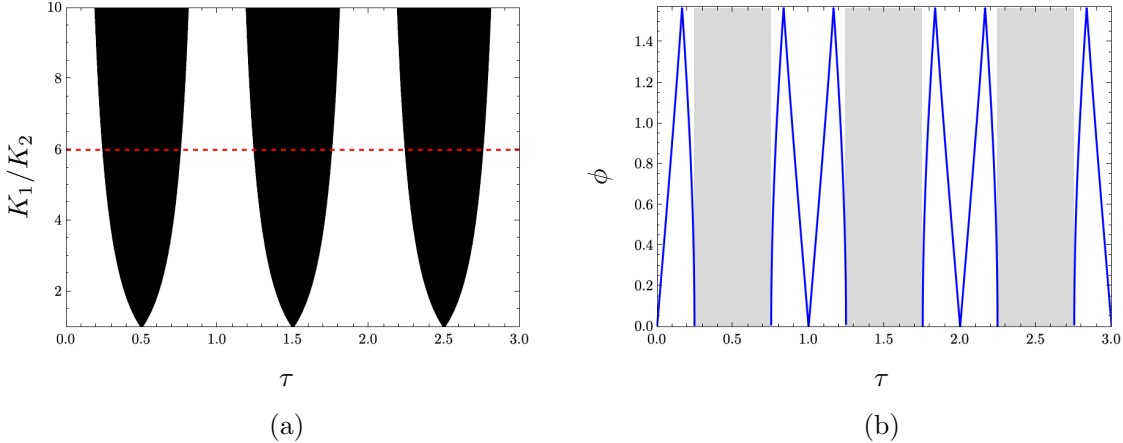

Figure 9: (a) Phase diagram for $n = 1$ when $\tau_1 = \tau_2$ parametrized in terms of $\tau = \tau_1 + \tau_2$ and $K_1/K_2$, interpreted as the period and the amplitude of the drive, respectively. The unstable regions are colored black. (b) Plot of $\phi = \arctan\left(\sqrt{(4-\sigma_1)/\sigma_1}\right)$, interpreted as the Mathieu characteristic exponent, using $\sigma_1$ in (122) as a function of $\tau = \tau_1/2 = \tau_2/2$ along the red dashed line ($K_1/K_2 = 6$) in (a). The blue curves give the plotted values when $\phi$ is real, and the shaded areas correspond to the unstable regions.

## 5.1 Loschmidt echo

The first quantity we study is the Loschmidt echo for the system initialized in the ground state or any excited state of $H_1$. We recall that the Floquet operator in (6) can be written as in (55). Similarly, it will also be convenient to express the $M$-cycle Floquet operator as a concatenation of several $q$-modified operators:

$$U_F^M = e^{-iME_2^0 t_2} q_1^{L_0+\bar{L}_0} \left[ \prod_{j=0}^{M-1} \mathcal{I}_\nu^{(q_1^j q_2^j)} \left( \mathcal{I}_\nu^{(q_1^j q_2^{j+1})} \right)^\dagger \right] (q_1^{M-1} q_2^M)^{L_0+\bar{L}_0} e^{-iM\left[H_2^{(0)}-(v_2/v_1)H_1^{(0)}\right]t_2} \quad (124)$$

for $M = 1, 2, \ldots$, generalizing (55). We recall that the overall phase $e^{-iME_2^0 t_2}$ will be of no consequence to our computations.

**For the ground state**

We begin by computing the Loschmidt echo $L_\Omega(M[t_1 + t_2]) = \left| \langle\Omega| U_F^M |\Omega\rangle \right|^2$ for the ground state $|\Omega\rangle$ of $H_1$ after $M$ cycles. Using (124) and the fact that $L_0$, $\bar{L}_0$, and $H_{1,2}^{(0)}$ annihilate $|\Omega\rangle$, we have

$$L_\Omega(M[t_1 + t_2]) = \left| \langle\Omega| \prod_{j=0}^{M-1} \mathcal{I}_\nu^{(q_1^j q_2^j)} \left( \mathcal{I}_\nu^{(q_1^j q_2^{j+1})} \right)^\dagger |\Omega\rangle \right|^2. \quad (125)$$

As in Sec. 4.1, our strategy to compute the product of the $q$-modified operators $\mathcal{I}_\nu^{(q)}$ is to first decompose them in terms of exponentials of the $\mathfrak{su}(1,1)$ generators in (41):

$$\langle\Omega| \prod_{j=0}^{M-1} \mathcal{I}_\nu^{(q_1^j q_2^j)} \left( \mathcal{I}_\nu^{(q_1^j q_2^{j+1})} \right)^\dagger |\Omega\rangle = \prod_{n>0} \langle\Omega| \exp\left(\xi_+^{(n)} K_+^{(n)}\right) \exp\left(\xi_0^{(n)} K_0^{(n)}\right) \exp\left(\xi_-^{(n)} K_-^{(n)}\right) |\Omega\rangle$$

$$= \prod_{n>0} \exp\left(\xi_0^{(n)}/2\right), \quad (126)$$

repeating the same steps as for the quench. Again, one efficient way to find the coefficients $\xi_0^{(n)}$ and $\xi_\pm^{(n)}$ is to use the $2 \times 2$-matrix representation of the $\mathfrak{su}(1,1)$ generators in (44). In this representation, using (48) and (52) with $\left(\mathcal{I}_\nu^{(q)}\right)^\dagger = \mathcal{I}_{-\nu}^{(q)}$ as a definition, we find that the $j$th factor in the product of $q$-modified operators has the form

$$\mathcal{I}_\nu^{(q_1^j q_2^j)}\left(\mathcal{I}_\nu^{(q_1^j q_2^{j+1})}\right)^\dagger\Big|_{2\times 2}^{(n)} = \begin{pmatrix} \cosh^2(\nu) - \sinh^2(\nu)q_2^{2n} & \frac{1}{2}\sinh(2\nu)\left(1 - q_2^{-2n}\right)q_2^{-2jn}q_1^{-2jn} \\ \frac{1}{2}\sinh(2\nu)\left(1 - q_2^{2n}\right)q_2^{2jn}q_1^{2jn} & \cosh^2(\nu) - \sinh^2(\nu)q_2^{-2n} \end{pmatrix} \quad (127)$$

for the $n$th mode. In analogy with (63), we have

$$e^{\xi_+^{(n)}K_+^{(n)}} e^{\xi_0^{(n)}K_0^{(n)}} e^{\xi_-^{(n)}K_-^{(n)}}\Big|_{2\times 2} = \begin{pmatrix} e^{-\xi_0^{(n)}/2} & \xi_-^{(n)}e^{-\xi_0^{(n)}/2} \\ -\xi_+^{(n)}e^{-\xi_0^{(n)}/2} & e^{-\xi_0^{(n)}/2} - \xi_-^{(n)}\xi_+^{(n)}e^{-\xi_0^{(n)}/2} \end{pmatrix}, \quad (128)$$

meaning that, at a practical level, we only need the $(1,1)$-component in the $2 \times 2$-matrix representation to determine $\exp\left(\xi_0^{(n)}/2\right)$ and thereby evaluate (126). Let us denote

$$\prod_{j=0}^{M-1}\mathcal{I}_\nu^{(q_1^j q_2^j)}\left(\mathcal{I}_\nu^{(q_1^j q_2^{j+1})}\right)^\dagger\Big|_{2\times 2}^{(n)} = \begin{pmatrix} I_{1,1}^{(n,M)} & I_{1,2}^{(n,M)} \\ I_{2,1}^{(n,M)} & I_{2,2}^{(n,M)} \end{pmatrix}, \quad (129)$$

which implies, using (126) and (128),

$$\langle\Omega|\prod_{j=0}^{M-1}\mathcal{I}_\nu^{(q_1^j q_2^j)}\left(\mathcal{I}_\nu^{(q_1^j q_2^{j+1})}\right)^\dagger|\Omega\rangle = \prod_{n>0}\frac{1}{I_{1,1}^{(n,M)}}. \quad (130)$$

From (127) and (129), we obtain the following recursion relation:

$$\begin{pmatrix} I_{1,1}^{(n,M)} & I_{1,2}^{(n,M)} \\ I_{2,1}^{(n,M)} & I_{2,2}^{(n,M)} \end{pmatrix} = \begin{pmatrix} I_{1,1}^{(n,M-1)} & I_{1,2}^{(n,M-1)} \\ I_{2,1}^{(n,M-1)} & I_{2,2}^{(n,M-1)} \end{pmatrix}$$
$$\times \begin{pmatrix} \cosh^2(\nu) - \sinh^2(\nu)q_2^{2n} & \frac{1}{2}\sinh(2\nu)\left(1 - q_2^{-2n}\right)q_2^{-2(M-1)n}q_1^{-2(M-1)n} \\ \frac{1}{2}\sinh(2\nu)\left(1 - q_2^{2n}\right)q_2^{2(M-1)n}q_1^{2(M-1)n} & \cosh^2(\nu) - \sinh^2(\nu)q_2^{-2n} \end{pmatrix}. \quad (131)$$

This can be solved for $I_{1,1}^{(n,M)}$, see Appendix B.2. The result is

$$I_{1,1}^{(n,M)} = (q_2^n q_1^n)^M \left| \frac{(1-\varepsilon_n)(\lambda_n^-)^M + (1+\varepsilon_n)(\lambda_n^+)^M}{2} \right|^2, \quad (132)$$

with $\lambda_n^\pm$ in (123) and

$$\varepsilon_n = -\frac{2\left[\sin(2\pi n\tau_1)\cos(2\pi n\tau_2) + \cos(2\pi n\tau_1)\sin(2\pi n\tau_2)\cosh(2\nu)\right]}{\sqrt{4 - \sigma_n^2}}, \quad (133)$$

using $\sigma_n$ in (122). In conclusion, the Loschmidt echo after $M$ cycles for the ground state is

$$L_\Omega(M[t_1 + t_2]) = \prod_{n>0} L_\Omega^{(n)}(M[t_1 + t_2]), \quad (134)$$

$$L_\Omega^{(n)}(M[t_1 + t_2]) = \left| \frac{2}{(1-\varepsilon_n)(\lambda_n^-)^M + (1+\varepsilon_n)(\lambda_n^+)^M} \right|^2,$$

with $\lambda_n^\pm$ in (123) and $\varepsilon_n$ in (133).

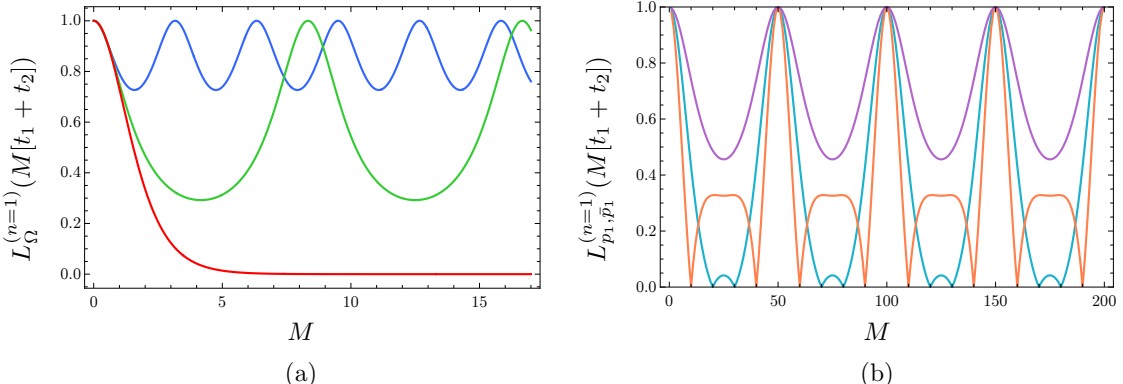

(a)                                                        (b)

Figure 10: (a) Stroboscopic time evolution of the single-mode Loschmidt echo $L_\Omega^{(n=1)}(M[t_1 + t_2])$ in (134) for the ground state $|\Omega\rangle$ of $H_1$ in a two-step drive with $K_1/K_2 = 4/3$ and different driving parameters $\tau_1 = \tau_2 = 8/50, 9.8/50, 12/50$ (blue, green red). In the stable phase, the Loschmidt echo displays periodic revivals with a period that depends on the driving parameters. In the unstable phase, the Loschmidt echo decays exponentially to zero. (b) Stroboscopic time evolution of the single-mode Loschmidt echo $L_{p_1,\bar{p}_1}^{(n=1)}(M[t_1 + t_2])$ in (152) for different initial states of the form $a_{-1}^{p_1}\bar{a}_{-1}^3|\Omega\rangle$ with $p_1 = 0, 1, 3$ (purple, cyan, orange) in a two-step drive with $K_1/K_2 = 4/3$ and driving parameters $(\tau_1, \tau_2) = (0.5, 0.51)$. The periodicity of the Loschmidt echo does not depend on the initial state, but temporal orthogonality can be observed for more general initial states.

**Single-mode analysis and order parameters**

We now restrict our analysis to the ground-state Loschmidt echo $L_\Omega^{(n)}(M[t_1 + t_2])$ for a single mode $n$. Its behavior as a function of the stroboscopic time $M(t_1 + t_2)$ depends crucially on the driving parameters $(\tau_1, \tau_2)$ and the Zamolodchikov distance $\nu$ between $H_1$ and $H_2$ due to the different properties of $\lambda_n^\pm$ in (123). Indeed, if $\sigma_n < 4$, we recall that $\lambda_n^\pm = \text{sgn}(\omega_n)e^{\pm i\phi}$ for $\phi \in (0, \pi/2]$, which leads to an overall oscillation with $M$ of the form

$$L_\Omega^{(n)}(M[t_1 + t_2]) = \left| \frac{2}{(1 - \varepsilon_n)e^{-iM\phi} - (1 + \varepsilon_n)e^{iM\phi}} \right|^2. \tag{135}$$

On the other hand, if $\sigma_n > 4$, we recall that $|\lambda_n^+|$ is larger than one, which implies that $L_\Omega^{(n)}(M[t_1 + t_2])$ decays exponentially,

$$L_\Omega^{(n)}(M[t_1 + t_2]) \sim e^{-\lambda_L M(t_1 + t_2)}, \tag{136}$$

with the rate

$$\lambda_L = \frac{\log|\lambda_n^+|}{t_1 + t_2}. \tag{137}$$

These two distinct dynamical behaviors of the single-mode Loschmidt echo can be observed in Fig. 10(a) and compared with the stability characterizations in Table 1.

We stress that the Loschmidt echo $L_\Omega(M[t_1 + t_2])$ is a product over all possible modes $n$, such that it would generically decay exponentially in time when all modes are taken into account. However, as discussed previously, one usually needs to impose a cutoff on the number of modes in order to connect with physical applications. Depending on the value of such a cutoff and the value of the Zamolodchikov distance, $\nu = \log\sqrt{K_1/K_2}$, some extended regions

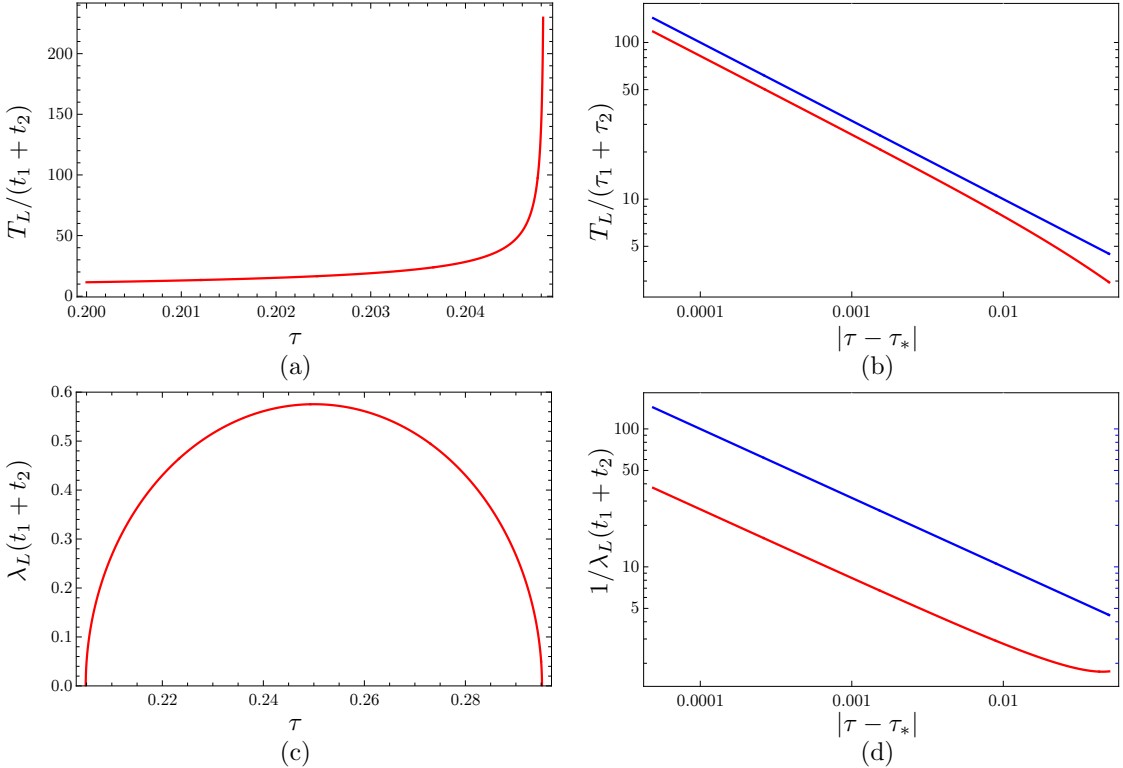

Figure 11: Critical properties of the order parameters in a two-step drive with $K_1/K_2 = 4/3$ when approaching the phase boundary from the stable or the unstable phase for a single mode $n = 1$. (a) The period $T_L/(t_1 + t_2)$ as a function of $\tau = \tau_1 = \tau_2$ when approaching the phase boundary at $\tau_* \approx 0.2048$ from the stable phase. (b) Red curve: $T_L/(t_1 + t_2)$ as a function of $|\tau - \tau_*|$. Blue curve: Expected scaling as $|\tau - \tau_*|^{-1/2}$. (c) The rate $(t_1 + t_2)\lambda_L$ as a function of $\tau = \tau_1 = \tau_2$ in the unstable phase. (d) Red curve: $1/\lambda_L(t_1 + t_2)$ as a function of $|\tau - \tau_*|$ when approaching the phase boundary at $\tau_* \approx 0.2048$ from the unstable phase. Blue curve: Expected scaling as $|\tau - \tau_*|^{-1/2}$.

in the parameter space $(\tau_1, \tau_2)$ may still be stable, leading to non-trivial phase diagrams with phase transitions between oscillating and exponentially decaying Loschmidt echo.

As a final consideration, we study the behavior of the Loschmidt echo for a single mode when approaching the phase boundary from the stable phase. As can be observed in Fig. 10(a), the period $T_L$ of the Loschmidt echo in the stable phase increases as we approach the boundary and can be interpreted as a natural order parameter. We can explicitly write the period as

$$T_L = \frac{\pi(t_1 + t_2)}{\arg(\lambda_n^+)} \,. \tag{138}$$

As we approach the phase boundary, say by varying $\tau = \tau_1 = \tau_2$,[15] the period diverges as a power law

$$T_L \sim |\tau - \tau_*|^{-\beta_T} \,, \tag{139}$$

where $\tau_*$ lies on the phase boundary and $\beta_T$ is the critical exponent at the transition, as seen in Fig. 11(a). By fitting the critical exponent, we infer that $\beta_T = 1/2$, see Fig. 11(b). Alternatively, the phase boundary can be approached from the unstable phase, in which case

---

[15]Note that $\tau$ here should not be confused with a modular parameter, such as the one in (111).

the natural order parameter is the rate $\lambda_L$, which is understood formally as the inverse of $T_L$ by comparing (138) and (137). Close to the phase boundary, see Fig. 11(c), $\lambda_L$ approaches zero as

$$\lambda_L \sim |\tau - \tau_*|^{\beta_\lambda}, \tag{140}$$

with the critical exponent $\beta_\lambda = 1/2$ inferred from Fig. 11(d). In conclusion, the rate $\lambda_L$ and the inverse period $1/T_L$ for the Loschmidt echo are natural order parameters for the stable-to-unstable transition in the respective phases and have the same critical exponent $\beta_T = \beta_\lambda = 1/2$ when approaching the phase boundary from each side. We note that such a critical scaling for the period in the stable phase and for the rate in the unstable phase has been observed in other classes of integrable Floquet systems [40].

**For excited states**

We now study the stroboscopic time evolution of the Loschmidt echo for general excited states of the form in (65).

First, we consider the state $\left(1/\sqrt{n^p p!}\right)a_{-n}^p|\Omega\rangle$, as before writing $p = p_n$ to lighten the notation. The Loschmidt echo after $M$ cycles is

$$L_p(M[t_1 + t_2]) = \frac{1}{\mathcal{N}_p^2}\left|\langle\Omega|a_n^p U_F^M a_{-n}^p|\Omega\rangle\right|^2, \qquad \mathcal{N}_p = n^p p!. \tag{141}$$

Analogous to Sec. 4.1 for the quench, the computation reduces to evaluating

$$C_p^{(M)} = \langle\Omega|a_n^p \prod_{j=0}^{M-1} \mathcal{I}_\nu^{(q_1^j q_2^j)}\left(\mathcal{I}_\nu^{(q_1^j q_2^{j+1})}\right)^\dagger a_{-n}^p|\Omega\rangle. \tag{142}$$

To this end, we consider the following generalized rotation relations:

$$\begin{aligned}
\mathcal{I}_\nu^{(q_1^j q_2^j)}\left(\mathcal{I}_\nu^{(q_1^j q_2^{j+1})}\right)^\dagger a_{-n}\mathcal{I}_\nu^{(q_1^j q_2^{j+1})}\left(\mathcal{I}_\nu^{(q_1^j q_2^j)}\right)^\dagger &= A_n a_{-n} + (q_1 q_2)^{-2nj} B_n \bar{a}_n, \\
\mathcal{I}_\nu^{(q_1^j q_2^j)}\left(\mathcal{I}_\nu^{(q_1^j q_2^{j+1})}\right)^\dagger \bar{a}_n\mathcal{I}_\nu^{(q_1^j q_2^{j+1})}\left(\mathcal{I}_\nu^{(q_1^j q_2^j)}\right)^\dagger &= \overline{A_n}\bar{a}_n + (q_1 q_2)^{2nj}\overline{B_n} a_{-n},
\end{aligned} \tag{143}$$

with $A_n = A_n(q_2)$, $\overline{A_n} = A_n(\overline{q_2})$, $B_n = B_n(q_2)$, and $\overline{B_n} = B_n(\overline{q_2})$ given by

$$A_n(q_2) = \cosh^2(\nu) - \sinh^2(\nu)q_2^{-2n}, \qquad B_n(q_2) = \frac{1}{2}\sinh(2\nu)(1 - q_2^{-2n}). \tag{144}$$

Written in matrix form, this amounts to an $SU(1,1)$ rotation:

$$\mathsf{T}_j\begin{pmatrix} a_{-n} \\ \bar{a}_n \end{pmatrix} = \begin{pmatrix} A_n & (q_1 q_2)^{-2nj}B_n \\ (q_1 q_2)^{2nj}\overline{B_n} & \overline{A_n} \end{pmatrix}\begin{pmatrix} a_{-n} \\ \bar{a}_n \end{pmatrix}. \tag{145}$$

The rotation in the other direction is given by

$$\mathsf{T}_j^{-1}\begin{pmatrix} a_{-n} \\ \bar{a}_n \end{pmatrix} = \begin{pmatrix} \overline{A_n} & -(q_1 q_2)^{-2nj}B_n \\ -(q_1 q_2)^{2nj}\overline{B_n} & A_n \end{pmatrix}\begin{pmatrix} a_{-n} \\ \bar{a}_n \end{pmatrix}. \tag{146}$$

For $M$ cycles, we need to apply these rotations $M$ times. The matrices implementing these rotations can be written

$$\prod_{j=M-1}^{0} \mathsf{T}_j = \begin{pmatrix} \mathcal{A}_n & \mathcal{B}_n \\ \overline{\mathcal{B}_n} & \overline{\mathcal{A}_n} \end{pmatrix}, \qquad \prod_{j=0}^{M-1} \mathsf{T}_j^{-1} = \begin{pmatrix} \overline{\mathcal{A}_n} & -\mathcal{B}_n \\ -\overline{\mathcal{B}_n} & \mathcal{A}_n \end{pmatrix}, \tag{147}$$

which define $\mathcal{A}_n = \mathcal{A}_n(M)$ and $\mathcal{B}_n = \mathcal{B}_n(M)$ as functions of $M$, see Appendix B.3 for how to obtain analytical expressions for the latter.

It follows from the above that

$$C_p^{(M)} = np\mathcal{A}_n\langle\Omega|a_n^{p-1}\prod_{j=0}^{M-1}\mathcal{I}_\nu^{(q_1^j q_2^j)}\left(\mathcal{I}_\nu^{(q_1^j q_2^{j+1})}\right)^\dagger a_{-n}^{p-1}|\Omega\rangle + \mathcal{B}_n\langle\Omega|a_n^p\bar{a}_n\prod_{j=0}^{M-1}\mathcal{I}_\nu^{(q_1^j q_2^j)}\left(\mathcal{I}_\nu^{(q_1^j q_2^{j+1})}\right)^\dagger a_{-n}^{p-1}|\Omega\rangle. \quad (148)$$

By moving $\bar{a}_n$ from the left to the right in the second term, we obtain the recursion relation

$$C_p^{(M)} = np\frac{\mathcal{A}_n}{1+|\mathcal{B}_n|^2}C_{p-1}^{(M)}, \qquad C_0^{(M)} = \langle\Omega|\prod_{j=0}^{M-1}\mathcal{I}_\nu^{(q_1^j q_2^j)}\left(\mathcal{I}_\nu^{(q_1^j q_2^{j+1})}\right)^\dagger|\Omega\rangle. \quad (149)$$

This has the solution

$$C_p^{(M)} = n^p p!\left(\frac{\mathcal{A}_n}{1+|\mathcal{B}_n|^2}\right)^p\langle\Omega|\prod_{j=0}^{M-1}\mathcal{I}_\nu^{(q_1^j q_2^j)}\left(\mathcal{I}_\nu^{(q_1^j q_2^{j+1})}\right)^\dagger|\Omega\rangle. \quad (150)$$

We conclude that the Loschmidt echo after $M$ cycles for the state $\left(1/\sqrt{n^{p_n}p_n!}\right)a_{-n}^{P_n}|\Omega\rangle$ is

$$L_{p_n}(M[t_1+t_2]) = \left(\frac{|\mathcal{A}_n(M)|}{1+|\mathcal{B}_n(M)|^2}\right)^{2p}L_\Omega(M[t_1+t_2]), \quad (151)$$

with $\mathcal{A}_n(M)$ and $\mathcal{B}_n(M)$ given by (144)–(147). The result in (151) is a generalization of the quench result in (75) to our Floquet drive. The Loschmidt echo after a quantum quench can be obtained as a special case by setting $q_1 = 1$, $q_2 = q = \mathrm{e}^{-2\pi i v_2 t/L}$, and $M = 1$, for which $\mathcal{A}_n(M=1) = A_n(t)$ and $\mathcal{B}_n(M=1) = B_n(t)$ given by (70).

Finally, following the derivation in Sec. 4.1 for excited states, we can write down the Loschmidt echo for the most general excited state of the form in (65). The result is

$$L_{\boldsymbol{p},\bar{\boldsymbol{p}}}(M[t_1+t_2]) = \frac{1}{\mathcal{N}_{\boldsymbol{p},\bar{\boldsymbol{p}}}^2}\left|\langle\Omega|\prod_{n=1}^\infty a_n^{p_n}\bar{a}_n^{\bar{p}_n}U_F^M\prod_{n=1}^\infty \bar{a}_{-n}^{\bar{p}_n}a_{-n}^{P_n}|\Omega\rangle\right|^2$$

$$= \prod_{n=1}^\infty L_{p_n,\bar{p}_n}^{(n)}(M[t_1+t_2]), \quad (152)$$

$$L_{p_n,\bar{p}_n}^{(n)}(M[t_1+t_2]), = L_\Omega^{(n)}(M[t_1+t_2])\left(\frac{|\mathcal{A}_n(M)|}{1+|\mathcal{B}_n(M)|^2}\right)^{2(p_n+\bar{p}_n)}\left|{}_2F_1(-p_n,-\bar{p}_n;1;-|\mathcal{B}_n(M)|^2)\right|^2,$$

with $\mathcal{A}_n(M)$ and $\mathcal{B}_n(M)$ given by (144)–(147). As was the case for (151), the result in (152) generalizes that in (85), which can be seen as a special case with $q_1 = 1$, $q_2 = q = \mathrm{e}^{-2\pi i v_2 t/L}$, and $M = 1$. It thus provides the most general form of the stroboscopic time evolution of the Loschmidt echo under a periodic drive starting from any eigenstate of the theory with $K = K_1$. As shown in Fig. 10(b), the periodicity of the Loschmidt echo is independent of the choice of initial state, and the discussion of the critical exponents of $T_L$ and $\lambda_L$ across the transition is thus unchanged. As already discussed in the quench case, by considering general initial states that mix right- and left-moving excitations, the stroboscopic Loschmidt echo can display non-analytic behavior. However, we note that the Loschmidt echo is now evaluated at discrete times $M(t_1+t_2)$, and thus the zeros in the return probability are only approximate, showing a pseudo-orthogonality at stroboscopic times.

## 5.2 Particle density

In TLL theory, the total particle density is

$$\rho(x) = J_+(x) + J_-(x), \quad (153)$$

which is expressible in terms of $a_n$ and $\bar{a}_n$ using (26) and (34). Thus, to study the Floquet time evolution of the particle density, it suffices to consider the evolution of the oscillator modes. It should be noted that (32) implies conservation of the particle-number charges $J_0 = \sqrt{K_1} a_0$ and $\bar{J}_0 = \sqrt{K_1} \bar{a}_0$, since they commute with all modes and thus with $H_1$ and $H_2$. However, in general, at the level of operators, the chiral densities $J_\pm(x)$ evolve non-trivially, unless evaluated with respect to a spatially homogeneous state, in which case trivially, since only the zero modes $J_0$ and $\bar{J}_0$ contribute.

To this end, consider the evolution of the operators $a_n$ and $\bar{a}_{-n}$ under one full cycle of the Floquet drive. (The change of sign in the subscript for the latter is for convenience, due to the way our drive mixes the modes.) From (49), (51), and (53), it follows that

$$U_F^{-1} \begin{pmatrix} a_n \\ \bar{a}_{-n} \end{pmatrix} U_F = \mathsf{C}(n) \begin{pmatrix} a_n \\ \bar{a}_{-n} \end{pmatrix}, \tag{154}$$

with the $2 \times 2$ matrix

$$\mathsf{C}(n) = \begin{pmatrix} [q_2^n \cosh^2(v) - q_2^{-n} \sinh^2(v)]q_1^n & (q_2^n - q_2^{-n})\cosh(v)\sinh(v)q_1^n \\ (q_2^{-n} - q_2^n)\cosh(v)\sinh(v)q_1^{-n} & [q_2^{-n}\cosh^2(v) - q_2^n \sinh^2(v)]q_1^{-n} \end{pmatrix}, \tag{155}$$

using $q_1$ and $q_2$ in (4). This matrix lies in SU(1, 1) and is non-trivial unless $n = 0$, in which case it is the identity matrix, consistent with particle-number conservation. It follows that the result after $M$ cycles is obtained by multiplication by $\mathsf{C}(n)^M$,

$$U_F^{-M} \begin{pmatrix} a_n \\ \bar{a}_{-n} \end{pmatrix} U_F^M = \mathsf{C}(n)^M \begin{pmatrix} a_n \\ \bar{a}_{-n} \end{pmatrix}, \tag{156}$$

meaning that all information can be obtained by studying the properties of $\mathsf{C}(n)$.

One can show that the eigenvalues of $\mathsf{C}(n)$ are precisely $\lambda_n^\pm = \omega_n \pm \mathrm{sgn}(\omega_n)\sqrt{\omega_n^2 - 1}$ in (123) in terms of $\omega_n$ in (122). As direct consequences,

$$\det[\mathsf{C}(n)] = \lambda_n^+ \lambda_n^- = 1, \qquad \mathrm{tr}[\mathsf{C}(n)] = \lambda_n^+ + \lambda_n^- = 2\omega_n = \mathrm{Tr}\left[ U_F^{(n)} \right], \tag{157}$$

for $U_F^{(n)} = e^{-iH_F^{(n)}(t_1 + t_2)}$ with $H_F^{(n)}$ in (120). More importantly, the effect of the $M$-cycle drive in (156) enters precisely through factors of the form $(\lambda_n^\pm)^M$. Thus, following the discussion below (123), the stability characterizations in Table 1 are directly observable in the particle density $\rho(x)$. Indeed, the $n$th-mode contribution exhibits parametric instability if $\sigma_n = 4\omega_n^2 > 4$, since this implies exponential growth in discrete time $M(t_1 + t_2)$ with the same rate $\lambda_L = (t_1 + t_2)^{-1}\log|\lambda_n^+|$ as in (137). Similarly, if $\sigma_n < 4$, one can deduce that it features oscillations with the same period $T_L = \pi(t_1 + t_2)/\arg(\lambda_n^+)$ as in (138).

A related quantity of interest are density fluctuations in $\rho(x)$, or phrased differently, density-density correlations in the form of expectations of $\rho(x_1)\rho(x_2)$. Again, it follows from (26) and (34) that the relevant objects in Fourier space are the bilinears $a_n a_m$, $a_n \bar{a}_{-m}$, $\bar{a}_{-n} a_m$, and $\bar{a}_{-n}\bar{a}_{-m}$. From (154) and elementary linear algebra, the evolution of these under one full cycle is given by the Kronecker product $\mathsf{C}(n) \otimes \mathsf{C}(m)$, whose eigenvalues are

$$\lambda_{n,m}^1 = \lambda_n^+ \lambda_m^+, \qquad \lambda_{n,m}^2 = \lambda_n^+ \lambda_m^-, \qquad \lambda_{n,m}^3 = \lambda_n^- \lambda_m^+, \qquad \lambda_{n,m}^4 = \lambda_n^- \lambda_m^-, \tag{158}$$

with $\lambda_n^\pm$ in (123). As before, it follows that

$$\det[\mathsf{C}(n) \otimes \mathsf{C}(m)] = 1, \qquad \mathrm{tr}[\mathsf{C}(n) \otimes \mathsf{C}(m)] = 4\omega_n \omega_m = \mathrm{Tr}\left[ U_F^{(n)} \right]\mathrm{Tr}\left[ U_F^{(m)} \right]. \tag{159}$$

Crucially, the discussion on stability involving $\lambda_n^\pm$ translates directly to $\lambda_{n,m}^j$ for $j = 1, 2, 3, 4$. The above results for $\mathsf{C}(n) \otimes \mathsf{C}(m)$ are particularly important when considering expectations

with respect to homogeneous states, since not only zero modes but also expectations of $a_n a_{-n}$ and $\bar{a}_{-n}\bar{a}_n$ ($n \neq 0$) then contribute, meaning that the eigenvalues $\lambda^1_{n,n} = (\lambda^+_n)^2$ and $\lambda^4_{n,n} = (\lambda^-_n)^2$ for $n \neq 0$ are always relevant to study. This implies that the stability characterizations in Table 1, including the exponential growth with $M$ for $\sigma_n > 4$ indicating parametric instability, are always observable in density-density correlations, which were the objects considered in [46, 47] as probes of instabilities.

## 5.3 Energy density

As seen earlier in Sec. 4.2, the energy density can be expressed as

$$\mathcal{E}(x) = v_1\big[T_+(x) + T_-(x)\big] \tag{160}$$

at an operator level. Using (26) and (35), it follows that the relevant objects to study in order to understand its Floquet time evolution are $:a_{n-m}a_m:$ and $:\bar{a}_{-n+m}\bar{a}_{-m}:$, cf. Sec. 4.2. (Again, the change of signs in the subscripts for the latter is for convenience.) However, as seen for particle-density fluctuations in Sec. 5.2, under individual cycles, our drive generates contributions of the form $a_{n-m}\bar{a}_{-m}$ and $\bar{a}_{-n+m}a_m$. It is thus necessary to consider all four of these bilinears. By the same argument as before, it follows from (154) that

$$U_F^{-1}\begin{pmatrix} :a_{n-m}a_m: \\ a_{n-m}\bar{a}_{-m} \\ \bar{a}_{-n+m}a_m \\ :\bar{a}_{-n+m}\bar{a}_{-m}: \end{pmatrix}U_F = \mathsf{A}(n,m)\begin{pmatrix} :a_{n-m}a_m: \\ a_{n-m}\bar{a}_{-m} \\ \bar{a}_{-n+m}a_m \\ :\bar{a}_{-n+m}\bar{a}_{-m}: \end{pmatrix} + \delta_{n,0}|m|\mathsf{B}(m), \tag{161}$$

with the $4 \times 4$ matrix

$$\mathsf{A}(n,m) = \mathsf{C}(n-m) \otimes \mathsf{C}(m) \tag{162}$$

obtained as the Kronecker product of two $2 \times 2$ matrices of the form in (155) and the vector

$$\mathsf{B}(m) = \begin{pmatrix} (q_2^{-m} - q_2^m)(q_2^m - q_2^{-m})\cosh^2(v)\sinh^2(v) \\ [q_2^{-m}\cosh^2(v) - q_2^m\sinh^2(v)](q_2^{-m} - q_2^m)\cosh(v)\sinh(v)q_1^{-2m} \\ [q_2^m\cosh^2(v) - q_2^{-m}\sinh^2(v)](q_2^m - q_2^{-m})\cosh(v)\sinh(v)q_1^{2m} \\ (q_2^{-m} - q_2^m)(q_2^m - q_2^{-m})\cosh^2(v)\sinh^2(v) \end{pmatrix}. \tag{163}$$

We note that the presence of $\mathsf{B}(m)$ is due to re-ordering of the right-hand side using (56).

As before, the result of our Floquet drive after $M$ cycles can be understood from the properties of the matrix in (162). More precisely,

$$U_F^{-M}\begin{pmatrix} :a_{n-m}a_m: \\ a_{n-m}\bar{a}_{-m} \\ \bar{a}_{-n+m}a_m \\ :\bar{a}_{-n+m}\bar{a}_{-m}: \end{pmatrix}U_F^M = \mathsf{A}(n,m)^M\begin{pmatrix} :a_{n-m}a_m: \\ a_{n-m}\bar{a}_{-m} \\ \bar{a}_{-n+m}a_m \\ :\bar{a}_{-n+m}\bar{a}_{-m}: \end{pmatrix} + \delta_{n,0}|m|\sum_{j=0}^{M-1}[\mathsf{A}(0,m)]^j\mathsf{B}(m). \tag{164}$$

Note that the second term still contributes even if the above expression is evaluated with respect to the ground state $|\Omega\rangle$ of the theory with $K = K_1$. Indeed,

$$\langle\Omega|U_F^{-M}L_nU_F^M|\Omega\rangle = \delta_{n,0}\langle\Omega|U_F^{-M}L_0U_F^M|\Omega\rangle = \delta_{n,0}\sum_{m=-\infty}^{\infty}\frac{|m|}{2}\begin{pmatrix}1 & 0 & 0 & 0\end{pmatrix}\sum_{j=0}^{M-1}[\mathsf{A}(0,m)]^j\mathsf{B}(m),$$

$$\langle\Omega|U_F^{-M}\bar{L}_nU_F^M|\Omega\rangle = \delta_{n,0}\langle\Omega|U_F^{-M}\bar{L}_0U_F^M|\Omega\rangle = \delta_{n,0}\sum_{m=-\infty}^{\infty}\frac{|m|}{2}\begin{pmatrix}0 & 0 & 0 & 1\end{pmatrix}\sum_{j=0}^{M-1}[\mathsf{A}(0,m)]^j\mathsf{B}(m), \tag{165}$$

where the vectors $\begin{pmatrix} 1 & 0 & 0 & 0 \end{pmatrix}$ and $\begin{pmatrix} 0 & 0 & 0 & 1 \end{pmatrix}$ were inserted to project the results to give the energies of right- and left-moving excitations. We recall the need to renormalize the above expressions, as discussed in Sec. 4.2, unless an ultraviolet cutoff is imposed on the interaction modulation.

From the above, it is clear that we are interested in the eigenvalues $\lambda^j_{n-m,m}$ of $A(n,m)$ in (162) as well as those of $\sum_{j=0}^{M-1}[A(0,m)]^j$. As before, using standard properties for Kronecker products, $\lambda^j_{n-m,m}$ are given by (158) and thus directly obtained from the eigenvalues of $C(n)$. Setting $n=0$, the eigenvalues of $A(0,m)^M$ are

$$(\lambda^1_{-m,m})^M = (\lambda^+_m)^{2M}, \qquad (\lambda^{2,3}_{-m,m})^M = 1, \qquad (\lambda^4_{-m,m})^M = (\lambda^-_m)^{2M}, \qquad (166)$$

which also implies that the eigenvalues of $\sum_{j=0}^{M-1}[A(0,m)]^j$ are[16]

$$\sum_{j=0}^{M-1}(\lambda^1_{-m,m})^j = \frac{1-(\lambda^+_m)^{2M}}{1-(\lambda^+_m)^2}, \qquad \sum_{j=0}^{M-1}(\lambda^{2,3}_{-m,m})^j = M, \qquad \sum_{j=0}^{M-1}(\lambda^4_{-m,m})^j = \frac{1-(\lambda^-_m)^{2M}}{1-(\lambda^-_m)^2}, \quad (167)$$

in terms of $\lambda^\pm_m$ in (123). Again, this results in the same stability characterizations depending on $\sigma_m$ as described in the beginning of this section, see Table 1 and Fig. 8: If $\sigma_m > 4$, there are parametric instabilities observable in the $m$th-mode contribution to (the $L_0$- and $\bar{L}_0$-parts of) the energy density in the form of exponential growth with the rate in (137), while if $\sigma_m < 4$, there are oscillations with the period in (138).

# 6 Rényi divergence and relative entropy

In this section we turn to a Euclidean setup. We consider a measure from quantum information theory, the so-called Rényi divergence, which quantifies the difference between thermal states of the undeformed Hamiltonian $H_1$ and the deformed Hamiltonian $H_2$. It is defined as the one-parameter generalization of the relative entropy, in the same way that Rényi entropy is the one-parameter generalization of von Neumann entropy. For any two normalized density matrices $\hat{\rho}_1$ and $\hat{\rho}_2$, the Rényi divergence $D_\alpha(\hat{\rho}_1||\hat{\rho}_2)$ is defined as [50]

$$D_\alpha(\hat{\rho}_1||\hat{\rho}_2) = \frac{1}{\alpha-1}\log\text{Tr}\left[\hat{\rho}_1^\alpha\hat{\rho}_2^{1-\alpha}\right]. \qquad (168)$$

The quantity $D_\alpha = D_\alpha(\hat{\rho}_1||\hat{\rho}_2)$ possesses several mathematical properties: (i) it is positive, $D_\alpha \geq 0$, (ii) monotonic, $D_{\alpha_1} \geq D_{\alpha_2}$ if $\alpha_1 > \alpha_2$, (iii) continuous, and (iv) $(1-\alpha)D_\alpha$ is concave in $\alpha$. Furthermore, the limit $\alpha \to 1$ enables us to recover the relative entropy or Kullback-Leibler divergence

$$S(\hat{\rho}_1||\hat{\rho}_2) = \text{Tr}\left[\hat{\rho}_1\log\hat{\rho}_1\right] - \text{Tr}\left[\hat{\rho}_1\log\hat{\rho}_2\right], \qquad (169)$$

which defines a measure of the distance between two density matrices that is of importance in quantum information [51], holography [82,83], and CFT [84–86]. The concept of Rényi divergence recently attracted attention in the context of holography, where it was used to put additional constraints than the second law of thermodynamics using the monotonicity of $D_\alpha$ [54,55]. In this context, the Rényi divergence for two-dimensional CFTs was computed from Euclidean quenches in a path integral formalism, and its computation amounts to evaluating new classes of generalized partition functions of deformed theories.

The goal of this section is to derive expressions for the Rényi divergence and the relative entropy between a thermal state $\hat{\rho}_1 = e^{-\beta H_1}/Z_1(\beta)$ with $Z_1(\beta) = \text{Tr}\left[e^{-\beta H_1}\right]$ of the TLL or compactified free boson theory $H_1$ with Luttinger parameter $K_1$ or radius $R_1$ and a thermal state

---

[16]Clearly, all four eigenvalues in (167) are equal to $M$ if $\lambda^\pm_m = 1$.

$\hat{\rho}_2 = \mathrm{e}^{-\beta H_2}/Z_2(\beta)$ with $Z_2(\beta) = \mathrm{Tr}\big[\mathrm{e}^{-\beta H_2}\big]$ of the marginally deformed theory $H_2$ with Luttinger parameter $K_2$ or radius $R_2$. The Rényi divergence as a measure of the distance between two TLLs at finite temperature is defined in (168). As mentioned earlier, existing calculations of (168) in QFTs have been perturbative, namely, order-by-order in the deformation parameter $\mu$ if $S_2 = S_1 + \mu \int \mathrm{d}^2 x\, \mathcal{O}$ for actions $S_1$ and $S_2$, where $\mathcal{O}$ is some operator. Here we will demonstrate that the TLL or compactified free boson CFT offers an example to evaluate the object in (168) non-perturbatively as a function of $K_1$ and $K_2$, i.e., by taking $\mathcal{O}$ to be the marginal operator $\Phi \sim J\bar{J}$ in Sec. 3 [cf. (2), (29), and (40)].

For simplicity we set $v_1 = v_2 = 1$ throughout this section.

## 6.1 Rényi divergence

For convenience, we introduce $\tilde{\alpha} = 1 - \alpha$. The Rényi divergence in (168) for the thermal density matrices $\hat{\rho}_1$ and $\hat{\rho}_2$ then takes the form

$$D_\alpha(\hat{\rho}_1 || \hat{\rho}_2) = -\frac{1}{\tilde{\alpha}} \log\left(\frac{Z(\tilde{\alpha}, \beta)}{Z_2(\beta)^{\tilde{\alpha}} Z_1(\beta)^{1-\tilde{\alpha}}}\right), \tag{170}$$

where $Z_j(\beta) = \mathrm{Tr}\big[\mathrm{e}^{-\beta H_j}\big]$ for $j = 1, 2$ is the partition function of the TLL or compactified free boson Hamiltonian $H_j$ and $Z(\tilde{\alpha}, \beta) = \mathrm{Tr}\big[\mathrm{e}^{-\tilde{\alpha}\beta H_2}\mathrm{e}^{-(1-\tilde{\alpha})\beta H_1}\big]$. The former are given by [7]

$$Z_j(\beta) = \frac{1}{|\eta(\mathrm{i}\beta/L)|^2} \sum_{m,w \in \mathbb{Z}} \exp\left[-\pi\frac{\beta}{L}\left(\frac{m^2}{2K_j} + 2w^2 K_j\right)\right] = \frac{\Theta_j(\beta/L)}{\eta(\mathrm{i}\beta/L)^2}, \tag{171}$$

where $\eta(\cdot)$ denotes the Dedekind eta function[17] and $\Theta_j(\cdot)$ is the Siegel theta function with $j$ indicating the dependence on the Luttinger parameter $K_j$. Therefore, the crucial object to evaluate is the generalized partition function $Z(\tilde{\alpha}, \beta)$ in the numerator of the logarithm in (170). In terms of path integrals, this quantity is a Euclidean quench amplitude, see [54] for more details. The evolution in the (periodic) imaginary time direction is under $H_2$ for a duration $\tilde{\alpha}\beta$ and under $H_1$ for the remaining time $(1-\tilde{\alpha})\beta$. We can write this quantity as

$$Z(\tilde{\alpha}, \beta) = \mathrm{Tr}\left[\mathcal{I}_\nu(\mathcal{I}_\nu^{(q^{\tilde{\alpha}})})^\dagger q^{\left(L_0^{(\mathrm{osc})} + \bar{L}_0^{(\mathrm{osc})} + \tilde{\alpha}H_2^{(0)} + (1-\tilde{\alpha})H_1^{(0)}\right)}\right] q^{-1/12}, \tag{172}$$

for $q = \mathrm{e}^{-2\pi\beta/L}$, where we have used (55) along with the cyclicity of the trace.[18] The above trace can be conveniently factorized into contributions from the primaries and their descendants, analogous to the usual torus partition function of the compactified free boson CFT. The quantity above then takes the form

$$Z(\tilde{\alpha}, \beta) = \tilde{\Theta}(\beta/L)\Xi(\beta/L)\langle\Omega|\mathcal{I}_\nu(\mathcal{I}_\nu^{(q^{\tilde{\alpha}})})^\dagger|\Omega\rangle\mathrm{e}^{\pi\beta/6L}. \tag{173}$$

We now spell out the factors of the above expression in turn.

The contribution from the primary states is $\tilde{\Theta}(\beta/L)$, where we take into account zero modes from $H_1$ as well as $H_2$ in (39):

$$\tilde{\Theta}(\beta/L) = \sum_{m,w \in \mathbb{Z}} \exp\left[-\pi\frac{\tilde{\alpha}\beta}{L}\left(\frac{m^2}{2K_2^2} + 2K_2^2 w^2\right) - \pi\frac{(1-\tilde{\alpha})\beta}{L}\left(\frac{m^2}{2K_1^2} + 2K_1^2 w^2\right)\right]. \tag{174}$$

Meanwhile, the contribution $\Xi(\beta/L)$ from the descendant states is given by the following: We introduce

$$\Xi(z, \bar{z}) = \frac{\mathrm{Tr}_{\mathcal{V}_{m,w}}\left[\mathcal{I}_\nu(\mathcal{I}_\nu^{(q^{\tilde{\alpha}})})^\dagger z^{L_0^{(\mathrm{osc})}} \bar{z}^{\bar{L}_0^{(\mathrm{osc})}}\right]}{\langle\Omega|\mathcal{I}_\nu(\mathcal{I}_\nu^{(q^{\tilde{\alpha}})})^\dagger|\Omega\rangle} = \sum_{\boldsymbol{p},\bar{\boldsymbol{p}}} \frac{\langle\Psi_{\boldsymbol{p},\bar{\boldsymbol{p}}}|\mathcal{I}_\nu(\mathcal{I}_\nu^{(q^{\tilde{\alpha}})})^\dagger|\Psi_{\boldsymbol{p},\bar{\boldsymbol{p}}}\rangle}{\langle\Omega|\mathcal{I}_\nu(\mathcal{I}_\nu^{(q^{\tilde{\alpha}})})^\dagger|\Omega\rangle} z^{\sum n p_n} \bar{z}^{\sum n\bar{p}_n}, \tag{175}$$

---

[17]We recall that $\eta(\tau) = \mathrm{e}^{\mathrm{i}\pi\tau/12}\prod_{n=1}^{\infty}\big(1 - \mathrm{e}^{2\pi n\mathrm{i}\tau}\big)$ for complex $\tau$ satisfying $\mathrm{Im}(\tau) > 0$.

[18]The $E_2^0$ contribution is omitted since the computations conspire to cancel it for $D_\alpha(\hat{\rho}_1 || \hat{\rho}_2)$.

where the trace is over a single Verma module, $\mathcal{V}_{m,w}$, of a primary operator with momentum $m$ and winding number $w$. In the second equality, we have used the same notation for descendants (i.e., excited states) as in Sec. 4.1. The quantity above is the generating function for normalized and analytically continued return amplitudes for descendant states; we have $q^{\tilde{\alpha}} = \mathrm{e}^{-2\pi\tilde{\alpha}\beta/L}$ as opposed to $q = \mathrm{e}^{-2\pi\mathrm{i}t/L}$. This can be explicitly computed using (67) and (82), yielding

$$\Xi(z,\bar{z}) = \prod_{n=1}^{\infty} \sum_{p_n,\bar{p}_n=0}^{\infty} \left(\frac{A_n}{1+B_nB_{-n}}\right)^{p_n+\bar{p}_n} {}_2F_1(-p_n,-\bar{p}_n;1;-B_nB_{-n})z^{np_n}\bar{z}^{n\bar{p}_n}, \qquad (176)$$

where

$$A_n = \cosh^2(\nu) - \sinh^2(\nu)\mathrm{e}^{4\pi\tilde{\alpha}\beta n/L}, \qquad B_n = \frac{1}{2}\sinh(2\nu)\left(1-\mathrm{e}^{4\pi\tilde{\alpha}\beta n/L}\right). \qquad (177)$$

Using the definition of the hypergeometric function ${}_2F_1(a,b;c;z)$ as well as properties of the binomial coefficients, the generating function in (176) simplifies to the infinite product

$$\Xi(z,\bar{z}) = \prod_{n=1}^{\infty} \sum_{p_n,\bar{p}_n=0}^{\infty} \sum_{j=0}^{\infty} \binom{p_n}{j} \zeta_n^{p_n} \binom{\bar{p}_n}{j} \bar{\zeta}_n^{\bar{p}_n} \beta_n^j = \prod_{n=1}^{\infty} \frac{1}{(1-\zeta_n)(1-\bar{\zeta}_n) - \beta_n\zeta_n\bar{\zeta}_n}, \qquad (178)$$

where $\zeta_n = z^n A_n/(1-\beta_n)$ and $\beta_n = -B_nB_{-n}$. The standard undeformed generating function for the descendant states is thus recovered by setting $\nu = 0$, leading to $\prod_{n=1}^{\infty} \frac{1}{(1-z^n)(1-\bar{z}^n)}$. On the other hand, the analytically continued generating function $\Xi(\beta/L)$ for $z = \bar{z} = \mathrm{e}^{-2\pi\beta/L}$ in (178), which takes into account the contribution from the deformed descendant states, takes the form

$$\Xi(\beta/L) = \prod_{n=1}^{\infty} \frac{1}{(1-\zeta_n)(1-\bar{\zeta}_n) - \beta_n\zeta_n\bar{\zeta}_n}, \qquad (179)$$

where

$$\zeta_n = \bar{\zeta}_n = \frac{A_n}{1-\beta_n}\mathrm{e}^{-2\pi\beta n/L}, \qquad \beta_n = \sinh^2(2\nu)\sinh^2(2\pi\tilde{\alpha}\beta n/L), \qquad (180)$$

with $A_n$ in (177). Finally, we recall that the analytically continued ground-state return amplitude appearing in (173) is

$$\langle\Omega|\mathcal{I}_\nu(\mathcal{I}_\nu^{(q^{\tilde{\alpha}})})^\dagger|\Omega\rangle = \prod_{n=1}^{\infty} \frac{1}{\cosh^2(\nu) - \sinh^2(\nu)\mathrm{e}^{-4\pi\tilde{\alpha}\beta n/L}}. \qquad (181)$$

Putting everything together in (173), one can readily verify that (170) implies that the trivial limit of two identical TLLs ($\nu \to 0$) consistently yields $\lim_{\nu\to 0} D_\alpha(\hat{\rho}_1||\hat{\rho}_2) = 0$.

As a first step toward the evaluation of the Rényi divergence between two different TLLs, let us find the contribution from the zero modes to (170). To this end, we compute

$$\frac{\tilde{\Theta}(\beta/L)}{\Theta_2(\beta/L)^{\tilde{\alpha}}\Theta_1(\beta/L)^{1-\tilde{\alpha}}} = \frac{\vartheta_3\left(\mathrm{i}\pi\frac{\beta}{2L}\left[\frac{\tilde{\alpha}}{K_2}+\frac{1-\tilde{\alpha}}{K_1}\right]\right)\vartheta_3\left(\mathrm{i}\pi\frac{2\beta}{L}\left[\tilde{\alpha}K_2+(1-\tilde{\alpha})K_1\right]\right)}{\left[\vartheta_3\left(\mathrm{i}\pi\frac{\beta}{2L}\frac{1}{K_2}\right)\vartheta_3\left(\mathrm{i}\pi\frac{2\beta}{L}K_2\right)\right]^{\tilde{\alpha}}\left[\vartheta_3\left(\mathrm{i}\pi\frac{\beta}{2L}\frac{1}{K_1}\right)\vartheta_3\left(\mathrm{i}\pi\frac{2\beta}{L}K_1\right)\right]^{1-\tilde{\alpha}}}, \qquad (182)$$

where we used the Jacobi theta function $\vartheta_3(\cdot)$.[19] This allows us to use its modular properties to derive the high-temperature limit of the zero-mode contribution to the Rényi divergence: Using the S-modular transformation

$$\vartheta_3(\tau) = (-\mathrm{i}\tau)^{-1/2}\vartheta_3(-1/\tau), \qquad (183)$$

---

[19]We recall that $\vartheta_3(\tau) = \sum_{n\in\mathbb{Z}} \mathrm{e}^{\pi\mathrm{i}\tau n^2}$ for complex $\tau$ satisfying $\mathrm{Im}(\tau) > 0$.

we find that in the high-temperature regime $\beta/L \ll 1$, (182) simplifies to

$$\frac{\tilde{\Theta}(\beta/L)}{\Theta_2(\beta/L)^{\tilde{\alpha}}\Theta_1(\beta/L)^{1-\tilde{\alpha}}} \approx \left(\frac{\pi^2\frac{\beta^2}{L^2}\left[\frac{\tilde{\alpha}}{K_2}+\frac{1-\tilde{\alpha}}{K_1}\right]\left[\tilde{\alpha}K_2+(1-\tilde{\alpha})K_1\right]}{\pi^2\frac{\beta^2}{L^2}}\right)^{-1/2}$$

$$= \left(\cosh^2(\nu)-(1-2\tilde{\alpha})^2\sinh^2(\nu)\right)^{-1/2}, \tag{184}$$

which yields

$$D_\alpha^{(0)}(\hat{\rho}_1||\hat{\rho}_2) = -\frac{1}{\tilde{\alpha}}\log\left(\frac{\tilde{\Theta}(\beta/L)}{\Theta_2(\beta/L)^{\tilde{\alpha}}\Theta_1(\beta/L)^{1-\tilde{\alpha}}}\right) \approx \frac{\log\left(\cosh^2(\nu)-(1-2\tilde{\alpha})^2\sinh^2(\nu)\right)}{2\tilde{\alpha}}. \tag{185}$$

In particular, taking the limit $\alpha \to 1$, i.e., $\tilde{\alpha} \to 0$, gives the zero-mode contribution

$$S^{(0)}(\hat{\rho}_1||\hat{\rho}_2) \approx 2\sinh^2(\nu) \tag{186}$$

to the relative entropy. Since this contribution does not scale with temperature, it will be sub-leading and can thus be ignored in the high-temperature regime.

We now consider the contribution from the oscillator modes. Their contribution to $Z(\tilde{\alpha},\beta)e^{-\pi\beta/6L}$ is

$$\Xi(\beta/L)\langle\Omega|\mathcal{I}_\nu(\mathcal{I}_\nu^{(q^{\tilde{\alpha}})})^\dagger|\Omega\rangle = \prod_{n=1}^\infty \frac{1}{\left[(1-\zeta_n)^2-\beta_n\zeta_n^2\right]\left[\cosh^2(\nu)-\sinh^2(\nu)e^{-4\pi\tilde{\alpha}\beta n/L}\right]}. \tag{187}$$

For $\alpha = 1$, i.e., $\tilde{\alpha} = 0$, this yields

$$\Xi(\beta/L)\langle\Omega|\mathcal{I}_\nu(\mathcal{I}_\nu^{(q)})^\dagger|\Omega\rangle e^{\pi\beta/6L} = \frac{1}{\eta(i\beta/L)^2}, \tag{188}$$

where $q = e^{-2\pi\beta/L}$, in which case this cancels with the contribution from the oscillator modes to $Z_2(\beta)^{\tilde{\alpha}}Z_1(\beta)^{1-\tilde{\alpha}}$ in (170) for the Rényi divergence, cf. (171) and (173). It follows that

$$D_\alpha^{(\text{osc})}(\hat{\rho}_1||\hat{\rho}_2) = -\frac{1}{\tilde{\alpha}}\left[\log\left(\frac{Z(\tilde{\alpha},\beta)}{Z_2(\beta)^{\tilde{\alpha}}Z_1(\beta)^{1-\tilde{\alpha}}}\right) - \log\left(\frac{\tilde{\Theta}(\beta/L)}{\Theta_2(\beta/L)^{\tilde{\alpha}}\Theta_1(\beta/L)^{1-\tilde{\alpha}}}\right)\right]$$

$$= -\frac{1}{\tilde{\alpha}}\log\left(\eta(i\beta/L)^2\Xi(\beta/L)\langle\Omega|\mathcal{I}_\nu(\mathcal{I}_\nu^{(q^{\tilde{\alpha}})})^\dagger|\Omega\rangle e^{\pi\beta/6L}\right)$$

$$= \frac{1}{\tilde{\alpha}}\sum_{n=1}^\infty \log\left(\frac{\left[(1-\zeta_n)^2-\beta_n\zeta_n^2\right]\left[\cosh^2(\nu)-\sinh^2(\nu)e^{-4\pi\tilde{\alpha}\beta n/L}\right]}{\left[1-e^{-2\pi\beta n/L}\right]^2}\right), \tag{189}$$

with $\zeta_n$ and $\beta_n$ in (180), where we recall that $\tilde{\alpha} = 1-\alpha$. Similar to previous results in this paper, since individual terms in the sum in (189) tends to $\log\cosh^2(\nu)$ for large $n$, the sum must be renormalized unless an ultraviolet cutoff is imposed. In Fig. 12(a), we plot the result for the oscillator part of the Rényi divergence for a fixed cutoff on the number of modes. Its properties of positivity, monotonicity, and continuity are clearly visible in the figure. Furthermore, the concavity of $(1-\alpha)D_\alpha^{(\text{osc})}(\hat{\rho}_1||\hat{\rho}_2)$ is shown in Fig. 12(b). We stress that the formula in (189) for the Rényi divergence was obtained non-perturbatively.

As a last step, we take the limit $\alpha \to 1$ for $D_\alpha^{(\text{osc})}(\hat{\rho}_1||\hat{\rho}_2)$ in (189) to compute the oscillator part of the relative entropy between two TLLs. Formally taking the limit inside the sum and recalling that $\tilde{\alpha} = 1-\alpha$, one obtains

$$S^{(\text{osc})}(\hat{\rho}_1||\hat{\rho}_2) = \sum_{n=1}^\infty \frac{4\pi\beta n}{L}\sinh^2(\nu)\coth(\pi\beta n/L). \tag{190}$$

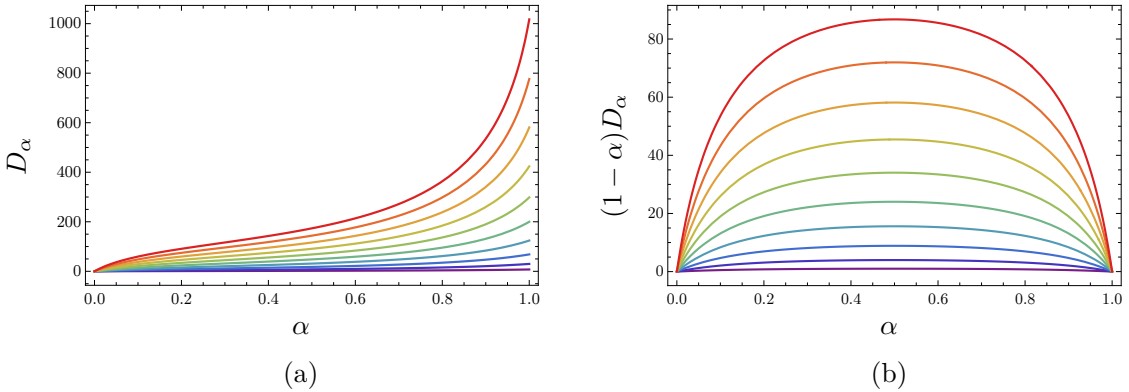

Figure 12: Plots of (a) the oscillator part of the Rényi divergence $D_\alpha = D_\alpha^{(\text{osc})}(\hat{\rho}_1||\hat{\rho}_2)$ in (189) and (b) the corresponding $(1-\alpha)D_\alpha$ as functions of $\alpha \in (0,1)$. The positivity, monotonicity, and continuity of $D_\alpha$ and the concavity of $(1-\alpha)D_\alpha$ are clearly visible. The parameters used in both (a) and (b) are $\beta/L = 0.01$ and $K_1/K_2 = e^{2\nu}$ for $\nu = 0.1, 0.2, ..., 1$ (bottom to top), and the results are plotted for a cutoff on the number of modes at $n = 100$.

This sum can be regularized (cf. Sec. 4.2) by writing it as

$$S^{(\text{osc})}(\hat{\rho}_1||\hat{\rho}_2) = \sum_{n=1}^{\infty} \frac{4\pi\beta n}{L} \sinh^2(\nu)\big[\coth(\pi\beta n/L) - 1\big] + \frac{4\pi\beta}{L} \sinh^2(\nu)\zeta(-1), \qquad (191)$$

where $\zeta(-1) = -1/12$ through analytic continuation. In the high-temperature regime $\beta/L \ll 1$, the sum in (191) can be approximated by an integral with respect to the dimensionless variable $\xi = \pi\beta n/L$ and computed analytically, yielding

$$S^{(\text{osc})}(\hat{\rho}_1||\hat{\rho}_2) \approx \frac{4L}{\pi\beta} \sinh^2(\nu) \int_0^{\infty} d\xi\, \xi\big[\coth(\xi) - 1\big] - \frac{\pi\beta}{3L}\sinh^2(\nu) = \frac{\pi L}{3\beta}\left(1 - \frac{\beta^2}{L^2}\right)\sinh^2(\nu). \quad (192)$$

Recalling that the zero-mode contribution in (186) is sub-leading in $L$, we conclude that the relative entropy between two TLLs with Luttinger parameters $K_1$ and $K_2$ is

$$S(\hat{\rho}_1||\hat{\rho}_2) \approx \frac{\pi L}{3\beta}\sinh^2(\nu) \qquad (193)$$

for large system sizes $L \gg 1$.

## 6.2 Relative entropy

As a consistency check of our results for the Rényi divergence and the formula in (193) for the relative entropy as its $\alpha \to 1$ limit, we now provide a direct calculation of the latter.

We start with the definition in (169). Since $\hat{\rho}_1$ and $\hat{\rho}_2$ are normalized thermal density matrices, we can rewrite the relative entropy as

$$S(\hat{\rho}_1||\hat{\rho}_2) = \beta\Big(\text{Tr}[\hat{\rho}_1 H_2] - \text{Tr}[\hat{\rho}_1 H_1]\Big) - \log\left(\frac{Z_2(\beta)}{Z_1(\beta)}\right). \qquad (194)$$

As before, we are interested in results for large system sizes. The last term vanishes due to the universality of high-temperature partition functions for CFTs: $Z(\beta) \approx \exp(\pi c L/6\beta)$ for

$\beta/L \ll 1$. The trace $\mathrm{Tr}[\hat{\rho}_1 H_1]$ appearing in (194) is simply the total energy of the undeformed theory at high temperatures. From (114), this is

$$\mathrm{Tr}[\hat{\rho}_1 H_1] = L\mathcal{E}_\beta \approx \frac{\pi L}{6\beta^2}. \tag{195}$$

We are then left to calculate the trace $\mathrm{Tr}[\hat{\rho}_1 H_2]$ at high temperatures. In order to proceed, we use the expression of the deformed Hamiltonian in (40) (omitting any constant terms subleading in $L$) together with (34) to obtain

$$\mathrm{Tr}[\hat{\rho}_1 H_2] = \cosh(2\nu)\,\mathrm{Tr}[\hat{\rho}_1 H_1] + \frac{2\pi\sinh(2\nu)}{L}\,\mathrm{Tr}\left[\hat{\rho}_1 \sum_{n=-\infty}^{\infty} a_n \bar{a}_n\right]$$
$$\approx \frac{\pi L}{6\beta^2}\cosh(2\nu) + \frac{2\pi\sinh(2\nu)}{L}\,\mathrm{Tr}[\hat{\rho}_1 a_0 \bar{a}_0]. \tag{196}$$

In the second step we used (195) and the fact that the contribution from the trace in the second term only comes from the zero modes. The last term vanishes in the thermodynamic limit; this is shown in Appendix B.4 using the flavoured partition function. Therefore, the relative entropy takes the form

$$S(\hat{\rho}_1 \| \hat{\rho}_2) \approx \frac{\pi L}{3\beta}\sinh^2(\nu) = \frac{\pi L}{3\beta}\frac{(K_1 - K_2)^2}{4K_1 K_2} \tag{197}$$

for $L \gg 1$, reproducing the result in (193). It obeys the general property of being non-negative and, as expected, gives zero when the Luttinger parameters (compactification radii) are equal. We stress that (197) yields a remarkably simple dependence on the Zamolodchikov distance $\nu$ and that it is an example of a relation between two different distance measures, namely a quantum information-theoretic distance and a geodesic distance in the space of theories.

# 7 Concluding remarks

In this paper, we studied the non-equilibrium dynamics of TLLs under interaction modulations modeled by quenching or periodically driving the Luttinger parameter. These modulations are marginal ($J\bar{J}$) deformations in the low-energy description of TLLs as compactified free bosons, which is the simplest CFT that belongs to a continuous family of CFTs. Two protocols were considered, a quantum quench and a two-step Floquet drive, switching between Hamiltonians $H_1$ and $H_2$ with different Luttinger parameters $K_1$ and $K_2$, or equivalently different compactification radii. Using Bogoliubov transformations and an underlying $\mathfrak{su}(1,1)$-algebraic structure, we derived a number of exact analytical results that depend crucially on the ratio of the Luttinger parameters, which corresponds to the Zamolodchikov distance $\nu = \log\sqrt{K_1/K_2}$ between the theories $H_1$ and $H_2$ in the space CFTs.

For the quench, we computed the Loschmidt echo and the time evolution of the energy density for the system initialized in any arbitrary eigenstate of $H_1$. We showed that the Loschmidt echo exhibits periodic revivals for all initial states, while if the initial state mixes right- and left-moving excitations, it also has Lee-Yang-Fisher zeros, which are defining features of dynamical quantum phase transitions. For the evolution of the energy-density expectation, we observed periodic discontinuities at times corresponding to the revivals in the Loschmidt echo. Moreover, starting from thermal states, its asymptotic (late-time) expression in the thermodynamic limit was shown to agree with that of the energy density evaluated in a thermal state at an effective temperature $\beta_{\mathrm{eff}}$ that depends on $\nu$.

For the two-step drive, we used a factorization of the Floquet operator into uncoupled discrete-time quantum parametric oscillators to obtain explicit criteria for stability or instability based on the value of the $\mathfrak{su}(1,1)$ Cartan-Killing form for the Floquet Hamiltonian for each individual mode. We showed that this is observable in physical quantities such as the stroboscopic time evolution of the Loschmidt echo for arbitrary eigenstates of $H_1$ and the particle and energy densities. In the stable phase, these quantities oscillate in time with a period that diverges as one approaches the phase boundary. On the other hand, in the unstable phase, the Loschmidt echo decays and the densities grow exponentially with a rate that vanishes as one approaches the phase boundary. This period and rate were identified as natural order parameters and shown to have critical exponents of $1/2$.

Lastly, we used our formalism to non-perturbatively compute the Rényi divergence between thermal states corresponding to the two Hamiltonians $H_1$ and $H_2$, while earlier QFT computations of the Rényi divergence have been perturbative. Taking a certain limit of our result, we obtained the relative entropy, which defines a quantum information-theoretic distance between density matrices, and which in our case has a remarkably simple dependence on $\nu$ in the thermodynamic limit. This relation between the relative entropy and the Zamolodchikov distance provides a concrete correspondence between two distance measures: It directly translates the geodesic distance in the moduli space to a quantum information-theoretic distance between thermal density matrices of the corresponding CFTs.

A common thread in all of our exact analytical results for TLLs is their dependence on the geometric distance in the space of theories related by marginal deformations. In this sense, the present work motivates further exploration of these connections between dynamics, quantum information-theoretic distance measures, and the geometry of moduli spaces.

There are several extensions of the present work that would be interesting to pursue:

**Quasi-periodic and random drives.** One direct extension is to consider drives that fully break time-translation invariance, either deterministically or randomly, in the form of quasi-periodic or random drives. The methods we used to compute, e.g., the Loschmidt echo for a periodic drive (see Sec. 5.1) are readily generalizable to these new drive protocols. One way is to use the generalized $SU(1,1)$ rotation relations and properties of products of random $SU(1,1)$ matrices and trace-map formulas for, e.g., Fibonacci quasi-periodic drive sequences, cf. [42, 45].

**Trapped ultra-cold atoms.** To relate to experiments, it would be interesting to generalize the constant Luttinger parameters $K_{1,2}$ in this work to functions $K_{1,2}(x)$ of position $x$. This arises naturally in cold-atom experiments as a consequence of the trapping potential [see (20) and (21)]. The dynamics in such a static environment was recently studied in [64], but the full quench problem or its driven counterpart have yet to be considered. To paint a complete picture, this would optimally also include a quantitative discussion of relevant length scales and effects of physical cutoffs on the number of modes. Another way to connect with experiments is to consider spatially inhomogeneous initial states, e.g., localized excitations on top of the ground state, which is realizable in cold-atom experiments. We expect that the way these excitations propagate under the periodic drive would lead to intricate spatial patterns of energy and particle density in both the stable and unstable phase.

**Driven dissipative TLLs** The solvability of the marginally driven TLL was greatly facilitated by identifying a closed $\mathfrak{su}(1,1)$ algebraic structure in the time-dependent Hamiltonian. A similar algebraic structure was identified and used to solve a dissipative harmonic oscillator in [87, 88]. It would be interesting to explore driven dissipative TLLs using methods developed in these works.

**Multi-component TLLs and strings with higher-dimensional target spaces.** It is natural to consider quenches and periodic drives by marginal deformations in $D$-component TLLs, which

have $U(1)^D$ current algebras generated by $J_n^I$ and $\bar{J}_n^I$ ($n \in \mathbb{Z}$, $I = 1, \ldots, D$). These deformations generate the Narain moduli space of the toroidally compactified $D$-component free boson CFT or $D$-dimensional bosonic string. The action is [cf. (2)]

$$S = \frac{1}{4\pi\alpha'} \int \mathrm{d}^2x \left( G_{IJ}\delta^{\alpha\beta} + \mathrm{i}B_{IJ}\epsilon^{\alpha\beta} \right) \partial_\alpha X^I \partial_\beta X^J, \tag{198}$$

where the space of marginal deformations is parametrized by the symmetric target-space metric $G^{IJ}$ and the anti-symmetric Kalb–Ramond field $B^{IJ}$. Applications include the low-energy description of a system of multiple copies of XXZ spin chains with the Hamiltonian [cf. (16)]

$$H = -J \sum_{I,I'=1}^{D} \sum_{j=1}^{N} \left( \delta_{I,I'} S_j^{x,I} S_{j+1}^{x,I'} + \delta_{I,I'} S_j^{y,I} S_{j+1}^{y,I'} - \Delta_{I,I'} S_j^{z,I} S_{j+1}^{z,I'} \right), \tag{199}$$

where $\Delta_{I,I'}$ is the anisotropy matrix, which is modulated in time. It would be interesting to study how the non-equilibrium dynamics depends on the Zamolodchikov distance in this case.

**Compactified orbifold boson CFT.** The moduli space of $c = 1$ CFTs contains two lines (that meet at a point) [2]. The first corresponds to the compactified free boson CFT and is parametrized by the compactification radius. Here we have studied the physical consequences of dynamically exploring this line. The second corresponds to the $\mathbb{Z}_2$ orbifold CFT and is parametrized by the radius of the orbifolded circle. Much of the formalism developed in this work can be adapted to marginal quenches or drives of this second line. Such protocols could be realized in the Ashkin-Teller quantum spin chain.

**Wess-Zumino-Witten (WZW) models.** A large class of CFTs that admit $J\bar{J}$ deformations is provided by $G$-WZW models, which have $\dim(\mathfrak{g})$ holomorphic and anti-holomorphic currents, where $\mathfrak{g}$ is the Lie algebra associated with the compact and simply connected Lie group G. A subset of the current-current deformations formed from these are exactly marginal and generate a moduli space of CFTs. More precisely, a $J\bar{J}$-type deformation is exactly marginal if (and only if) both the holomorphic and anti-holomorphic currents belong to a commutative current algebra [69, 89]. How our results generalize to the dynamics of WZW models under time-dependent exactly marginal deformations is an open question.

**$T\bar{T}$ deformations.** The recently introduced $T\bar{T}$ deformation of CFTs and integrable QFTs [90, 91] provides another arena to explore the dynamics of quenches and drives. This is an irrelevant deformation where the spectrum of the deformed theory is exactly solvable in terms of the undeformed spectrum and degeneracies remain unchanged. A practical starting point to study such dynamics would be to consider a $T\bar{T}$ quench of the free fermion CFT. As the deformation brings about changes in signal propagation velocities, it would be tantalizing to see how individual quantities, such as return probabilities and correlation functions, evolve following the quench. Some work in this direction was carried out in [92].

**Holography.** It would be interesting to consider how the dynamics of marginal quenches and drives translate into bulk or gravitational terms through the AdS/CFT correspondence. In the prototypical example of $AdS_3/CFT_2$ described by the D1-D5 system, the holographic CFT contains exactly marginal operators [93].[20] The marginal deformations in this case allow an interpolation between stringy and (classical) gravity regimes in $AdS_3$. Since we have structures reminiscent of boundary states and conformal interfaces (cf. Sec. 3.4), it is natural to expect that these will have counterparts in the bulk.

---

[20]Exactly marginal operators do not acquire anomalous dimensions upon deformation, i.e., they are protected by supersymmetry.

## Acknowledgments

We are grateful to Ramasubramanian Chitra, Diptarka Das, Eugene Demler, Axel Kleinschmidt, Andrew McLeod, Stefano Scopa, and Pierre Vanhove for fruitful discussions. S.D. thanks ETH Zurich and AEI Potsdam for hospitality during the course of this project. B.L. is supported by the European Research Council (ERC) under the European Union's Horizon 2020 research and innovation program ERC-StG-Neupert-757867-PARATOP. P.M. gratefully acknowledges financial support from the Wenner-Gren Foundations through grant no. WGF2019-0061. A.T. is supported by the Swedish Research Council (VR) through grants no. 2019-04736 and 2020-00214.

## A  Lerch zeta function and regularization

In this appendix, we provide formulas that are crucial in order to regularize the sum in (93). As a first step we re-express the sum in terms of the Lerch zeta function

$$\zeta(s|v,w) = \sum_{m=0}^{\infty}(m+v)^{-s}e^{2\pi imw}. \tag{A.1}$$

Note that the Lerch zeta function reduces to the Riemann zeta function $\zeta(-1)$ at $s=-1$ and $v=w=0$, i.e.,

$$\zeta(-1|0,0) = -\frac{1}{12}. \tag{A.2}$$

The divergence in (93) can be regularized using

$$\sum_{m=1}^{\infty}m e^{2\pi imw} = \zeta(-1|0,w), \tag{A.3}$$

where the right-hand side is defined through the following analytic continuation of the Lerch zeta function [94]:

$$\zeta(s|v,w) = ie^{-2\pi ivw}(2\pi)^{s-1}\Gamma(1-s)\left[e^{-\pi is/2}\zeta(1-s|w,-v) - e^{\pi is/2}e^{2\pi iv}\zeta(1-s|1-w,v)\right]. \tag{A.4}$$

## B  Computational details

### B.1  Primary state contribution to return amplitudes

Below we give an argument as for why the return amplitude starting from a primary state $|h,\bar{h}\rangle$ gives the same result as starting from the ground state $|\Omega\rangle$, i.e., for why

$$\langle h,\bar{h}|\mathcal{I}_{\nu}\left(\mathcal{I}_{\nu}^{(q)}\right)^{\dagger}|h,\bar{h}\rangle = \langle\Omega|\mathcal{I}_{\nu}\left(\mathcal{I}_{\nu}^{(q)}\right)^{\dagger}|\Omega\rangle, \tag{B.1}$$

up to a zero-mode contribution which is an overall phase.

Let us first use the operator-state correspondence and write

$$\langle h,\bar{h}|\mathcal{I}_{\nu}\left(\mathcal{I}_{\nu}^{(q)}\right)^{\dagger}|h,\bar{h}\rangle = \lim_{\substack{z,\bar{z}\to 0 \\ \omega,\bar{\omega}\to 0}}\langle\Omega|O^{\dagger}(\omega,\bar{\omega})\mathcal{I}_{\nu}\left(\mathcal{I}_{\nu}^{(q)}\right)^{\dagger}O(z,\bar{z})|\Omega\rangle, \tag{B.2}$$

where $O(z,\bar{z})$ is a primary field with conformal weights $(h,\bar{h})$. The following commutation relations can be derived from the OPEs of $O(z,\bar{z})$ with the conserved U(1) currents $J(z)$ and

$\bar{J}(\bar{z})$ [cf. Sec. 3.1]:

$$[a_n, O(z,\bar{z})] = q_O z^n O(z,\bar{z}),  \tag{B.3}$$

$$[a_n \bar{a}_n, O(z,\bar{z})] = [\bar{q}_O \bar{z}^n a_n + q_O z^n \bar{a}_n - q_O \bar{q}_O (z\bar{z})^n] O(z,\bar{z}).  \tag{B.4}$$

Here, $q_O$ and $\bar{q}_O$ are the charges of the $U(1)_+$ and $U(1)_-$ current algebras, respectively, for the primary field $O(z,\bar{z})$; equivalently, these are the right and left momenta of the vertex operators.

As a consequence, it is clear that

$$\lim_{z,\bar{z}\to 0} [a_n \bar{a}_n, O(z,\bar{z})] = 0, \qquad \lim_{z,\bar{z}\to 0} \left[ \left( \mathcal{I}_\nu^{(q)} \right)^\dagger, O(z,\bar{z}) \right] = 0,  \tag{B.5}$$

where we used (48), which implies

$$\langle h, \bar{h} | \mathcal{I}_\nu \left( \mathcal{I}_\nu^{(q)} \right)^\dagger | h, \bar{h} \rangle = \lim_{\substack{z,\bar{z}\to 0 \\ \omega,\bar{\omega}\to 0}} \langle \Omega | \mathcal{I}_\nu O^\dagger(\omega,\bar{\omega}) O(z,\bar{z}) \left( \mathcal{I}_\nu^{(q)} \right)^\dagger | \Omega \rangle.  \tag{B.6}$$

Using the fact that the primary states of the compactified free boson CFT are vertex operators, we conclude that $\lim_{z,\bar{z}\to 0} \lim_{\omega,\bar{\omega}\to 0} O^\dagger(\omega,\bar{\omega}) O(z,\bar{z}) = \mathbb{I}$ and thus the equality in (B.1) holds.

## B.2 Solving the Floquet recursion relation

Here we solve the recursion relation in (131) for the matrix in (129), restated here for ease of reference:

$$\begin{pmatrix} I_{1,1}^{(n,M)} & I_{1,2}^{(n,M)} \\ I_{2,1}^{(n,N)} & I_{22}^{(n,M)} \end{pmatrix} = \begin{pmatrix} I_{1,1}^{(n,M-1)} & I_{1,2}^{(n,M-1)} \\ I_{2,1}^{(n,M-1)} & I_{2,2}^{(n,M-1)} \end{pmatrix}$$
$$\times \begin{pmatrix} \cosh^2(\nu) - \sinh^2(\nu)q_2^{2n} & \frac{1}{2}\sinh(2\nu)(1-q_2^{-2n})q_2^{-2(M-1)n}q_1^{-2(M-1)n} \\ \frac{1}{2}\sinh(2\nu)(1-q_2^{2n})q_2^{2(M-1)n}q_1^{2(M-1)n} & \cosh^2(\nu) - \sinh^2(\nu)q_2^{-2n} \end{pmatrix}.  \tag{B.7}$$

Note that it is enough to solve the recursion for $I_{1,1}^{(n,M)}$ and $I_{1,2}^{(n,M)}$, since they are coupled to each other but decoupled from the rest:

$$I_{1,1}^{(n,M)} = \left[ \cosh^2(\nu) - \sinh^2(\nu)q_2^{2n} \right] I_{1,1}^{(n,M-1)}$$
$$+ \frac{1}{2}\sinh(2\nu)\left(1-q_2^{2n}\right)q_2^{2(M-1)n}q_1^{2(M-1)n} I_{1,2}^{(n,M-1)},$$
$$I_{1,2}^{(n,M)} = \frac{1}{2}\sinh(2\nu)\left(1-q_2^{-2n}\right)q_2^{-2(M-1)n}q_1^{-2(M-1)n} I_{1,1}^{(n,M-1)}$$
$$+ \left[ \cosh^2(\nu) - \sinh^2(\nu)q_2^{-2n} \right] I_{1,2}^{(n,M-1)}.  \tag{B.8}$$

The seed conditions for the recursion is $I_{1,1}^{(n,0)} = 1$ and $I_{1,2}^{(n,0)} = 0$. The second equation above can be written more symmetrically as

$$I_{1,2}^{(n,M)} q_2^{2(M-1)n} q_1^{2(M-1)n} = \frac{1}{2}\sinh(2\nu)\left(1-q_2^{-2n}\right) I_{1,1}^{(n,M-1)}$$
$$+ \left[ \cosh^2(\nu) - \sinh^2(\nu)q_2^{-2n} \right] I_{1,2}^{(n,M-1)} q_2^{2(M-1)n} q_1^{2(M-1)n}.  \tag{B.9}$$

Therefore, multiplying by $x^M$ for an arbitrary $x \in \mathbb{R}$,

$$I_{1,1}^{(n,M)} x^M = \left[ \cosh^2(\nu) - \sinh^2(\nu)q_2^{2n} \right] I_{1,1}^{(n,M-1)} x^M$$
$$+ \frac{1}{2}\sinh(2\nu)\left(1-q_2^{2n}\right) I_{1,2}^{(n,M-1)} q_2^{2(M-1)n} q_1^{2(M-1)n} x^M,$$
$$I_{1,2}^{(n,M)} q_2^{2(M-1)n} q_1^{2(M-1)n} x^M = \frac{1}{2}\sinh(2\nu)\left(1-q_2^{-2n}\right) I_{1,1}^{(n,M-1)} x^M$$
$$+ \left[ \cosh^2(\nu) - \sinh^2(\nu)q_2^{-2n} \right] I_{1,2}^{(n,M-1)} q_2^{2(M-1)n} q_1^{2(M-1)n} x^M.  \tag{B.10}$$

Summing over $M$ from 1 to $\infty$, we obtain

$$
\begin{aligned}
I_{1,1}^{(n)}(x) - 1 &= \left[\cosh^2(v) - \sinh^2(v)q_2^{2n}\right]xI_{1,1}^{(n)}(x) \\
&\quad + \frac{1}{2}\sinh(2v)\left(1 - q_2^{2n}\right)xI_{1,2}^{(n)}(q_2^{2n}q_1^{2n}x), \\
q_2^{-2n}q_1^{-2n}I_{1,2}^{(n)}(q_2^{2n}q_1^{2n}x) &= \frac{1}{2}\sinh(2v)\left(1 - q_2^{-2n}\right)xI_{1,1}^{(n)}(x) \\
&\quad + \left[\cosh^2(v) - \sinh^2(v)q_2^{-2n}\right]xI_{1,2}^{(n)}(q_2^{2n}q_1^{2n}x),
\end{aligned}
\tag{B.11}
$$

where we defined the generating functions

$$
I_{1,1}^{(n)}(x) = \sum_{M=0}^{\infty} I_{1,1}^{(n,M)}x^M, \qquad I_{1,2}^{(n)}(x) = \sum_{M=0}^{\infty} I_{1,2}^{(n,M)}x^M.
\tag{B.12}
$$

The solutions to the generating functions are

$$
\begin{aligned}
I_{1,1}^{(n)}(x) &= \frac{2 - 2xq_1^{2n}\left[\cosh^2(v)q_2^{2n} - \sinh^2(v)\right]}{2x^2q_2^{2n}q_1^{2n} - x\left(1 + q_2^{2n}\right)\left(1 + q_1^{2n}\right) - x\cosh(2v)\left(1 - q_2^{2n}\right)\left(1 - q_1^{2n}\right) + 2}, \\
I_{1,2}^{(n)}(q_2^{2n}q_1^{2n}x) &= \frac{x\sinh(2v)\left(q_2^{2n} - 1\right)q_1^{2n}}{2x^2q_2^{2n}q_1^{2n} - x\left(1 + q_2^{2n}\right)\left(1 + q_1^{2n}\right) - x\cosh(2v)\left(1 - q_2^{2n}\right)\left(1 - q_1^{2n}\right) + 2}.
\end{aligned}
\tag{B.13}
$$

We note that the first expression can be rewritten as

$$
I_{1,1}^{(n)}(x) = \frac{1 - \alpha_n q_2^n q_1^n x}{1 - \beta_n q_2^n q_1^n x + (q_2^n q_1^n x)^2} = \frac{1 - \alpha_n q_2^n q_1^n x}{(q_2^n q_1^n x - [\beta_n - \gamma_n]/2)(q_2^n q_1^n x - [\beta_n + \gamma_n]/2)},
\tag{B.14}
$$

with

$$
\begin{aligned}
\alpha_n &= \left[\cosh^2(v)q_2^n - \sinh^2(v)q_2^{-n}\right]q_1^n, \\
\beta_n &= \frac{\left(q_1^n + q_1^{-n}\right)\left(q_2^n + q_2^{-n}\right) + \left(q_1^n - q_1^{-n}\right)\left(q_2^n - q_2^{-n}\right)\cosh(2v)}{2}, \\
\gamma_n &= \sqrt{\beta_n^2 - 4}.
\end{aligned}
\tag{B.15}
$$

Noting that $(\beta_n \pm \gamma_n)/2 = \lambda_n^{\pm}$ in (123) and re-expanding $I_{1,1}^{(n)}(x) = \sum_{M=0}^{\infty} I_{1,1}^{(n,M)}x^M$ as a power series in $x$, we obtain

$$
I_{1,1}^{(n,M)} = (q_2^n q_1^n)^M \frac{(\alpha_n - \lambda_n^-)(\lambda_n^-)^M - (\alpha_n - \lambda_n^+)(\lambda_n^+)^M}{\sqrt{\sigma_n - 4}}.
\tag{B.16}
$$

Finally, using that $\alpha_n = \omega_n - i(\varepsilon_n/2)\sqrt{4 - \sigma_n^2}$ in terms of $\omega_n$ in (122) and $\varepsilon_n$ in (133), the desired result for $I_{1,1}^{(n,M)}$ in (132) follows.

## B.3  SU(1, 1) matrix elements for $M$-cycle rotations

Below we explain how to obtain analytical expressions for $\mathcal{A}_n = \mathcal{A}_n(M)$ and $\mathcal{B}_n = \mathcal{B}_n(M)$ in (147). We recall that these are defined by products of SU(1, 1) matrices,

$$
\begin{pmatrix} \mathcal{A}_n & \mathcal{B}_n \\ \overline{\mathcal{B}_n} & \overline{\mathcal{A}_n} \end{pmatrix} = \prod_{j=M-1}^{0} \mathsf{T}_j, \qquad \begin{pmatrix} \overline{\mathcal{A}_n} & -\mathcal{B}_n \\ -\overline{\mathcal{B}_n} & \mathcal{A}_n \end{pmatrix} = \prod_{j=0}^{M-1} \mathsf{T}_j^{-1},
\tag{B.17}
$$

implementing $M$-cycle rotations, where $\mathsf{T}_j$ and $\mathsf{T}_j^{-1}$ are given by (145) and (146), respectively. To compute such products explicitly, consider a general matrix of the form

$$\mathsf{A}_j = \begin{pmatrix} a & b\mathrm{e}^{-\mathrm{i}\phi j} \\ \overline{b}\mathrm{e}^{\mathrm{i}\phi j} & \overline{a} \end{pmatrix} \tag{B.18}$$

for $a, b \in \mathbb{C}$ and $\phi \in \mathbb{R}$. Noting that

$$\mathsf{A}_j = \begin{pmatrix} 0 & \mathrm{e}^{-\mathrm{i}(\phi/2)j} \\ \mathrm{e}^{\mathrm{i}(\phi/2)j} & 0 \end{pmatrix} \begin{pmatrix} \overline{a} & \overline{b} \\ b & a \end{pmatrix} \begin{pmatrix} 0 & \mathrm{e}^{-\mathrm{i}(\phi/2)j} \\ \mathrm{e}^{\mathrm{i}(\phi/2)j} & 0 \end{pmatrix} = \mathsf{S}_j \mathsf{B} \mathsf{S}_j \tag{B.19}$$

and defining $\mathsf{D} = \mathsf{S}_{j-1}\mathsf{S}_j = \mathrm{diag}(\mathrm{e}^{\mathrm{i}\phi/2}, \mathrm{e}^{-\mathrm{i}\phi/2})$, products of $\mathsf{A}_j$s can be written

$$\begin{aligned} \mathsf{A}_0 \mathsf{A}_1 \ldots \mathsf{A}_{M-1} &= \mathsf{S}_0 (\mathsf{B}\mathsf{D})^{M-1} \mathsf{B} \mathsf{S}_{M-1}, \\ \mathsf{A}_{M-1} \ldots \mathsf{A}_1 \mathsf{A}_0 &= \mathsf{S}_{M-1} \mathsf{B} (\overline{\mathsf{D}}\mathsf{B})^{M-1} \mathsf{S}_0. \end{aligned} \tag{B.20}$$

Thus, the products of matrices in (B.17) implementing the $M$-cycle rotations can be expressed as simpler products of new matrices, which straightforwardly can be used to compute explicit coefficients for the $M$-cycle rotation of $a_{-n}$ and $\bar{a}_n$.

## B.4 Evaluation of $\mathrm{Tr}\!\left[\mathrm{e}^{-\beta H} a_0 \bar{a}_0\right]$

We encountered the following trace in (196) while calculating the relative entropy:

$$\mathrm{Tr}\!\left[\mathrm{e}^{-\beta H} \sum_{n=-\infty}^{\infty} a_n \bar{a}_n\right] = \mathrm{Tr}\!\left[\mathrm{e}^{-\beta H} a_0 \bar{a}_0\right]. \tag{B.21}$$

The above object can be evaluated in general by taking derivatives with respect to the chemical potentials of the flavoured partition function

$$Z(\tau, \bar{\tau}, \chi, \bar{\chi}) = \mathrm{Tr}\!\left[q^{L_0 - c/24} \bar{q}^{\bar{L}_0 - c/24} \mathrm{e}^{2\pi\mathrm{i}\chi a_0} \mathrm{e}^{-2\pi\mathrm{i}\bar{\chi}\bar{a}_0}\right], \tag{B.22}$$

$$\frac{1}{4\pi^2} \partial_\chi \partial_{\bar{\chi}} Z(\tau, \bar{\tau}, \chi, \bar{\chi})\big|_{\chi,\bar{\chi}=0} = \mathrm{Tr}\!\left[q^{L_0 - c/24} \bar{q}^{\bar{L}_0 - c/24} a_0 \bar{a}_0\right], \tag{B.23}$$

where $q = \mathrm{e}^{2\pi\mathrm{i}\tau}$ and $\chi$ and $\bar{\chi}$ are chemical potentials conjugate to the right and left U(1) currents, respectively. As we are interested in the high-temperature regime, we need the S-modular transformation of the partition function above. It is well known that the object transforms in a manner similar to a weak Jacobi form [95, Appendix A]:

$$Z(\tau, \bar{\tau}, \chi, \bar{\chi}) = \exp\!\left(-\frac{\mathrm{i}\pi\chi^2}{\tau} + \frac{\mathrm{i}\pi\bar{\chi}^2}{\bar{\tau}}\right) Z\!\left(-\frac{1}{\tau}, -\frac{1}{\bar{\tau}}, \frac{\chi}{\tau}, \frac{\bar{\chi}}{\bar{\tau}}\right), \tag{B.24}$$

where $\tau = \mathrm{i}\beta/L$ in our case. The dominant contribution at low temperatures arises from the vacuum. S-modular transforming this result using the above, we get the universal high-temperature behavior

$$Z(\beta/L, \chi, \bar{\chi}) \approx \exp\!\left(-\frac{\mathrm{i}\pi L(\chi^2 + \bar{\chi}^2)}{\beta} + \frac{\pi L}{6\beta}\right) \tag{B.25}$$

for $\beta/L \ll 1$. Corrections beyond this are exponentially suppressed. Now taking derivatives and setting the chemical potentials to zero, as prescribed by the second equation in (B.22), we get $\mathrm{Tr}\!\left[\mathrm{e}^{-\beta H} a_0 \bar{a}_0\right] = 0$ in the thermodynamic limit.

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
