# Peer review of "Marginal quenches and drives in Tomonaga-Luttinger liquids"

_SciPost Physics, doi:SciPost Phys. 14, 108 (2023)_

## Round 1 · Referee Report · Anonymous (Referee 1) · 2022-10-6

Strengths

Very detailed paper

Weaknesses

References not always complete for the paper to be self-contained

Report

The paper studies the dynamics of Tomonaga-Luttinger liquids (TLLs) after quenching or periodically driving the Luttinger parameter (equivalently, the compactification radius). Using Bogoliubov transformation and an underlying su(1,1) algebraic structure, various exact results are obtained. These includes results for the Loschmidt echo and the energy density in the aforementioned protocols. A relatively disconnected but interesting section is Section 6, where the same methods are applied to the calculation of Renyi divergences and relative entropy, via the replica limit, between two equilibrium thermal states of two different TLLs.

The paper contains interesting calculations which go clearly beyond the known results in the TLL physics. Moreover, it is clearly written and contains rather detailed calculations.

What is to be slightly improved are references, in particular:

  • Recently, other relevant works considering quenches in TTL appeared, e.g., arXiv:2203.06740

  • In Section 3, some more references could be included, as the material can be non-standard for physicists which are not mathematical physicists

  • When discussing relative entropy, there are no references about the initial works where the replica approach has been used in the context of QFT (more specifically CFT), e.g., Phys. Rev. Lett. 113, 051602 (2014), Phys. Rev. Lett. 117, 041601, JHEP 02 (2017) 039

Once these are included, I recommend the paper for publication in Scipost Physics.

  • validity: high
  • significance: good
  • originality: good
  • clarity: good
  • formatting: excellent
  • grammar: excellent

Author:  Per Moosavi  on 2023-01-11  [id 3229]

(in reply to Report 1 on 2022-10-06)

We thank the referee for their positive recommendation. We now added several references as suggested by the referee. In particular, we added four references in the introduction, where we discuss additional recent past works on interaction quenches in Tomonaga-Luttinger liquids, and added several references and a clarifying footnote in Sec. 3 which would aid the reader unfamiliar with those tools. We also added the suggested references related to initial works where the replica approach was used to study relative entropy in QFT.

Please see the list of changes for details.

---

## Round 1 · Referee Report · Anonymous (Referee 2) · 2022-10-16

Strengths

Clearly written paper

Report

The paper involves studying a non-equilibrium exactly solvable problem, namely a Luttinger liquid. By now this is rather a matured field, yet the authors do arrive at some interesting new observations. Perhaps the most interesting is the Floquet analysis and the study of the Renyi divergence.

In the Floquet part of the paper, I would suggest the authors do the following
a). compare their results with the famous stability phase diagram of the Matheiu equation, well known from the 1940s.
b). Explicitly compare their exponent and phase diagram with that obtained by Eggert et al ( a paper that has been cited, but not discussed).

After these two changes, the paper may be accepted for publication.
  • validity: -
  • significance: -
  • originality: -
  • clarity: -
  • formatting: -
  • grammar: -

Author:  Per Moosavi  on 2023-01-11  [id 3230]

(in reply to Report 2 on 2022-10-16)

We thank the referee for their useful suggestions and positive recommendation. We now added an explanation as well as a new Figure (Fig. 9 in the updated manuscript) contextualizing the stability analysis carried out in our work with past works (in particular Phys. Rev. Lett. 126, 243401 by Eggert et al.) related to the stability phase diagram of the quantum parametric oscillator. Specifically, the new Fig. 9 contains a stability diagram that parallels Fig. 2 of Phys. Rev. Lett. 126, 243401 by identifying (i) the ratio of Luttinger parameters $K_1/K_2$ as the amplitude in the related continuous drive and (ii) the phase argument of the eigenvalues of the 2x2 su(1,1) matrix obtained from the Floquet unitary as the Mathieu characteristic exponent. We note that it is interesting that such a correspondence can indeed be made due to the shared underlying su(1,1) algebraic structure despite the fact that we use a step-like Floquet protocol as opposed to the quantum parametric oscillator, which is continuously driven.

Please see the list of changes for details.

---

## Round 2 · Author Response

Resubmission with minor adjustments, fixed typos, improved discussions, and added references following the referee reports. We thank both referees for their comments and positive recommendations.

---

## Round 2 · List of Changes

1. In Sec. 1: a) Added the following four references on interaction quenches in Tomonaga-Luttinger liquids (TLLs):
  2. Bernier et al., Phys. Rev. Lett. 112 (2014) 065301.
  3. Dóra and Pollmann, Phys. Rev. Lett. 115 (2015) 096403.
  4. Ruggiero et al., SciPost Phys. 13 (2022) 111, mentioned by Referee 1.
  5. Moosavi, arXiv:2208.14467. b) Added the following reference on conformal interfaces:
  6. Bachas et al., J. High Energ. Phys. 2002 (2002) 027.

  7. In Sec. 3: Added several references in Sec. 3 that would aid the reader with potentially unfamiliar topics. The added references include: a) For our overall notation and introduction to CFT in the beginning of Sec. 3, we referenced:

  8. Di Francesco, Mathieu, and Senechal, Conformal field theory (1997),
  9. Gawƒôdzki et al., J. Stat. Phys. 172 (2018) 353.
  10. Moosavi, Ann. Henri Poincaré (2021). b) Added the following reference discussing marginal operators and their conformal weight in Sec. 3.1:
  11. Ginsparg, Nucl. Phys. B 295 (1988) 153. c) Added the following reference for the Sugawara construction in Sec. 3.2:
  12. Di Francesco, Mathieu, and Senechal, Conformal field theory (1997). d) Added the following new references for the su(1,1) algebra in Sec. 3.3:
  13. Perelomov, Generalized coherent states and their applications (1986).

  14. Other changes in Sec. 3: a) Clarified the definition of the Zamolodchikov metric in Sec. 3.1. b) Added Footnote 9 at the end of the first paragraph in Sec. 3.5 to make the discussion more self-contained. c) Updated the notation for the su(1,1) generators. d) Added explicit expression for the su(1,1) Cartan-Killing form as the new Eq. (3.21).

  15. In Sec 5: a) Added brief remarks in the beginning about the parallels with quantum parametric oscillators to put the use of the su(1,1) algebraic tools in context, together with the following two new references:

  16. Perelomov and Popov, Theor. Math. Phys. 1, (1969) 275.
  17. Gritsev and Polkovnikov, SciPost Phys. 2 (2017) 021. b) Added a new paragraph and a new figure (Fig. 9) in Sec. 5 describing the connection between our stability analysis and the stability phase diagram obtained for the continuously driven TLL based on the Mathieu equation. c) Corrected sign typos in formulas in the beginning of Sec. 5.

  18. In Sec.6: a) Added the following references for the replica approach used to study relative entropy in QFT:

  19. Lashkari, Phys. Rev. Lett. 113, (2014) 051602.
  20. Lashkari, Phys. Rev. Lett. 117, (2016) 041601.
  21. Ruggiero and Calabrese, JHEP 02 (2017) 039. b) Updated the discussion in Sec. 6.1 to improve consistency and clarity.

  22. In Sec. 7: Added a discussion on extensions of our work to driven dissipative TLLs.

  23. Optimized the text at various places.

---

## Editorial Decision

published